# Relational Structural Causal Models

**Adiba Ejaz** [1]   **Elias Bareinboim** [1]

## Abstract

An artificial intelligence must have a model of its environment that is *causal*, supporting reasoning about interventions and counterfactuals, and also *combinatorial*, supporting generalization to unseen combinations of objects. In this work, we formally study when and how such a model can be learned. We develop *relational structural causal models*, extending structural causal models (Pearl, 2009) to settings where objects and their relations vary. First, we show how answers to not only causal but also observational queries about unseen combinations of objects can not be identified without further assumptions. To enable such identification—including in the presence of unobserved confounding—we define *relational causal graphs* and derive symbolic identification criteria. Finally, we propose *relational neural causal models*, a provably correct approach that outperforms non-relational baselines on simulated traffic scenes with varying cars, signals, and pedestrians.

## 1. Introduction

Behind a Rube Goldberg machine is a sequence of simple mechanisms. A ball rolls down a ramp, tipping a weight, pulling a string, swinging a hammer, and striking a gong. Predicting what happens next and why requires a model of how these bodies interact. This is precisely what *world models* aim to provide for AI systems to learn efficiently and generalize across environments (Ha & Schmidhuber, 2018; LeCun, 2022; Gurnee & Tegmark, 2024; Richens & Everitt, 2024; Vafa et al., 2024). In this work, we consider two important problems that such a model must address.

The first problem is that of representing objects and composing them via relations (Battaglia et al., 2018; Lake et al., 2017; Tenenbaum et al., 2011; Chollet, 2019). Downstream of such representations is the ability to answer questions about unseen combinations of objects, e.g., a new Rube Goldberg machine with an added ramp. Such combinatorial structure arises in many domains. Robots must reason about varying types of objects and their spatial relations to navigate and manipulate the world (Li et al., 2019; Wang et al., 2025b; Locatello et al., 2020); language is generated from unbounded combinations of nouns related by verbs (Chomsky, 1965); and biological systems are naturally described in terms of interacting proteins, metabolites, and cells (Barabási et al., 2011; Veličković, 2023; Regev et al., 2017). The generality of this problem has inspired active research into relational and object-centric machine learning (Koller & Friedman, 2009; Getoor & Taskar, 2007; Veličković et al., 2018; Kipf & Welling, 2017; Zambaldi et al., 2018) aimed at such combinatorial generalization.

The second problem is that of answering causal questions: what if the weight were lighter, the string were cut, or the ramp angle were changed in our Rube Goldberg machine? A common view is that such questions cannot be answered from observations of the environment alone, requiring either interventions, or *causal* inductive biases, or often both (Pearl, 2009; Pearl & Mackenzie, 2018; Bareinboim et al., 2022; Schölkopf et al., 2021; Bareinboim, 2025). In our case, evidence for this point of view comes from the weaknesses of relational machine learning. Despite strong in-distribution predictive performance, relational and non-relational methods alike can still exploit correlations that are unstable under interventions or distribution shift (de Haan et al., 2019; Park et al., 2021; Fan et al., 2022; Wu et al., 2022; Vo et al., 2025). Relational structure alone does not guarantee answers to causal questions.

Causal machine learning methods aim to answer such questions in the contexts of decision-making, generative modeling, fairness, and more (Schölkopf, 2022; Plečko & Bareinboim, 2024; Bareinboim et al., 2024; Pan & Bareinboim, 2024; 2025). However, many of these results do not easily lend themselves to combinatorial generalization because they rely on a fixed causal graph over a fixed set of variables. In domains such as autonomous driving, where traffic scenes differ in how many signals, pedestrians and cars appear and how they relate (Fig. 1), causal methods must make assumptions that simplify these relations. They often assume that objects are unrelated (i.i.d.) or that the rela-

---

[1]Causal AI Lab, Department of Computer Science, Columbia University, NY, USA. Correspondence to: Adiba Ejaz <adiba.ejaz@cs.columbia.edu>.

*Proceedings of the 43^{rd} International Conference on Machine Learning*, Seoul, South Korea. PMLR 306, 2026. Copyright 2026 by the author(s).

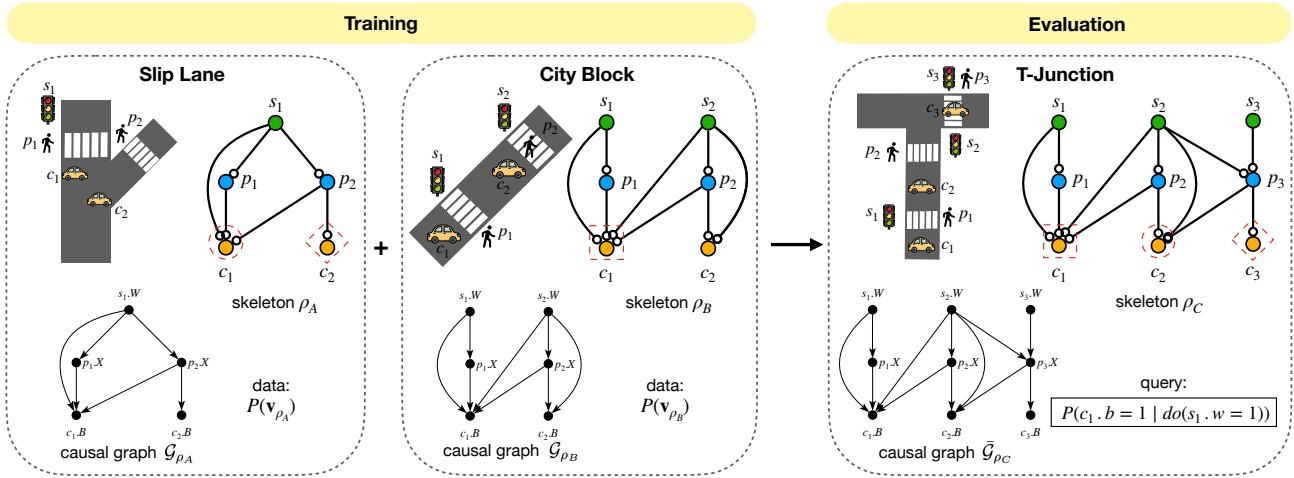

*Figure 1.* A schematic for the problem of relational identification across varying traffic scenes, following the schema in Ex. 1. Each panel shows: (i) a relational skeleton $\rho$ representing a particular combination of signals ($s$), pedestrians ($p$), and cars ($c$), (ii) the corresponding causal graph $\bar{\mathcal{G}}_\rho$, and (iii) the available data or the query of interest. The goal is to use data from the *source* skeletons $\rho_A$ and $\rho_B$ to answer a query about the *target* skeleton $\rho_B$. In any $\rho$, an edge $(s_i, p_j)$ indicates that signal $s_i$ controls pedestrian $p_j$; an edge $(s_i, c_j)$ indicates that signal $s_i$ controls car $c_j$; and an edge $(p_i, c_j)$ indicates that pedestrian $p_i$ is in the path of car $c_j$. A bubble marks the second tuple element. Cars with similar relational neighborhoods skeletons are circumscribed by similar shapes (red dotted line).

tional structure is fixed. Causal relational learning (Lee & Honavar, 2016; Maier et al., 2010; Salimi et al., 2020) and methods for causal inference from non-i.i.d. data (Rubin, 1990; Sobel, 2006; Hudgens & Halloran, 2008; Ogburn & VanderWeele, 2014; Weinstein & Blei, 2024) make progress towards relaxing this assumption, but assume that the objects and their relations are fixed. For instance, they study what can be inferred about a given traffic scene given data from that same scene, but not data from a different scene. As such, they do not address the problem of generalizing across combinations where both object counts and their relations can vary.

**Contributions.** We develop causal models for object-relational settings, enabling causal inference across varying combinations of objects. More specifically, our contributions are as follows.

1. **Relational SCMs.** In Sec. 3, we formalize how different combinations of objects can be unified by the same data-generating process: a relational structural causal model (Def. 3.1). Based on this formalization, we prove the limits of learning distributions over seen and unseen combinations of objects, showing that even observational distributions of unseen combinations cannot be learned without further assumptions.

2. **Graphical identification.** In Sec. 4, given the previous impossibility results, we introduce a graphical language for encoding assumptions that enable *relational identification* (Def. 4.2) across combinations of objects. We show when and how existing causal inference tools can be used for this task.

3. **Relational neural causal models.** In Sec. 5, we develop *relational neural causal models* (Def. 5.1) which form the basis of a sound and complete neural approach for relational identification in practice.

Experiments with simulated traffic scenes (Sec. 6, Sec. E) support our findings. We give an extended discussion of related literature in Sec. A; further definitions and examples, including a comparison with standard SCMs in Secs. B and C; as well as proofs and further results in Sec. D.

## 2. Preliminaries

**Notation.** Capital letters $(X)$ denote variables, $\mathrm{dom}(X)$ denotes their domains, small letters $(x)$ denote values in their domains, and bold letters denote sets of variables $(\mathbf{X})$ and their values $(\mathbf{x})$. $P(\mathbf{X})$ denotes the probability distribution over a set of variables $\mathbf{X}$. We consistently use $P(\mathbf{x})$ to abbreviate probabilities $P(\mathbf{X} = \mathbf{x})$.

**Structural causal models.** Our framework extends that of *structural causal models* (SCMs) (Pearl, 2009; Bareinboim, 2025), a formalism for data-generating processes. An SCM $\mathcal{M}$ is a four-tuple $\mathcal{M} = \langle \mathbf{V}, \mathbf{U}, \mathcal{F}, P(\mathbf{U}) \rangle$ where $\mathbf{V}$ and $\mathbf{U}$ are sets of endogenous (observed) and exogenous (unobserved) variables respectively. $\mathcal{F}$ is a set of mechanisms: each $V \in \mathbf{V}$ takes the value $f_V(\mathbf{pa}_V, \mathbf{u}_V)$, a function of the values of its endogenous and exogenous parents, $\mathbf{Pa}_V \subseteq \mathbf{V}$ and $\mathbf{U}_V \subseteq \mathbf{U}$, respectively. $P(\mathbf{U})$ is a joint distribution over $\mathbf{U}$; as in prior work (Zhang et al., 2022b; Xia et al., 2021), we assume the variables in $\mathbf{U}$ are jointly independent, although a given $U$ may affect more than one $V$.

Every SCM induces a *causal graph*, constructed as follows: (1) add a vertex for every $V \in \mathbf{V}$ (2) add an edge $V_i \rightarrow V_j$ for every $V_i, V_j \in \mathbf{V}$ if $V_i \in \mathbf{Pa_{V_j}}$ (3) add a dashed bidirected edge between $V_i, V_j$ if $\mathbf{U}_{V_i} \cap \mathbf{U}_{V_j} \neq \emptyset$. See Sec. B for additional background.

**Objects and relations.** We build on the entity-relationship (ER) model (Ullman & Widom, 2002). A *relational schema* is a 3-tuple $\mathcal{S} = \langle \mathcal{E}, \mathcal{R}, \mathcal{A} \rangle$ where $\mathcal{E}$ is a set of entity (or object) types; $\mathcal{R}$ a set of relation types over $\mathcal{E}$; and $\mathcal{A}$ a set of observed attribute types $O.A$ for types $O \in \mathcal{E} \cup \mathcal{R}$. A *relational skeleton* $\rho$ of $\mathcal{S}$ is a finite set of ground entities and relations $o$ of the specified types $O \in \mathcal{E} \cup \mathcal{R}$. We write $\rho(O)$ for the set of instances $o$ in $\rho$ of type $O$.

*Example* 1 (Relational schema and skeleton for traffic scene). A simple relational schema for traffic scenes would be

$$\mathcal{E} = \{\mathsf{Signal\ (Sig),\ Car\ (Car),\ Pedestrian\ (Ped)}\}$$
$$\mathcal{R} = \{\mathsf{Ctrl(Sig, Ped), Ctrl(Sig, Car), Path(Ped, Car)}\}$$
$$\mathcal{A} = \{\mathsf{Sig}.W, \mathsf{Ped}.X, \mathsf{Car}.B\},$$

with all attributes binary-valued: $\mathsf{Sig}.W \in \{1, 0\}$ denotes walk/drive; $\mathsf{Ped}.X \in \{1, 0\}$ cross/wait; and $\mathsf{Car}.B \in \{1, 0\}$ brake/go. $\mathsf{Ctrl(Sig, Ped)}$ (resp. $\mathsf{Ctrl(Sig, Car)}$) indicates that a signal controls a pedestrian (resp. car), and $\mathsf{Path(Ped, Car)}$ indicates that the pedestrian is in the car's path. Figure 1 shows three skeletons for three traffic scenes, e.g., in $\rho_A$ (slip lane), one signal $s_1$ controls pedestrians $p_1, p_2$ and car $c_1$ (but not $c_2$). □

## 3. Defining and Characterizing Relational SCMs

In this section, we introduce relational structural causal models (RSCMs) with the goal of specifying a data-generating process that underlies varying combinations of objects.

### 3.1. Defining Relational SCMs

An RSCM generalizes a standard SCM in two ways. First, different types of objects carry different attributes, and hence different sets of variables in an RSCM. Second, an attribute of one object may affect that of another only when the two objects stand in a particular relation. Following previous work on relational modeling (Koller & Friedman, 2009), we capture such contingent dependencies using *relational constraints* (Def. B.1).

*Example* 2 (Relational constraints for traffic scene). In Ex. 1, consider a signal Sig, pedestrian Ped, and car Car. In skeleton $\rho_A$, the constraint $\phi : \mathsf{Ctrl(Sig, Car)}$ is true for $\mathsf{Sig} = s_1$ and $\mathsf{Car} = c_1$ but not $\mathsf{Car} = c_2$. In $\rho_B$, the constraint $\phi' : \mathsf{Path(Ped, Car)}$ is true for $\mathsf{Ped} = p_1$ and $\mathsf{Car} = c_1$ but not $\mathsf{Car} = c_2$. □

An RSCM specifies one mechanism per attribute (e.g., for

whether a car brakes). Its output can depend on the attributes of related objects (e.g., the crossing states of all pedestrians in the car's path), possibly via aggregation (Def. B.2).

**Definition 3.1** (Relational structural causal model (RSCM)). A *relational structural causal model* (RSCM) is a 5-tuple $\mathcal{M} = \langle \mathcal{S}, \mathbf{V}, \mathbf{U}, \mathcal{F}, P(\mathbf{U}) \rangle$, where $\mathcal{S} = \langle \mathcal{E}, \mathcal{R}, \mathcal{A} \rangle$ is a relational schema; $\mathbf{V}$ is a set of endogenous variables $O.A$ for each attribute $O.A$ in $\mathcal{A}$; $\mathcal{F}$ is a set of mechanisms $f_{O.A}$ for each variable $O.A$ in $\mathbf{V}$; $\mathbf{U}$ is a set of exogenous variables $O.U$ tied to objects $O \in \mathcal{E} \cup \mathcal{R}$; and $P(\mathbf{U})$ is a probability distribution over $\mathbf{U}$ factorizing as $P(\mathbf{U}) = \prod_{O.U \in \mathbf{U}} P(O.U)$. Each mechanism has the form

$$O.A \leftarrow f_{O.A}(\mathbf{Pa}_{O.A}, \mathbf{U}_{O.A}, \mathbf{Pa}^r_{O.A}, \mathbf{U}^r_{O.A}).$$

Here, $\mathbf{Pa}_{O.A} \subseteq \mathbf{V}$ and $\mathbf{U}_{O.A} \subseteq \mathbf{U}$ are *non-relational parents* comprising attributes of the same object instance. On the other hand, $\mathbf{Pa}^r_{O.A}$ and $\mathbf{U}^r_{O.A}$ are *relational parents*. Each endogenous relational parent is a tuple $(\mathbf{W}, \phi, \mathrm{AGG})$, where $\mathbf{W} \subseteq \mathbf{V}$ are variables belonging to some type $T \in \mathcal{E} \cup \mathcal{R}$; $\phi$ is a relational constraint over entities associated with $O$ and $T$; and AGG is an optional list of aggregators for each $T.W \in \mathbf{W}$. Exogenous relational parents are analogous.

*Example* 3 (RSCM for traffic scene). Continuing Ex. 1 and 2, we define an RSCM for traffic scenes. The endogenous variables are $\mathbf{V} = \{\mathsf{Sig}.W,\ \mathsf{Ped}.X,\ \mathsf{Car}.B\}$. The exogenous variables $\mathbf{U}$ are $\mathsf{Sig}.U_W \sim \mathcal{B}(0.3), \mathsf{Ped}.U_X \sim \mathcal{B}(0.4)$ and $\mathsf{Car}.U_B \sim \mathcal{B}(0.2)$, capturing unobserved factors such as a pedestrian's intent to cross or a driver's alertness. The mechanisms are

$$\mathsf{Sig}.W \leftarrow \mathsf{Sig}.U_W,$$
$$\mathsf{Ped}.X \leftarrow \mathsf{Ped}.U_X \oplus \bigwedge_{\mathsf{Ctrl(Sig,Ped)}} \mathsf{Sig}.W,\ \text{and}$$
$$\mathsf{Car}.B \leftarrow \mathsf{Car}.U_B \oplus \left( \bigvee_{\mathsf{Ctrl(Sig,Car)}} \mathsf{Sig}.W \right.$$
$$\left. \vee \bigvee_{\mathsf{Path(Ped,Car)}} \mathsf{Ped}.X \right).$$

For example, $\mathsf{Ped}.X$ has the non-relational parent $\mathsf{Ped}.U_X$ and relational parent $(\{\mathsf{Sig}.W\}, \mathsf{Ctrl(Sig, Ped)}, \wedge)$. Intuitively, $f_{\mathsf{Ped}.X}$ makes a pedestrian cross when all controlling signals are in the 'walk' state, up to noise (e.g., the pedestrian does not intend to cross). Similarly, $f_{\mathsf{Car}.B}$ makes a car brake when any controlling signal says 'walk' or any pedestrian in its path is crossing, again up to noise (e.g., the driver is not alert). See Ex. 10 for an extended example. □

Note how mechanisms in an RSCM differ from those in an SCM.[1] Since the number of objects satisfying a constraint

---

[1]We give a side-by-side comparison of RSCMs with standard SCMs for the traffic example in Table B.3.1, as well as for an additional example in Sec. C.1.

(e.g., pedestrians in a car's path) can vary across skeletons, each $f_{O.A}$ in an RSCM must accept multisets of varying size, while in an SCM, $f_{O.A}$ accepts a fixed-size input. In practice, relational learning often uses permutation-invariant *aggregators* (Def. B.2) such as mean, sum, max, majority, attention pooling, etc. to implement functions on sets.

An RSCM $\mathcal{M}$ may additionally be *Markovian*.

**Definition 3.2** (RSCM Markovianity). We say an RSCM $\mathcal{M} = \langle \mathcal{S}, \mathbf{V}, \mathbf{U}, \mathcal{F}, P(\mathbf{U}) \rangle$ is *$\rho$-Markovian* if for each variable $O.A \in \mathbf{V}$, the set of exogenous relational parents $\mathbf{U}^r_{O.A}$ is empty. We say $\mathcal{M}$ is *Markovian* if it is $\rho$-Markovian and no two variables $O.A, T.B \in \mathbf{V}$ share a non-relational exogenous parent.

An RSCM can be instantiated for any skeleton $\rho$. It induces a standard *ground* RSCM (Def. B.4) with a ground variable $o.A$ for each attributes $O.A$ and instance $o \in \rho(O)$. The function determining $o.A$ substitutes the relational parents in $f_{O.A}$ with ground variables $t.W$ where $o$ and $t$ stand in the required relation.

*Example* 4 (Ground RSCM for traffic scene). For the RSCM $\mathcal{M}$ in Ex. 3 and skeleton $\rho_A$ in Fig. 1, the ground RSCM $\mathcal{M}_{\rho_A} = \langle \mathbf{V}_{\rho_A}, \mathbf{U}_{\rho_A}, \mathcal{F}_{\rho_A}, P(\mathbf{U}_{\rho_A}) \rangle$ is as follows.

$$\mathbf{V}_{\rho_A} = \{s_1.W, \ p_1.X, \ p_2.X, \ c_1.B, \ c_2.B\}$$
$$\mathbf{U}_{\rho_A} = \{s_1.U_W, \ p_1.U_X, \ p_2.U_X, c_1.U_B, \ c_2.U_B\}$$
$$s_1.W \leftarrow s_1.U_W$$
$$p_1.X \leftarrow p_1.U_X \oplus \bigwedge\{s_1.W\}$$
$$p_2.X \leftarrow p_2.U_X \oplus \bigwedge\{s_1.W\}$$
$$c_1.B \leftarrow c_1.U_B \oplus \left( \bigvee\{s_1.W\} \vee \bigvee\{p_1.X, p_2.X\} \right)$$
$$c_2.B \leftarrow c_2.U_B \oplus \left( \bigvee \emptyset \vee \bigvee\{p_2.X\} \right)$$

with $s_1.U_W \sim \mathcal{B}(0.3)$; $p_1.U_X, \ p_2.U_X \sim_{\text{iid}} \mathcal{B}(0.4)$; and $c_1.U_B, \ c_2.U_B \sim_{\text{iid}} \mathcal{B}(0.2)$. $\mathcal{M}_{\rho_A}$ describes the generative process for various traffic scenes with the structure $\rho_A$. $\square$

We assume, throughout, that for any skeleton $\rho$, the ground RSCM $\mathcal{M}_\rho$ is recursive (or acyclic, Def. B.3).[2]

### 3.2. Limits of Learning Relational SCMs

In most domains, the true data-generating process, or RSCM, is unknown (Pearl, 2009; Bareinboim et al., 2022; Bareinboim, 2025). What we observe instead is data from many skeletons, each with its own combination of objects

---

[2]This is weaker than requiring the template RSCM itself to be acyclic. For instance, an RSCM may specify that for cars $C$ and $C'$, Car.$B$ affects $C'.B$ if $C'$ is behind $C$. This appears cyclic at the template level; however, in any grounding $\mathcal{M}_\rho$, two cars cannot both be behind each other, and so $\mathcal{M}_\rho$ is acyclic. We implement such an RSCM in Exp. 6.2.

and relations. In this section, we consider what can be learned from such data about the true RSCM, and what this implies for unseen relational structures.

A ground RSCM $\mathcal{M}_\rho$ induces observational, interventional, and counterfactual distributions over $\mathbf{V}_\rho$ (Def. B.5).

*Example* 5 (RSCM distributions for traffic scene). Consider $\mathcal{M}_\rho$ in Ex. 4. The observational query $P(s_1.W = 1)$ is the probability that signal $s_1$ says 'walk'. The interventional query $P(c_1.B = 1 \mid do(p_1.X = 1))$ is the probability that car $c_1$ brakes when pedestrian $p_1$ crosses under intervention, irrespective of the signal (e.g., by an officer). The counterfactual $P\big(c_1.B_{s_1.W=1} = 1 \,\big|\, s_1.W = 0, \ p_1.X = 1, \ c_1.B = 0\big)$ asks: in a scene where the signal was 'drive', $p_1$ crossed, and $c_1$ did not brake, what is the probability that $c_1$ *would have* braked had $s_1$ been set to 'walk'? $\square$

The classic challenge in causal inference is inferring causal effects from observational data for a fixed skeleton. Generalizing this, say we have data from a given set of source skeletons, and we want to answer a query about an unseen target skeleton. We show that even the *observational* distribution for this target is not identifiable.

**Theorem 3.3** (Impossibility of observational inference across skeletons). *Consider a schema $\mathcal{S}$, source skeletons $\rho_1, \dots, \rho_l$, and target skeleton $\rho_\star$. Then, for any RSCM $\mathcal{M}$ over $\mathcal{S}$, there exists another RSCM $\mathcal{M}'$ over $\mathcal{S}$ such that $\mathcal{M}$ and $\mathcal{M}'$ agree on observational distributions $P(\mathbf{v}_{\rho_k})$ for every source skeleton $\rho_k$ but disagree on the observational distribution $P(\mathbf{v}_{\rho_\star})$ of the target skeleton.*

*Example* 6 (Impossibility of observational inference across skeletons). Consider the source skeleton $\rho_A$ and target skeleton $\rho_C$ (Fig. 1). Say we know $P(\mathbf{v}_{\rho_A})$ and want to learn $P(\mathbf{v}_{\rho_C})$. We can show this is not possible, following Thm. 3.3 above. Let the true RSCM be $\mathcal{M}$ given in Ex. 3, where the mechanism determining whether a car brakes is

$$\mathsf{Car}.B \leftarrow \mathsf{Car}.U_B \oplus \left( \bigvee_{\mathsf{Ctrl}(\mathsf{Sig},\mathsf{Car})} \mathsf{Sig}.W \vee \bigvee_{\mathsf{Path}(\mathsf{Ped},\mathsf{Car})} \mathsf{Ped}.X \right).$$

Consider another RSCM $\mathcal{M}'$ which is identical to $\mathcal{M}$, except with a slightly different braking mechanism:

$$\mathsf{Car}.B \leftarrow \mathsf{Car}.U_B \oplus \left( \bigoplus_{\mathsf{Ctrl}(\mathsf{Sig},\mathsf{Car})} \mathsf{Sig}.W \vee \bigvee_{\mathsf{Path}(\mathsf{Ped},\mathsf{Car})} \mathsf{Ped}.X \right).$$

In skeleton $\rho_A$, any car is controlled by at most one signal, so we can show that $\mathcal{M}$ and $\mathcal{M}'$ yield identical distributions

for $\rho_A$. In particular $c_1$ is controlled only by $s_1$, and we have $\bigvee\{s_1.W\} = \bigoplus\{s_1.W\}$; $c_2$ is not controlled by any signals, and $\bigvee \emptyset = \bigoplus \emptyset$. However, in the target skeleton $\rho_C$, car $c_1$ is controlled by two signals; here, $\bigvee\{s_1.W, s_2.W\} \neq \bigoplus\{s_1.W, s_2.W\}$ in general. Consider the conditionals:

$$P^{\mathcal{M}_{\rho_C}}(c_1.B = 1 \mid p_1.X = 0, p_2.X = 0,$$
$$s_1.W = 1, s_2.W = 1)$$
$$= P^{\mathcal{M}_{\rho_C}}(c_1.U_B \oplus ((1 \vee 1) \vee (0 \vee 0)) = 1)$$
$$= P^{\mathcal{M}_{\rho_C}}(c_1.U_B = 0) = 0.8$$
$$P^{\mathcal{M}'_{\rho_C}}(c_1.B = 1 \mid p_1.X = 0, p_2.X = 0,$$
$$s_1.W = 1, s_2.W = 1)$$
$$= P^{\mathcal{M}'_{\rho_C}}(c_1.U_B \oplus ((1 \oplus 1) \vee (0 \vee 0)) = 1)$$
$$= P^{\mathcal{M}'_{\rho_C}}(c_1.U_B = 1) = 0.2$$

Thus, the two SCMs disagree on $P(\mathbf{v}_{\rho_C})$. $\qquad\square$

If we want to learn an interventional distribution for an unseen target skeleton $\rho_\star$. Thm. 3.3 already limits our ability to do this. It implies that even the observational distribution $P(\mathbf{v}_{\rho_\star})$, a prerequisite of existing causal inference methods, may not be identified by the source data. An independently interesting question, however, is: when we *do* know some distributions for the target skeleton, e.g., $P(\mathbf{v}_{\rho_\star})$, does this suffice to identify other interventional (or counterfactual) queries in the target? We give a negative answer.[3]

**Theorem 3.4** (Impossibility of causal inference within a skeleton). *Consider a schema $\mathcal{S}$ where at least one entity or relation type has more than one observed attribute. For any relational SCM $\mathcal{M}$ over $\mathcal{S}$ and skeleton $\rho$, there exists another relational SCM $\mathcal{M}'$ over $\mathcal{S}$ such that $\mathcal{M}$ and $\mathcal{M}'$ agree on the observational distribution $P(\mathbf{v}_\rho)$ but disagree on some interventional distribution over $\mathbf{V}_\rho$.*

Thms. 3.3 and 3.4 hold even when the relational structure is known. This suggests the need for assumptions about the causal structure, in addition to the relational structure.

# 4. Relational Identification

Previously, we showed that without further assumptions, even the observational distribution for unseen combinations of objects cannot be identified. In this section, we develop a graphical model approach to overcome this impossibility.

[3]It may seem that a negative answer follows immediately from the causal hierarchy theorem (CHT) (Bareinboim et al., 2022). However, the proof of the CHT relies on being able to construct an *arbitrary* SCM $\mathcal{M}'$ that matches the true SCM $\mathcal{M}$ on the given distribution(s) but not on the query. We are in a stricter setting where $\mathcal{M}'$ must share exogenous distributions and functions across objects of the same type, i.e., be a grounding of an RSCM.

## 4.1. Defining Relational Causal Graphs

First, we extend causal graphs to include relational constraints (Pearl, 2009; Koller & Friedman, 2009).

**Definition 4.1** (Relational causal graph). An RSCM $\mathcal{M} = \langle \mathcal{S}, \mathbf{V}, \mathbf{U}, \mathcal{F}, P(\mathbf{U}) \rangle$ induces a *relational causal graph $\mathcal{G}$* constructed as follows.

- **Non-relational subgraph.** For each object type $O \in \mathcal{E} \cup \mathcal{R}$, let $\mathcal{G}$ contain nodes for each variable $O.A \in \mathbf{V}$, a directed edge $O.B \to O.A$ for any $O.B \in \mathbf{Pa}_{O.A}$, and a dashed bidirected edge $O.A \leftrightarrow O.B$ for any $O.B \in \mathbf{V}$ such that $\mathbf{U}_{O.A} \cap \mathbf{U}_{O.B} \neq \emptyset$, annotated with the constraint $O = O'$.

- **Relational subgraph.** For each variable $O.A \in \mathbf{V}$ and each relational parent $R = (\mathbf{W}, \phi, \text{AGG}) \in \mathbf{Pa}^r_{O.A}$, let $\mathcal{G}$ contain a *relational node $O.R$* and an edge $O.R \to O.A$. For each $T.W \in \mathbf{W}$, add an edge $T.W \to O.R$ annotated with $\phi$ and AGG. Finally, for any $T.B \in \mathbf{V}$ such that $O.A$ and $T.B$ have exogenous relational parents $(\mathbf{W}_1, \phi_1, \text{AGG}_1)$ and $(\mathbf{W}_2, \phi_2, \text{AGG}_2)$ respectively such that some $Z.U \in \mathbf{W}_1 \cap \mathbf{W}_2$, add a dashed bidirected edge $O.A \leftrightarrow T.B$ annotated with the constraint $\exists Z : \phi_1 \wedge \phi_2$, or append the constraint to an existing $O.A \leftrightarrow T.B$ edge.

Fig. 2 shows the graph for the traffic RSCM (Ex. 3). Like an RSCM, a relational causal graph $\mathcal{G}$ can be instantiated for any skeleton to yield a *ground graph $\mathcal{G}_\rho$* (Def. B.6). The relational nodes in $\mathcal{G}_\rho$ can be *marginalized* (Def. B.7, Fig. 1) to yield a standard causal graph $\bar{\mathcal{G}}_\rho$.

Causal graphs can be used to encode assumptions about the space of possible RSCMs. They help circumvent the impossibility results of Sec. 3, enabling *relational identification*.

**Definition 4.2** (Relational counterfactual identification). Consider a schema $\mathcal{S}$, relational causal graph $\mathcal{G}$, source skeletons $\rho_1, \ldots, \rho_l$, source distributions $\mathbb{P} = \{\{P(\mathbf{v}_{\rho_k} \mid do(\mathbf{x}_{k,j}))\}_{j=1}^{m_k}\}_{k=1}^l$, and target skeleton $\rho_\star$. Let $P(\mathbf{y}_\star \mid \mathbf{x}_\star)$ be a target query with $\mathbf{Y}_\star, \mathbf{X}_\star \subseteq \mathbf{V}_{\rho_\star}$.

We say $P(\mathbf{y}_\star \mid \mathbf{x}_\star)$ is *relationally identifiable* from $\mathcal{G}$ and $\mathbb{P}$ if for any RSCMs $\mathcal{M}, \mathcal{M}'$ consistent with $\mathcal{G}$ agreeing on the source data, so that for every $\rho_k$ and $j = 1, \ldots, m_k$,

$$P^{\mathcal{M}_{\rho_k}}(\mathbf{v}_{\rho_k} \mid do(\mathbf{x}_{k,j})) = P^{\mathcal{M}'_{\rho_k}}(\mathbf{v}_{\rho_k} \mid do(\mathbf{x}_{k,j})) > 0,$$

they also agree on the query:

$$P^{\mathcal{M}_{\rho_\star}}(\mathbf{y}_\star \mid \mathbf{x}_\star) = P^{\mathcal{M}'_{\rho_\star}}(\mathbf{y}_\star \mid \mathbf{x}_\star).$$

Otherwise, the query is *relationally non-identifiable*.

A special case of the above definition is when all skeletons (source and target) are *isomorphic* to each other (Def. B.1.

This is the task of same-skeleton identification. When the target skeleton is non-isomorphic to every source skeleton, we refer to the task as cross-skeleton identification.

*Example* 7 (Relational identification across skeletons). Continuing Ex. 1, suppose we are given the graph $\mathcal{G}$ in Fig. 2. Let the target skeleton be $\rho_B$ (Fig. 1) and consider the query $P^{\rho_B}(c_1.B = 1 \mid do(s_1.W = 1))$, the effect of setting signal $s_1$ to 'walk' on whether car $c_1$ brakes. A non-relational identification task for this query would assume all available distributions are from the same skeleton $\rho_B$. On the other hand, relational identification can ask whether the same query is answerable from a distribution for a different skeleton, e.g., $P(\mathbf{v}_{\rho_A})$.

Crucially, the target query is not interchangeable with $P^{\rho_A}(c_1.B = 1 \mid do(s_1.W = 1))$: in $\rho_A$, $s_1$ controls two pedestrians in the path of $c_1$, whereas in $\rho_B$, $s_1$ controls only one pedestrian in the path of $c_1$. So, the contribution of $s_1.W$ need not be the same. The target quantity is also not interchangeable with $P^{\rho_B}(c_2.B = 1 \mid do(s_1.W = 1))$, since $s_1$ has no effect on $c_2$. A non-relational query such as $P(\mathsf{Car}.B = 1 \mid do(\mathsf{Sig}.W = 1))$, the usual target of causal inference, hides this variation. □

### 4.2. Identification Machinery

We now introduce tools for relational identification.

**Observational inference.** We first ask when observational distributions can be transferred across skeletons, addressing the limitation in Thm. 3.3. Say a ground variable $o.A$ is 'unconfounded' if there are no bidirected edges incident to it in the ground graph $\mathcal{G}_\rho$.

**Theorem 4.3** (Observational identification across skeletons). *Consider a schema $\mathcal{S}$, relational causal graph $\mathcal{G}$, source skeleton $\rho$, and target skeleton $\rho_\star$. Let $o.A$ be an unconfounded variable in $\mathbf{V}_{\rho_\star}$. The conditional $P(o.a \mid \mathbf{pa}_{o.A}, \mathbf{pa}^r_{o.A})$ is relationally identifiable from $\mathcal{G}$ and $P(\mathbf{v}_\rho)$ if there exists a source instance $o' \in \rho$ such that $o'.A$ is unconfounded and $\mathrm{dom}(\mathbf{Pa}^r_{o.A}) \subseteq \mathrm{dom}(\mathbf{Pa}^r_{o'.A})$. In this case, $P(o.a \mid \mathbf{pa}_{o.A}, \mathbf{pa}^r_{o.A}) = P(o'.a \mid \mathbf{pa}_{o'.A}, \mathbf{pa}^r_{o'.A})$.*

If $\mathcal{G}$ is Markovian, (1) holds automatically, allowing us to recover the full target observational distribution $P(\mathbf{v}_{\rho_\star})$ if the support condition is satisfied for each instance (Cor. D.4).

*Example* 8 (Observational identification across skeletons). Continuing Ex. 1, let $\rho_A$ and $\rho_C$ be the source and target skeletons respectively, and $\mathcal{G}$ (Fig. 2) the relational causal graph. The conditional $P^{\rho_C}(c_2.b \mid s_2.w, p_2.x, p_3.x)$ in the target is identifiable from $P(\mathbf{v}_{\rho_A})$ and $\mathcal{G}$. This is because $c_1.B$ in $\mathbf{V}_{\rho_A}$ and $c_2.B$ in $\mathbf{V}_{\rho_C}$ are both unconfounded, and affected by exactly two pedestrians and one signal (and therefore have the same parent domains). In particular, writ-

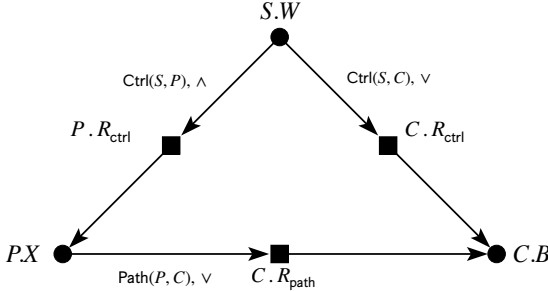

*Figure 2.* A relational causal graph (Def. 4.1) for the traffic RSCM in Ex. 3. $\mathsf{Sig}.W$ denotes the state of a signal, $\mathsf{Ped}.X$ whether a pedestrian crosses, and $\mathsf{Car}.B$ whether a car brakes. A signal affects a pedestrian or car only if it controls them ($\mathsf{Ctrl}$), and a pedestrian affects a car only if they are in the car's path ($\mathsf{Path}$). The relational nodes represent aggregated values ($\mathsf{or}$, $\mathsf{and}$) of the related objects.

ing multisets of values $\bar{w} = \{s_2.w\}$ and $\bar{x} = \{p_2.x, p_3.x\}$,

$$P^{\rho_C}(c_2.b \mid s_2.w, p_2.x, p_3.x)$$
$$= P^{\rho_C}(c_2.b \mid \{s_2.W\} = \bar{w}, \{p_2.X, p_3.X\} = \bar{x})$$
$$= P^{\rho_A}(c_1.b \mid \{s_1.W\} = \bar{w}, \{p_1.X, p_2.X\} = \bar{x}).$$

Consider a more complex query $P^{\rho_C}(c_1.b \mid s_1.w, s_2.w, p_2.x, p_2.x)$ in the target. There is no car in the source controlled by two signals. If the parent domains are multisets of values, no car in the source meets the support condition for this query. However, note the aggregation constraint in $\mathcal{G}$: $\mathsf{Car}.B$ only depends on its controlling signals via the aggregate $\vee$. Then, letting $\bar{w} = \vee(s_1.w, s_2.w)$ and $\bar{x} = \vee(p_2.x, p_3.x)$, we have

$$P^{\rho_C}(c_1.b \mid s_1.w, s_2.w, p_2.x, p_2.x)$$
$$= P^{\rho_C}(c_1.b \mid \vee(s_1.W, s_2.W) = \bar{w}, \vee(p_1.X, p_2.X) = \bar{x})$$
$$= P^{\rho_A}(c_1.b \mid \vee(s_1.W) = \bar{w}, \vee(p_1.X, p_2.X) = \bar{x}),$$

rendering the query identifiable. □

**Causal inference.** For the task of same-skeleton identification, we show how the *ctf-calculus* (Correa & Bareinboim, 2025), a recent generalization of do-calculus (Pearl, 2009), can be used to show identifiability.

**Proposition 4.4** (Sufficient condition for same-skeleton relational identification). *Consider a schema $\mathcal{S}$, relational causal graph $\mathcal{G}$, skeleton $\rho$, and family of interventional distributions $\mathbb{P}$ over $\mathbf{V}_\rho$. If $P(\mathbf{y}_* \mid \mathbf{x}_*)$ is identifiable via ctf-calculus from the marginalized ground graph $\bar{\mathcal{G}}_\rho$ and $\mathbb{P}$, then it is also relationally identifiable from $\mathcal{G}$ and $\mathbb{P}$.*

As a corollary, we recover the well-known backdoor adjustment formula (Pearl, 2009) in the relational setting (Cor. D.2).

*Example* 9 (Same-skeleton relational backdoor.). Continuing Ex. 1, let $\rho = \rho_A$ be the skeleton of interest, $\mathcal{G}$

(Fig. 2) the graph, and $P(c_1.B \mid do(p_1.X))$ the query. In the marginalized ground graph $\bar{\mathcal{G}}_{\rho_A}$ (Fig. 1), $\mathbf{Z} = \{s_1.W, p_2.X\}$ is a valid backdoor adjustment set for the query. This gives

$$P(c_1.b \mid do(p_1.x)) =$$
$$\sum_{p_2.x, s_1.w} P(c_1.b \mid p_1.x, p_2.x, s_1.2) \cdot P(p_2.x, s_1.w)$$

Next, we turn to non-identifiability. Many practical queries are within-instance: they ask how intervening on an individual's treatment affects that same individual's outcome when individuals interact (e.g., showing an online ad to one user in a social network and measuring that user's purchases). We show that if a within-instance query is standardly non-identifiable, then it is also relationally non-identifiable.

**Proposition 4.5** (Necessary condition for within-instance relational identification). *Consider a schema $\mathcal{S}$, relational causal graph $\mathcal{G}$, source skeletons $\rho_1, \ldots, \rho_l$ with available interventional distributions $\mathbb{P}$, and a target skeleton $\rho_\star$. Let $o \in \rho_\star$ be a target instance and consider a counterfactual query $P(\mathbf{y}_\star \mid \mathbf{x}_\star)$ with $\mathbf{Y}_\star, \mathbf{X}_\star \subseteq \mathbf{V}_o$, the attributes of $o$.*

*Let the restriction $\mathbb{P}|_O$ be as follows. For each source skeleton $\rho_k$, each distribution $P(\mathbf{v}_{\rho_k} \mid do(\mathbf{x}_{k,j})) \in \mathbb{P}$, and each object $o' \in \rho_k(O)$, include $P(\mathbf{v}_{\rho_k,o'} \mid do(\mathbf{x}_{k,j} \cap \mathbf{v}_{\rho_k,o'}))$ in $\mathbb{P}|_O$, with instance identifiers omitted. Let $\mathcal{G}_o$ be the induced subgraph of the marginalized ground graph $\bar{\mathcal{G}}_{\rho_\star}$ on $\mathbf{V}_o$ with instance identifiers omitted.*

*If $P(\mathbf{y}_\star \mid \mathbf{x}_\star)$ is non-identifiable via ctf-calculus from $\mathbb{P}|_O$ and $\mathcal{G}_o$, then it is relationally non-identifiable from $\mathcal{G}$ and $\mathbb{P}$.*

See Ex. C.2 for an application of the above result.

This section developed symbolic criteria for relational identification across a range of settings. We next introduce a neural approach that generalizes these criteria and provides a practical route to identification.

## 5. Relational Neural Causal Models

Neural causal models (NCMs) (Xia et al., 2021; 2023) parameterize an SCM with neural networks. We adopt the same idea in the relational setting, developing a neural RSCM constrained by a given graph to enable relational identification from graph and data.

A neural RSCM contains one neural network for each variable $O.A$; this network is reused for all ground $o.A$. The given graph $\mathcal{G}$ constrains which inputs each network may use. In the traffic example, this means one network for $Car.B$ is applied to every car's $c.B$, taking as input that car's non-relational parents (e.g., $c.U_B$) and relational parents (e.g., the counts of $s.W$ for controlling signals and $Ped.X$ for in-path pedestrians).

---

**Algorithm 1** RelationalNeuralID

**Input:** schema $\mathcal{S}$, relational causal graph $\mathcal{G}$, source data $\mathcal{D} = \left\{ \left( \rho_k, \{P(\mathbf{v}_{\rho_k} \mid do(\mathbf{x}_{k,j}))\}_{j=1}^{m_k} \right) \right\}_{k=1}^{l}$, target skeleton $\rho_\star$, query $P(\mathbf{y}_* \mid \mathbf{x}_*)$

$\hat{M} \leftarrow \mathcal{G}\text{-RNCM}$

$\hat{\theta}_l \leftarrow \arg\min_{\theta \in \Theta(\hat{M})} P^{\hat{M}_{\rho_\star}(\theta)}(\mathbf{y}_* \mid \mathbf{x}_*)$ subject to $\forall k, j$

$\quad P^{\hat{M}_{\rho_k}(\theta)}(\mathbf{v}_{\rho_k} \mid do(\mathbf{x}_{k,j})) = P(\mathbf{v}_{\rho_k} \mid do(\mathbf{x}_{k,j}))$

$q_l \leftarrow P^{\hat{M}_{\rho_\star}(\hat{\theta}_l)}(\mathbf{y}_* \mid \mathbf{x}_*)$

$\hat{\theta}_r \leftarrow \arg\max_{\theta \in \Theta(\hat{M})} P^{\hat{M}_{\rho_\star}(\theta)}(\mathbf{y}_* \mid \mathbf{x}_*)$ subject to $\forall k, j$

$\quad P^{\hat{M}_{\rho_k}(\theta)}(\mathbf{v}_{\rho_k} \mid do(\mathbf{x}_{k,j})) = P(\mathbf{v}_{\rho_k} \mid do(\mathbf{x}_{k,j}))$

$q_r \leftarrow P^{\hat{M}_{\rho_\star}(\hat{\theta}_r)}(\mathbf{y}_* \mid \mathbf{x}_*)$

**if** $q_l = q_r$ **then**

$\quad$ **return** $q_l$

**else**

$\quad$ **return** FAIL

**end if**

---

**Assumptions.** We assume throughout this section that observed attributes are discrete and finite, that $\mathcal{G}$ is $\rho$-Markovian (no unobserved confounding between different instances), and that each relational-parent multiset has bounded size (see Sec. D.3).

**Definition 5.1** ($\mathcal{G}$-Constrained Relational Neural Causal Model ($\mathcal{G}$-RNCM)). Consider a schema $\mathcal{S}$ and a relational causal graph $\mathcal{G}$. A $\mathcal{G}$-RNCM $\mathcal{N} = \langle \mathcal{S}, \mathbf{V}, \mathbf{U}, \mathcal{F}, P(\mathbf{U}) \rangle$ is an RSCM constructed as follows.

1. $\mathbf{V}$ contains a variable $O.A$ for every non-relational node $O.A$ in $\mathcal{G}$.

2. For every object type $O \in \mathcal{E} \cup \mathcal{R}$, and every maximal bidirected clique $\mathbf{C}$ in $\mathcal{G}$ over variables belonging to type $O$, $\mathbf{U}$ contains a variable $O.U_{\mathbf{C}} \sim \mathcal{U}([0,1])$.[4]

3. For each $O.A \in \mathbf{V}$, the mechanism $f_{O.A}$ is a feed-forward neural network

$$O.A \leftarrow f_{O.A}(\mathbf{pa}_{O.A}, \mathbf{u}_{O.A}, \mathbf{pa}_{O.A}^r).$$

Above, $\mathbf{Pa}_{O.A}$ consists of variables $O.B \in \mathbf{V}$ with a directed edge $O.B \rightarrow O.A$ in $\mathcal{G}$; $\mathbf{U}_{O.A}$ consists of variables $O.U_{\mathbf{C}}$ such that $O.A$ is in the clique $\mathbf{C}$; and $\mathbf{Pa}_{O.A}^r$ consists of multisets (or aggregates as specified in $\mathcal{G}$) of variables $\mathbf{W}$ such that there is an edge $O.R \rightarrow O.A$ in $\mathcal{G}$ where $O.R$ is a relational node with $R = (\mathbf{W}, \phi, \text{AGG})$.

Note that since $\mathcal{G}$ is $\rho$-Markovian, the set $\mathbf{U}_{O.A}^r$ is empty for each $O.A$ in the definition above. We show that $\mathcal{G}$-RNCMs

---

[4]See Sec. B.2 for the definition of a maximal bidirected clique.

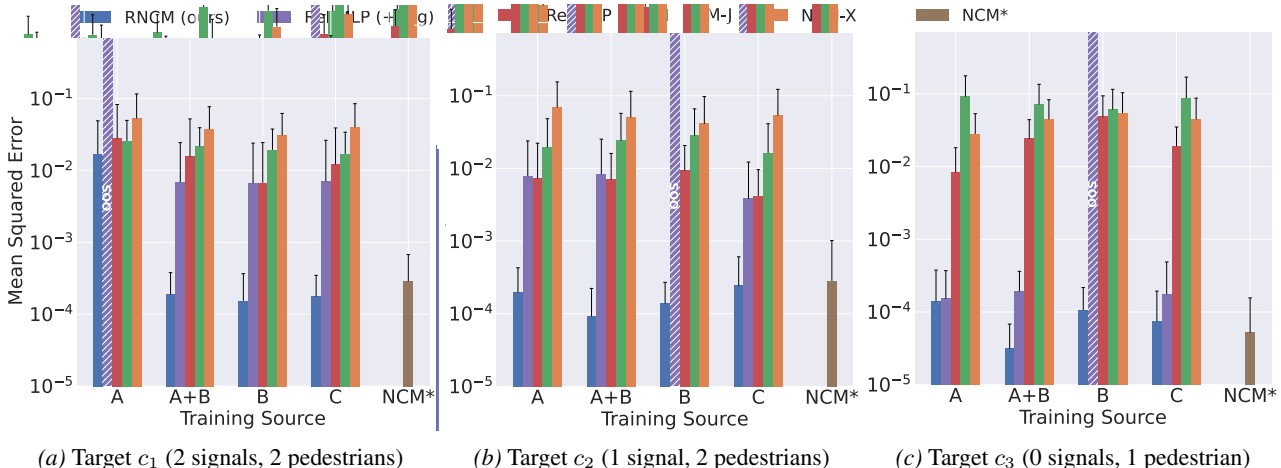

*(a)* Target $c_1$ (2 signals, 2 pedestrians)     *(b)* Target $c_2$ (1 signal, 2 pedestrians)     *(c)* Target $c_3$ (0 signals, 1 pedestrian)

*Figure 3.* Estimation accuracy of RNCMs for causal effects across the traffic scenes in Fig. 1 (Exp. 6.1). The x-axis denotes the source skeleton used for training. The y-axis denotes $\log(\text{MSE})$ to ground truth of various methods in estimating an interventional query in the target skeleton $\rho_C$, averaged over 10 seeds (**lower is better**). Each panel represents a different car for which the query is formulated. RNCMs consistently outperform baselines —even when the baselines are trained directly on the target skeleton while the RNCM is not. In identifiable cases, RNCMs often match the gold-standard NCM* trained directly on the target. Performance degrades only in non-identifiable settings (training on $\rho_A$ and evaluating on $c_1$). 'OOS' denotes out-of-support cases for which the estimate is undefined.

are expressive enough to represent any RSCM consistent with $\mathcal{G}$, under our stated assumptions.

**Theorem 5.2** (Expressivity of RNCMs). *Consider a relational schema $\mathcal{S}$. For every RSCM $\mathcal{M}$ over $\mathcal{S}$ inducing relational causal graph $\mathcal{G}$, there exists a $\mathcal{G}$-RNCM $\mathcal{N}$ such that for every skeleton $\rho$, the ground RSCMs $\mathcal{M}_\rho$ and $\mathcal{N}_\rho$ induce the same counterfactual distributions over $\mathbf{V}_\rho$.*

The expressivity of RNCMs lays the groundwork for causal identification via neural methods. In particular, we can now train the parameters of an RNCM to fit the source data while minimizing or maximizing the query on our target skeleton; this procedure is given in Alg. 1. As a corollary of Thm. 5.2, we can derive that Alg. 1 is sound and complete for the task of relational identification (Cor. D.7).

While Alg. 1 is stated as taking distributions for inputs, in practice, and in our implementation, each distribution is observed only through finitely many samples. The equality constraints are replaced by approximate constraints induced by a maximum-likelihood objective, adapting standard NCM training procedures to allow for parameter sharing across instances (Xia et al., 2021; 2023) (Appendix E).

## 6. Experiments

### 6.1. Estimation accuracy across traffic scenes

We evaluate how well RNCMs can estimate identifiable queries on both seen and unseen skeletons using Alg. 2. We use the traffic schema in Ex. 1 and graph $\mathcal{G}$ in Fig. 2, but

omit the aggregators, creating a more challenging task.[5]

**Setup.** We train on four source settings from Fig. 1: three use a single skeleton for training ($\rho_A$, $\rho_B$ or $\rho_C$) and one uses a pair of skeletons ($\rho_A$ and $\rho_B$). For each source skeleton, we generate $n = 10^4$ observational samples and train five models: (i) a $\mathcal{G}$-RNCM; (ii) NCM-X, a non-relational causal baseline that trains a standard NCM on data consisting of a Cartesian product of all $(s.W, p.X, c.B)$ triples; (iii) NCM-J, a causal baseline with partial relational information, similar to NCM-X but restricting to 'joined' triples where $s, p, c$ are pairwise related; (iv) REL-MLP, a relational non-causal baseline that predicts $c.B$ from the number and states of related cars and pedestrians; and (v) REL-MLP + DEG, a variant of REL-MLP that predicts $c.B$ only from data on cars with the same degree as $c$ (see Appendix E.4 for details). We also train a gold-standard NCM* directly on the target data and ground graph (without parameter sharing). Then, we evaluate each trained model on the target skeleton $\rho_\star = \rho_C$, estimating the query, for various cars $c$: what is the probability that $c$ will brake given that all the pedestrians in its path are set to 'cross'? For the three cars appearing in $\rho_C$, this results in three queries total (Table E.4.1); each query requires backdoor adjustment on $s.W$ for signals $s$ that both control $c$ and control pedestrians $p$ in the path of $c$.

**Comparison with baselines.** RNCMs consistently outperform flat NCMs and REL-MLP variants, often by $\approx 100\text{x}$

---

[5]We use a histogram of counts of the different values in the domain of the relational parents. For discrete variables, this is a sufficient statistic for the multiset of relational parent values.

in identifiable source-target settings (Fig. 3). Notably, even when the RNCM is trained on a source skeleton distinct from the target, it outperforms baselines trained directly on the target. For instance, an RNCM trained on source skeleton $\rho_A$ outperforms NCM-X and NCM-J trained on $\rho_C$ when estimating the query for car $c_2$ in $\rho_C$ (Fig. 3b). This highlights the importance of relational structure in determining causal effects. A notable strength of REL-MLP + DEG is on car $c_3$ (Fig. 3c),. For car $c_3$, it happens to be that, due to the absence of confounding, $P(c_3.b \mid do(p_3.x)) = P(c_3.b \mid p_3.x)$, explaining the success of this non-causal method. Finally, RNCMs trained on sources distinct from the target are typically strong enough to match or even exceed the gold-standard NCM$^*$ trained directly on the target. The main exception is training on $\rho_A$ and evaluating on $c_1$ (Fig. 3a), a non-identifiable case which we discuss next.

**Role of training source.** Our source-target combinations cover three cases: generalization to exactly matched neighbourhoods (e.g., source $\rho_A$ to car $c_2$ in $\rho_C$); generalization to smaller neighbourhoods (e.g., source $\rho_B$ to car $c_2$ in $\rho_C$); and generalization to larger neighbourhoods (e.g., source $\rho_A$ to car $c_1$ in $\rho_C$). We find that RNCMs succeed at the first two, while failing at the third, as predicted by the identifiability theory of Thm. 4.3 and Prop. 4.4. For instance, training on $\rho_A$ underperforms on $c_1$ (Fig. 3a): in $\rho_A$, every car is controlled by at most one signal, so it fails the support condition (Thm. 4.3) for $c_1$ (two signals). Still, combining $\rho_A$ and $\rho_B$ in training recovers performance, illustrating how RNCMs integrate sources to improve accuracy.

### 6.2. Identification accuracy across traffic scenes

We evaluate how well RNCMs are able to decide when a causal effect is identifiable, following Alg. 1.

**Setup.** We use a schema with one entity type Car ($C$) and one relation Behind(Car$_1$, Car$_2$). Each car has two observed attributes Car.$X$ and Car.$Y$. The source skeleton $\rho$ has two cars $c_1, c_2$ with Behind($c_1, c_2$). The target skeleton $\rho_\star$ has three cars $c_1, c_2, c_3$ with Behind($c_1, c_2$), Behind($c_2, c_3$), and Behind($c_1, c_3$). We consider two causal graphs, the relational bow $\mathcal{G}_{\text{bow}}$ and IV $\mathcal{G}_{\text{iv}}$ (Figs. 4, E.6.1). Data-generation follows Exp. 6.1 but with majority aggregation. For each graph, we consider two target queries on $\rho_\star$," $Q_d := P^{\rho_\star}(c_3.Y \mid do(c_2.X))$, an identifiable effect across cars; and $Q_s := P^{\rho_\star}(c_3.Y \mid do(c_3.X))$, a non-identifiable effect within the same car. Results of Alg. 1 are in Fig. 4.

**Results.** Though the target car $c_3$ in $\rho_\star$ has a larger relational neighborhood (two cars behind it) than any car in the source $\rho$, $\mathcal{G}$-RNCMs correctly assess identifiability in the target, matching the theory. In the non-relational IV and bow graphs, the causal effect $P(y \mid do(x))$ is non-identifiable from purely observational data (Pearl, 2009). By Prop. 4.5,

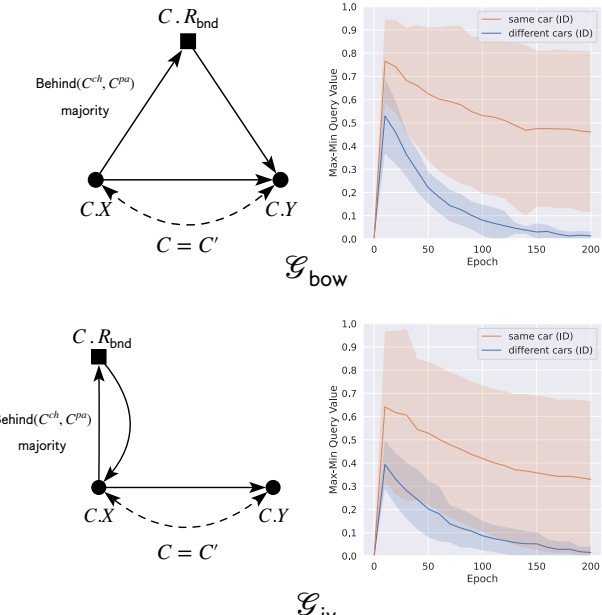

*Figure 4.* Identification accuracy of RNCMs on unseen traffic scenes (Exp. 6.2). **(left)** Relational causal graphs $\mathcal{G}_{\text{bow}}$ (top) and $\mathcal{G}_{\text{iv}}$ (bottom). Within each car $C$, Car.$X$ affects and is confounded with Car.$Y$ in both graphs. **(right)** The average max–min gap produced by RelationalNeuralID (Alg. 1) collapses toward zero for the identifiable different-car query (blue) but remains large for the non-identifiable same-car query (orange). Shaded regions denote the 25th and 75th percentile across 10 random seeds.

the same conclusion carries over to the within-car query $Q_s$. Consistent with this, RNCMs trained to minimize vs. maximize $Q_s$ remain far apart (orange curve in Fig. 4). In contrast, for both graphs we show that the cross-car query $Q_d$ on $\rho_\star$ *is* identifiable from source data $P(\mathbf{v}_\rho)$ (Prop. E.4). Accordingly, the RNCM max–min gap for $Q_d$ collapses to (approximately) zero (solid blue curve in Fig. 4), indicating that Alg. 1 correctly certifies identifiability in this setting.

In Sec. E, we evaluate RNCMs on larger relational structures, model misspecification, and front-door estimation.

## 7. Conclusions

In this paper, we introduced *relational structural causal models* (RSCMs), a generalization of SCMs to object-relational domains. We characterized the limits of learning in this setting (Thm. 3.3, Thm. 3.4), and gave graphical conditions for identification within and across relational skeletons (Thm. 4.3, Cor. D.4, Props. 4.4, 4.5). Finally, we developed *relational neural causal models* for identification that are provably correct (Alg. 1, Alg. 2) Thm. 5.2, Cor. D.7) and empirically outperform existing neural-causal baselines (Sec. 6). We hope our work informs causal reasoning in domains where relations between objects are of scientific importance.

## Acknowledgements

This research was supported in part by the NSF, ONR, AFOSR, DoE, Amazon, JP Morgan, and The Alfred P. Sloan Foundation. We thank Kevin Xia and Yushu Pan for their helpful comments, as well as the anonymous reviewers for their thoughtful feedback.

## Impact Statement

This paper presents results that advance the field of machine learning, bringing together the areas of causal inference and relational learning. In particular, we contribute methodological foundations for causal inference in object-relational settings and validate our methods on simulated traffic scenes. If adopted in practice, our work could enable better generalization in settings such as autonomous driving and robotics.

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

# Appendices

## Contents

## A. Background and Related Works

Much of machine learning is 'Euclidean' (Papillon et al., 2025). It operates on data represented as coordinates in a high-dimensional space. Such representations are also described as 'flat', 'variable-based', 'feature-vector', 'attribute-value,' and 'propositional' (Getoor & Taskar, 2007; Koller & Friedman, 2009). Statistical learning guarantees assume that the data are values of independent and identically distributed (i.i.d.) random variables. In this section, we discuss in which settings it is useful to relax this assumption, and present methods for such settings, causal and otherwise.

### A.1. Relational Data

#### A.1.1. WHAT IS RELATIONAL DATA?

There is no single definition of what counts as relational data. Broadly construed, it is data that is not flat. We describe, below, what various literatures mean by the term 'relational.'

*In database theory*, it refers to data organized under the relational model in a relational database (Codd, 1970; Ullman & Widom, 2002). A vast amount of enterprise, government, and academic data is stored in relational databases.

*In statistical relational learning*, 'relational' concerns domains modeled not as a fixed set of random variables, but rather structured spaces consisting of many types of objects (entities) related in different ways (Getoor et al., 2003; Koller & Friedman, 2009). Both entities and relations may have attributes. For example, to model inheritance across family trees, each tree consists of its own set of individuals, each with their own attributes (Koller & Friedman, 2009). These individuals have varying relations: Mother-of, Father-of, Married-to, Sibling-of. Both the number of individuals and their relations vary across family trees. Since individuals are related, inferences about one individual may be made based on observations of another, e.g., a child's risk of a disease based on their parents'.

*In deep learning*, 'relational' refers not only to a type of input data but also to a type of learning problem that involves reasoning about objects in various relations with one another. For example, while images can be encoded in a fixed Euclidean space, the task of answering questions like 'are there any rubber things that have the same size as the yellow metallic cylinder?' for a given image is considered relational (Santoro et al., 2017); intermediate representations of the data for this task might be structured as objects and relations. Relational reasoning involves composing together different objects (entities) via relations following certain rules (Battaglia et al., 2018).

#### A.1.2. HOW IS RELATIONAL DATA REPRESENTED?

We give some standard representations in increasing order of generality.

**Sequences and sets (without explicit relations)**    At the least structured end, an instance is represented as a finite set of entities (objects) with attributes. Their relations are not specified explicitly, covering cases where relations are absent, unobserved, or deliberately abstracted away. A common modeling assumption is that entity indices are arbitrary, so the distribution (and the model) should be invariant to permutations of entities. For such exchangeable sequences, de Finetti's theorem implies they can be viewed as i.i.d. draws conditional on a latent variable (de Finetti, 1931; Orbanz & Roy, 2015). For sets, modern architectures (Zaheer et al., 2017; Lee et al., 2019; Locatello et al., 2020) enforce permutation invariance (or equivariance) directly while constructing representations. Examples include object-centric representations of scenes (a set of detected objects), multi-agent systems (a set of agents with state vectors), and point clouds (a set of points in space). A limitation of set-based representations is they do not take relational structure as an explicit input. However, they may be used to learn the relations between objects, e.g., by attention between all pairs.

**Matrices and tensors (special cases of graphs).**    Matrices arise when there is a single relation between two types of entities, one indexed by rows and the other by columns, e.g., user–item interactions in recommender system. This can be viewed as a *bipartite graph*. The matrix entries are edge attributes (ratings, clicks, links), and missing entries correspond to

absent/unobserved edges. Tensors extend this to multi-way relations (e.g., user × item × context) or to multiple relation types. The gain relative to sets is that some interaction structure is now explicit (who is connected to whom), but the limitation is that the representation presupposes a small number of entity types and a small number of relations that can be aligned to axes; many relational domains do not naturally factor into a single rectangular array.

**Relational databases (tables).** A relational database contains one or more tables ('relations') with attributes as columns and records as rows. Each attribute has a data type, and each record specifies a value of that type. The primary key is an attribute that uniquely identifies each row, whereas the foreign keys are attributes that connect a given row to other rows in other tables. Relational databases can be encoded as heterogenous graphs (Fey et al., 2024), which we explain next.

**Graphs and heterogeneous graphs (explicit interaction structure).** A graph input consists of explicit entities (nodes) and relations (edges), possibly with attributes on both and possibly with multiple node/edge types. This representation is natural for domains such as social networks (people connected by friendship/follow), transportation networks (intersections connected by road segments), molecular graphs (atoms connected by bonds), or citation/knowledge graphs (papers or entities connected by typed relations). It is often used in relational neural methods that compute by passing information along edges (Gilmer et al., 2017; Battaglia et al., 2018). Compared to set representations, graphs commit to a notion of locality (neighbors) and hence constrain which entities can directly influence which others; compared to matrices, graphs permit arbitrary sparsity patterns, multiple relation types, and heterogeneous entities without forcing them into a single array.

**More general non-Euclidean spaces.** Finally, some domains require modeling more than just edges or pairwise relations between objects. Higher-order relations (hyper-edges) and interactions between edges, triangles, and cliques are often useful in domains spanning physical systems, traffic forecasting, drug discovery, and more (Papillon et al., 2025; Papamarkou et al., 2024; Hajij et al., 2023). Structures such as hypergraphs, simplicial complexes, sheaves, and combinatorial complexes are used to represent such data in the active field of topological deep learning.

## A.2. Probabilistic Models for Relational Data

Much of the probabilistic modeling literature in the 20th century operated on flat, i.i.d. data. For instance, Bayesian networks and related graphical models (Pearl, 2009; Koller & Friedman, 2009)captured dependencies among attributes of a fixed-dimensional random vector. An important exception was the tradition of hierarchical modeling—including multi-level and cross-classified models—for non-i.i.d. data (Gelman & Hill, 2006; Orbanz & Roy, 2015). However, these models typically involve two or fewer entity and relation types and simple, homogeneous relational neighborhoods (i.e., sequence- or matrix-structured data).

A new literature emerged in the 1990s, often grouped under *statistical relational learning* (SRL), that shifted from attribute-value representations of data to object-relational ones (Koller & Friedman, 2009; Getoor & Taskar, 2007). The goal of these methods was to enable probabilistic inference in object-relational settings; for e.g., infer a patient's risk of a disease based on that of their family members', or a user's rating for a movie based on that of their friends. To do so, these methods define a template-level model for a particular schema, laying out probabilistic independencies that could be grounded for any relational instances.

**Plate models.** Plate models represent objects as plates, and relations as intersections between these plates (Buntine, 1994; Spiegelhalter, 2002). Their primary use is in modeling domains with repeated measurements, encoding parameter sharing for objects in the same plate. For example, in a recommender system, one may have one plate for users, and another for movies; the intersection of these plates may define a variable 'rating'. This variable exists for every (user, movie) pair. Plate models are widely used in applied Bayesian statistics (Blei, 2012), but do not express dependencies that are contingent on relational constraints, and assume simple, pre-defined relational neighbourhoods.[6] For example, a plate model can express that a user's rating for a movie depends on their preferences and the movie's quality. However, they can not express that this rating additionally depends on the user's friends' ratings for that movie.

**Directed relational models.** More general directed relational graphical models such as probabilistic relational models (PRMs, also known as relational Bayesian networks) (Friedman et al., 1999; Getoor et al., 2003; Koller & Friedman, 2009)

---

[6]While this is true of plate models as typically used in the literature, (Heckerman et al., 2004) define a generalized plate model that can express constraints.

and the directed acyclic probabilistic entity-relationship model (DAPER) (Heckerman et al., 2004; Getoor & Taskar, 2007) address the limitation of plate models by defining dependencies using first-order constraints and allowing variable-sized neighbourhoods. Application areas include citation networks and web hyperlinks (including link prediction and topic classification) (Getoor & Taskar, 2007), medical diagnosis (Xu et al., 2005), and IT security risk analysis (Sommestad et al., 2010). Nevertheless, like Bayesian networks, they specify factorizations of observational distributions but do not provide a causal semantics for interventions and counterfactuals.

**Other approaches.** In our work, we build on the directed graphical model approach for object-relational domains. However, we note there are a number of other prominent approaches to SRL. Undirected models address the limitation of directed models in capturing cyclic or symmetric dependencies (e.g., feedback loops, dynamical systems, dependencies such as "these individuals are likely to enjoy the same movies"). They include relational Markov networks (Taskar et al., 2002), Markov logic networks (Richardson & Domingos, 2006), and conditional random fields (Sutton & McCallum, 2007). Probabilistic inference tends to be more difficult in undirected models than directed models; still, undirected models have proved useful for entity resolution in citation networks (e.g., is author $o$ the same as author $o'$?) (Singla & Domingos, 2006); for classifying webpages (Taskar et al., 2002); and for extracting knowledge from unstructured text (Bunescu & Mooney, 2004). In addition, given that causation is directed and asymmetric, such undirected models do not suffice to represent causality in relational settings (Maier et al., 2010). Besides graphical models, probabilistic logic programs constitute another (and in some cases, equivalent) approach, and have been applied to link prediction in heterogenous biological networks (Kersting & Raedt, 2001; Cussens, 2000; Bach et al., 2017; De Raedt et al., 2007).

### A.3. Relational Deep Learning

Relational deep learning combines and extends the tradition of graphical models with deep neural networks for scalable inference. Unlike SRL, it is not necessarily probabilistic; like SRL, it operates on non-Euclidean data, most commonly graphs and databases. Its goal is to design architectures with 'relational inductive biases' that leverage relational information for prediction and promote *combinatorial generalization*, understood as generalization across combinations of objects (Battaglia et al., 2018).

**Graph neural networks.** Graph neural networks (GNNs) are perhaps the most popular architecture for relational deep learning (Gori et al., 2005; Scarselli et al., 2009a;b; Hamilton et al., 2017; Kipf & Welling, 2017; Veličković et al., 2018; Veličković, 2023). Given a graph with node (entity) features and edge (relation) features, message passing architectures iteratively update each node representation by aggregating messages from its neighbors. GNNs have been applied to molecular property prediction (Duvenaud et al., 2015; Gilmer et al., 2017; Sypetkowski et al., 2024), physical reasoning and interaction networks in vision and control (Raposo et al., 2017; Zambaldi et al., 2018; Hamrick et al., 2018), knowledge graphs (Bordes et al., 2013; Oñoro et al., 2017; Hamaguchi et al., 2017), and spatiotemporal forecasting on transportation networks (Li et al., 2018; Cui et al., 2019; Derrow-Pinion et al., 2021). GNNs also provide a computational interface between SRL and modern representation learning, since many SRL domains can be compiled into heterogeneous graphs. A number of works have integrated undirected SRL methods into GNN architectures (Dai et al., 2016; Gao et al., 2019; Spalević et al., 2020; Qu et al., 2019; Zhang et al., 2020).

**Causality and GNNs.** An important distinction to make is that the graphs used as input to GNNs are not causal graphs. They are relational graphs, with nodes depicting entities and edges depicting relations between then. Causal graphs, on the other hand, are naturally formulated over features (of nodes or edges). As such GNNs, like the previously discussed SRL approaches, lack the architecture and guaranties for predicting the effects of interventions and counterfactuals, as distinct from observations. For instance, changing a node feature (a movie's log line) or adding an edge (recommending a movie to a user) are not modeled as do-interventions. (Cotta et al., 2023) take an important step towards causal prediction using graph embeddings for the task of link prediction. (Zečević et al., 2021) propose GNNs for causal inference; their method, restricted to settings without unobserved confounding, is equivalent to neural causal models (Xia et al., 2021) under this restriction. However, it applies only to i.i.d. attribute-value data, thus leaving open the relational setting.

**Relational deep learning on databases.** A distinct and increasingly active direction studies deep learning directly on relational databases, with the goal of avoiding manual, error-prone feature engineering to flatten relational data (Fey et al., 2024; Robinson et al., 2024). These works model typically model databases as heterogenous graphs to leverage the power of graph neural networks. They fall into roughly two categories: models are trained on and applicable to a fixed schema (Wu

et al., 2025; Chen et al., 2025; Dwivedi et al., 2026) versus 'foundation models' that are trained on and applicable to diverse schemas (Wang et al., 2025a; Ranjan et al., 2025).

### A.4. Combinatorial and Compositional Generalization

The focus of our work echoes the problem of *combinatorial generalization* studied in artificial intelligence. There is no agreed-upon definition of combinatorial generalization. Sometimes, it is used interchangeably with the term *compositional generalization* (e.g., in (Liu et al., 2022)). However, compositional generalization may also involve the composition of functions or tasks (e.g., subtask reinforcement learning (Mendez et al., 2022; Jothimurugan et al., 2023) and compositional instruction following or skill composition in LLMs (Yang et al., 2024; Zhao et al., 2025; Sakai et al., 2025; Zhou et al., 2023)).

**Combining features vs combining objects.**  We distinguish between two types of combinatorial generalization. First, *feature-combination generalization* holds the underlying object-relational structure fixed but varies the combinations of features of these objects. A canonical example is attribute binding: training on blue circles and red squares and testing on blue squares. Second, *object-combination generalization* varies the underlying set of objects and relations itself (often including the number of objects), e.g., in going from 2 stacked blocks to 3 stacked blocks, or from small interaction graphs to larger ones. In this work, we study the latter type of combinatorial generalization.

**Computer vision.**  In computer vision, combinatorial generalization is often studied for visual scenes with varying objects, attributes, and relations (Okawa et al., 2023; Hwang et al., 2023). CLEVR is a paradigmatic dataset for combinatorial generalization in visual-question answering (Johnson et al., 2016). Combinatorial generalization is also a goal of image generation. (Liu et al., 2022; Du et al., 2023) develop an approach that composes pre-trained diffusion models, explicitly enforce compositionality during inference using logical operators, instead of relying on implicit learning. This approach, alongside several other works (Schott et al., 2022; Montero et al., 2021; Liang et al., 2025), challenges a common view that disentangled representation learning is sufficient for combinatorial generalization.

**Decision-making and world models.**  For tasks such as robotic manipulation and autonomous driving, combinatorial generalization is unavoidable since scenes naturally vary in the number of objects and their relations (Cui et al., 2019; Derrow-Pinion et al., 2021; Lin et al., 2022). (Zambaldi et al., 2018) introduce an approach for relational deep reinforcement learning that uses attention over entity representations, showing improved generalization to more complex instances than those seen during training. (Duan et al., 2025) introduce a formal definition of 'out-of-combination' generalization in the decision-making context that assumes a fixed number of objects and requires generalization to regions out of the *support* of the state space training distribution. While their diffusion-based approach shows promising zero-shot generalization for this task, both the task definition and their approach assume that all objects (or 'base elements') are seen during training (Duan et al., 2025, Sec 3.3), thus ruling out varying numbers of objects. (Song et al., 2024) solve a similar problem as (Duan et al., 2025) using an approach that maps unseen states to the closest state seen during training. An active recent literature on object-oriented (as opposed to monolithic) *world models* aims to learn object-centric representations of pixel data (Nakano et al., 2023; Wu et al., 2023; Baek et al., 2025; Ferraro et al., 2023; Veerapaneni et al., 2020; Wang et al., 2025b; Mosbach et al., 2025; Feng et al., 2025; Zhao et al., 2022), for instance, by decomposing the latent space into 'slots' for different objects and sometimes explicitly modeling relations between objects. While these methods are evaluated on out-of-distribution generalization tasks, their performance on unseen combinations of objects remains relatively understudied.

### A.5. Relational Causal Models

The intersection of causal and relational modeling is relatively understudied. In the graphical framework of causality, most works focus on relational causal discovery–the task of learning a relational causal graph from data. Works on relational causal inference–answering causal queries from graph and data–are few in number, and deal with special types of relations, as we describe below.

**Relational causal discovery.**  (Maier et al., 2010) provide the first algorithm for causal discovery over data stored in relational databases. They use the DAPER model to encode conditional independencies in relational data, and extend the PC algorithm (Spirtes et al., 2000) to the relational setting. (Lee & Honavar, 2016) build on this DAPER framework, coining the term 'relational causal model' for the DAPER model, and providing more efficient and informative algorithms for

relational causal discovery; (Ahsan et al., 2022) extend this to include cyclic dependencies. However, the DAPER model is a probabilistic model, offering a compact encoding of conditional independencies in relational data (as Bayesian networks do for flat data). It lacks a causal semantics for interventions and counterfactuals, just as Bayesian networks do. This is precisely what motivated the formulation of structural causal models (SCMs) and causal/counterfactual Bayesian networks (Pearl, 2009; Bareinboim et al., 2022; Bareinboim, 2025; Correa & Bareinboim, 2025). Therefore, these works leave open the grounding of causality in relational domains, and, therefore, the definition and inference of causal queries. Additionally, all of them address only those conditional independence structures that can be represented using graphs with directed edges, leaving out settings with unobserved confounding (often represented via bidirected edges) (Bareinboim et al., 2022; Jeong et al., 2025).

**Causal inference under interference.** Causal inference under interference can be seen as a special case of relational causal inference where all objects and relations are of the same type (Sobel, 2006; Rosenbaum, 2007; Ogburn & VanderWeele, 2014; Bhattacharya et al., 2020; Hudgens & Halloran, 2008; Zhang et al., 2022a; Sherman & Shpitser, 2018). The interference literature assumes a fixed number of 'units' (or objects) and a fixed interaction structure between them: this captures settings such as people in a given neighborhood, students in a given school, or patients in a given hospital. The query of interest is typically an aggregated causal effect across all units (e.g., an average direct effect, an average spillover effect, or a global average treatment affect). Unlike DAPER, however, these models do not necessarily enforce parameter-sharing across units, e.g., (Zhang et al., 2022a) study interference for linear models without enforcing mechanism sharing across instances of the same type, and allowing different coefficients for different neighbors (violating permutation-invariance). Additionally, works in graphical causality under interference assume the absence of unobserved confounding, an assumption violated in many real-world settings.

**Relational causal inference.** Relational causal inference generalizes inference under interference, allowing heterogeneous objects and relations (Arbour et al., 2016; Jensen et al., 2020; Salimi et al., 2020; Weinstein & Blei, 2024). (Jensen et al., 2020), (Guo et al., 2024), and (Weinstein & Blei, 2024) provide sound methods for causal inference in plate models, showing how 'object-conditioning' can lead to greater identifiability even in the presence of unobserved confounding. However, these models capture only a subset of the restricted types of relations (not including first-order constraints) expressible in plate models. (Arbour et al., 2016) are the first to generalize causal inference to the entity-relationship model, giving a backdoor criterion for identifying interventional queries from abstract ground graphs (Maier et al., 2010). They assume a ground relational network for a particular relational skeleton as input. As such, they do not define a template-level causal model which can be instantiated on different skeletons, tying them together. (Salimi et al., 2020) define a template-level formalism with 'causal rules' that define first-order constraints for when one variable affects another, and give a similar backdoor adjustment criterion. However, these causal rules do not specify the *mechanism* by which this effect unfolds, and thus resemble the coarse-grained information encoded in a causal graph.

Firstly, neither (Arbour et al., 2016) nor (Salimi et al., 2020) provide a mechanism-level definition of relational causal models, defining interventions only using the truncated Markov factorization (Pearl, 2009). As such, their set-up is limited to modeling interventions in settings without unobserved confounding, and does not provide semantics or identification criteria for counterfactuals. Secondly, both works assume that for causal identification in a given skeleton, observational data from that skeleton is available; they do not address the problem of cross-skeleton inference. Finally, while (Arbour et al., 2016) allows the effect of one variable on another to depend on the exact relation satisfied, (Salimi et al., 2020) does not; for example, if both friends' and family members' vaccination statuses affect a given individual's infection risk, they are assumed to affect it in the same way. While possibly beneficial for estimation in practice, this assumption is quite restrictive in real-world settings.

## B. Further Definitions and Summary of Concepts

**Graphical terminology.** Consider a graph $\mathcal{G} = (\mathbf{V}_\mathcal{G}, \mathbf{E}_\mathcal{G})$ with directed and bidirected edges. For a node $O$ in $\mathcal{G}$, the *graphical parents* of $O$ are the set of nodes $\{Y : Y \to X \in \mathbf{E}_\mathcal{G}\}$. The descendants of $O$ are nodes $Y$ such that there is a directed path $X \rightsquigarrow Y$ in $\mathcal{G}$. For a set of nodes $\mathbf{X} \subseteq \mathbf{V}$, the *induced subgraph* $\mathcal{G}_\mathbf{X}$ is the graph containing nodes $\mathbf{X}$ with an edge $V, W$ between $V, W \in \mathbf{X}$ if and only if this edge is present in $\mathcal{G}$. A *bidirected clique* in $\mathcal{G}$ is a set of nodes every pair of which is connected by a bidirected edge in $\mathcal{G}$. Such a clique is *maximal* if there is no larger bidirected clique in which it is strictly contained.

## B.1. Relational Structural Causal Models

We use a modified definition of relational constraints from (Heckerman et al., 2004, Def. 6).

**Definition B.1** (Relational constraint). Consider a relational schema $\mathcal{S} = \langle \mathcal{E}, \mathcal{R}, \mathcal{A} \rangle$. Let $\mathcal{X} \subseteq \mathcal{E}$ be a set of entity types. A relational constraint $\phi(\mathcal{X})$ is a first-order expression whose atoms are relationship symbols in $\mathcal{R}$ (alongside pre-defined constraints such as equality) and whose only free variables range over the space of ground entities of the types in $\mathcal{X}$.

RSCMs are defined using permutation-invariant functions on multiset inputs (Zaheer et al., 2017). Often, in relational learning, such mutlisets are summarized using *aggregators* (Koller & Friedman, 2009; Getoor & Taskar, 2007). We define such aggregators below.

**Definition B.2** (Aggregator). Let $\mathcal{X}$ and $\mathcal{Y}$ be finite sets, and let $\mathcal{X}^{\mathsf{multiset}}$ be the set of all finite multisets over $\mathcal{X}$. An *aggregator* is a function

$$\mathrm{AGG} : \mathcal{X}^{\mathsf{multiset}} \to \mathcal{Y}$$

that maps a finite multiset of elements from $\mathcal{X}$ to an element of $\mathcal{Y}$.

Note that since aggregators take (multi-)sets as input, they do not depend on any ordering over the elements of their input.

We also define what it means for two relational skeletons to be considered the 'same'.

**Definition B.1** (Skeleton isomorphism). An isomorphism between skeletons $\rho$ and $\rho'$ over a given schema $\mathcal{S}$ is a bijection $\pi : \rho \to \rho'$ on entities and relations that (a) preserves types and (b) preserves relations between entities. We write $\rho \cong \rho'$ if such a $\pi$ exists.

In Prop. D.1, we show how RSCM-induced distributions are invariant to skeleton isomorphism.

Next, we define what it means for an RSCM to be acyclic, or recursive.

**Definition B.3** (Recursive RSCM). An RSCM $\mathcal{M} = \langle \mathcal{S}, \mathbf{V}, \mathbf{U}, \mathcal{F}, P(\mathbf{U}) \rangle$ is said to be *recursive* (or *acyclic*) if its relational causal graph $\mathcal{G}$ contains no directed cycles. Equivalently, there exists a topological ordering of the variables $\mathbf{V}$ such that for every attribute $O.A \in \mathbf{V}$, the structural function $f_{O.A}$ only takes as input (relational or non-relational) variables that precede $O.A$ in this ordering.

For a given skeleton $\rho$, an RSCM $\mathcal{M}$ induces a *ground RSCM*, a standard SCM with functions and exogenous distributions shared across variables of the same type.

**Definition B.4** (Ground relational structural causal model). Given an RSCM $\mathcal{M} = \langle \mathcal{S}, \mathbf{V}, \mathbf{U}, \mathcal{F}, P(\mathbf{U}) \rangle$ and a relational skeleton $\rho$ following the schema $\mathcal{S}$, a ground relational structural causal model of $\mathcal{M}$ for $\rho$ is an SCM $\mathcal{M}_\rho = \langle \mathbf{V}_\rho, \mathbf{U}_\rho, \mathcal{F}_\rho, P(\mathbf{U}_\rho) \rangle$ with

1. ground endogenous variables $\mathbf{V}_\rho = \{o.A \mid O.A \in \mathbf{V}, o \in \rho(O)\}$,

2. ground exogenous variables $\mathbf{U}_\rho = \{o.U \mid O.U \in \mathbf{U}, o \in \rho(O)\}$ with distributions $o.U \sim P(O.U)$ given by $P(\mathbf{U})$, and

3. ground mechanisms $\mathcal{F}_\rho$, with $f_{o.A}$ obtained by substituting relational parents $(\mathbf{W}, \phi, \mathrm{AGG})$ in $f_{O.A}$ with (the specified aggregate of) the set of ground variables $\{t.W \mid T.W \in \mathbf{W}, t \in \rho(T), \phi(o, t) \text{ holds in } \rho\}$. This yields a function $f_{o.A}(\mathbf{pa}_{o.A}, \mathbf{u}_{o.A}, \mathbf{pa}_{o.A}^r, \mathbf{u}_{o.A}^r)$ with $\mathbf{Pa}_{O.A} \subseteq \mathbf{V}_\rho$ and $\mathbf{U}_{O.A} \subseteq \mathbf{U}_\rho$.

Note that a ground RSCM can be viewed as a standard SCM with typed variables and function/noise sharing across variables of the same type. The standard mechanism for each relational mechanism $f_{o.A}(\mathbf{pa}_{o.A}, \mathbf{u}_{o.A}, \mathbf{pa}_{o.A}^r, \mathbf{u}_{o.A}^r)$ is the same, but with the argument signature $f_{o.A}(\mathbf{pa}_{o.A} \cup \mathbf{pa}_{o.A}^r, \mathbf{u}_{o.A} \cup \mathbf{u}_{o.A}^r)$. A ground RSCM $\mathcal{M}_\rho$ is said to be recursive if it is recursive viewed as a standard RSCM (Pearl, 2009).

Next, we define the various observational, interventional, and counterfactual distributions that a ground RSCM induces. Note the similarity to standard SCMs (Pearl, 2009; Bareinboim, 2025), since ground RSCMs are simply standard SCMs.

**Definition B.5** (RSCM-induced distributions). Fix an RSCM $\mathcal{M} = \langle \mathcal{S}, \mathbf{V}, \mathbf{U}, \mathcal{F}, P(\mathbf{U}) \rangle$, a relational skeleton $\rho$ following the schema $\mathcal{S}$, and a corresponding ground RSCM $\mathcal{M}_\rho = \langle \mathbf{V}_\rho, \mathbf{U}_\rho, \mathcal{F}_\rho, P(\mathbf{U}_\rho) \rangle$. For any counterfactual events $\mathbf{Y}_\mathbf{x}, \ldots, \mathbf{Z}_\mathbf{w}$

over $\mathbf{V}_\rho$, $\mathcal{M}_\rho$ induces the distribution

$$P^{\mathcal{M}_\rho}(\mathbf{y_x}, \dots, \mathbf{z_w})$$
$$:= \sum_{\mathbf{u}_\rho} \mathbf{1}[\mathbf{Y_x}(\mathbf{u}_\rho) = \mathbf{y}, \dots, \mathbf{Z_w}(\mathbf{u}_\rho) = \mathbf{z}] P(\mathbf{u}_\rho)$$

where the value $\mathbf{Y_x}(\mathbf{u})$ is obtained by the standard SCM semantics over $\mathcal{F}_\rho$.

A distribution over ground variables $\mathbf{V}_\rho$ is said to be *observational*, *interventional*, or *counterfactual* depending on the form of the counterfactual event(s) it assigns probability to.

1. If none of the events involve interventions, i.e., each event is of the form $\mathbf{Y}$ (equivalently $\mathbf{Y}_\emptyset$), then $P^{\mathcal{M}_\rho}$ is *observational*. In this case we write $P^{\mathcal{M}_\rho}(\mathbf{y})$ for $\mathbf{Y} \subseteq \mathbf{V}_\rho$ or $P^{\mathcal{M}_\rho}(\mathbf{v}_\rho)$.

2. If all events share the same intervention and contain no nested interventions, i.e., all events are of the form $\mathbf{Y_x}$ for a single intervention $\mathbf{X}{=}\mathbf{x}$, then $P^{\mathcal{M}_\rho}$ is *interventional*. In this case we write $P^{\mathcal{M}_\rho}(\mathbf{y} \mid do(\mathbf{x}))$ or $P^{\mathcal{M}_\rho}(\mathbf{y_x})$, and similarly $P^{\mathcal{M}_\rho}(\mathbf{v}_\rho \mid do(\mathbf{x}))$.

3. If the distribution involves at least two events $\mathbf{Y_x}, \mathbf{Z_w}$ such that $\mathbf{X}$ and $\mathbf{W}$ are distinct variables or $\mathbf{x}$ and $\mathbf{w}$ are distinct values, then it is *counterfactual*. In this case we write $P^{\mathcal{M}_\rho}(\mathbf{y_x}, \dots, \mathbf{z_w})$.

## B.2. Relational Identification

**Definition B.6** (Ground relational causal graph). Consider a relational causal graph $\mathcal{G}$ over schema $\mathcal{S}$ and a skeleton $\rho$. The *ground relational causal graph* $\mathcal{G}_\rho$ is constructed as follows.

- For each node $O.V$ in $\mathcal{G}$ (relational or otherwise), and each instance $o \in \rho(O)$, include a node $o.V$ in $\mathcal{G}_\rho$.

- For each edge $O.B \to O.A$ in $\mathcal{G}$ where $O.A$ is a non-relational node, include an edge $o.B \to o.A$ in $\mathcal{G}_\rho$.

- For each edge $O.B \leftrightarrow T.B$ in $\mathcal{G}$ annotated with constraint $\phi$, include an edge $o.A \leftrightarrow t.B$ in $\mathcal{G}_\rho$ for each $o \in \rho(O), t \in \rho(T)$ such that $\phi(o, t)$ holds.

- For each edge $T.W \to O.R$ in $\mathcal{G}$ where $O.R$ is a relational node with $R = (\mathbf{W}, \phi, \mathrm{AGG})$, include edges $t.W \to o.R$ for every $y \in \rho(T)$ such that $\phi(o, t)$ holds.

**Definition B.7** (Marginalized ground relational causal graph). Fix $\mathcal{G}$ and $\rho$, and let $\mathcal{G}_\rho$ be the ground relational causal graph (Def. B.6). The *marginalized ground relational causal graph* $\bar{\mathcal{G}}_\rho$ is the graph on node set $\mathbf{V}_\rho$ obtained by marginalizing out all ground relational role nodes. Equivalently, start from $\mathcal{G}_\rho$, delete every relational node $o.R$, and for each deleted node add directed edges $t.W \to o.A$ whenever $\mathcal{G}_\rho$ contained $t.W \to o.R \to o.A$. Bidirected edges among nodes in $\mathbf{V}_\rho$ are inherited unchanged.

## B.3. Summarizing RSCMs versus SCMs

To help relate our terminology to standard Pearl-style SCMs, Table B.3.1 compares the main objects in the standard and relational settings. Informally, the schema provides the types of objects and relations common to various domains, the skeleton provides the concrete objects and relations in one domain, the RSCM specifies reusable template-level mechanisms, and grounding an RSCM on a skeleton produces an ordinary SCM over the instantiated object attributes.

# Relational Structural Causal Models

| Concept | Standard setting | Relational setting | Traffic example (Ex. 1) |
|---|---|---|---|
| Conceptual role | Models causal relationships among a fixed set of variables. | Models causal relationships among attributes of objects whose number and relations may vary across domains. | The traffic RSCM expresses reusable rules such as: cars brake for relevant signals and pedestrians in their path, regardless of how many cars, pedestrians, or signals are present in a particular scene. |
| Vocabulary | Variables fixed in advance, e.g. $W$ (signal state), $X$ (crossing state), $B$ (braking state). | Object, relation, and attribute types fixed in advance in a *relational schema* $\mathcal{S}$. | Object types: $\mathsf{Sig}, \mathsf{Ped}, \mathsf{Car}$. Relation types: $\mathsf{Ctrl}(\mathsf{Sig}, \mathsf{Ped})$, $\mathsf{Ctrl}(\mathsf{Sig}, \mathsf{Car})$, $\mathsf{Path}(\mathsf{Ped}, \mathsf{Car})$. Attributes: $\mathsf{Sig}.W, \mathsf{Ped}.X, \mathsf{Car}.B$. |
| Domain | A set of $n$ 'units', assumed to be i.i.d., following a fixed distribution: joined triplets of signals, pedestrians, and cars $(W_i, X_i, B_i)_{i=1}^n \sim P(W, X, B)$ | A *relational skeleton* $\rho$ specifies the actual object instances and relation instances in one domain. | A particular traffic scene with concrete signals, pedestrians, and cars, e.g. $s_1, p_1, p_2, c_1, c_2$, together with relations such as $\mathsf{Ctrl}(s_1, p_1)$ and $\mathsf{Path}(p_1, c_1)$. |
| Template-level causal model | An SCM specifies variables, exogenous noise, structural functions (one for each variable), and a noise distribution. Functions are contained within a unit; e.g., for a given triple $(W_i, X_i, B_i)$, we have $X_i \leftarrow f_X(W_i)$. | An RSCM specifies variables, exogenous noise, structural functions, and a noise distribution at the level of object and relation types. Mechanisms and distributions can be reused across different instances of these types. | A generic car's braking decision depends on signals controlling the car and pedestrians in its path. $$C.B \leftarrow f_B(S.W \text{ for every } S \text{ s.t. } \mathsf{Ctrl}(S, C), \\ P.X \text{ for every } P \text{ s.t. } \mathsf{Path}(P, C))$$ |
| Ground-level causal model | Usually not emphasized separately. For $n$ i.i.d. units, one can view the model as $n$ independent copies of the same SCM. | Given an RSCM $M$ and a skeleton $\rho$, the *ground RSCM* $M_\rho$ is an ordinary SCM over all instantiated attributes of all objects in $\rho$. | The ground variables include $s_1.W, p_1.X, p_2.X, c_1.B, c_2.B$. The mechanism for $c_1.B$ is obtained by plugging in the actual signals controlling $c_1$ and pedestrians in $c_1$'s path. $$c_1.B \leftarrow f_{C.B}(s_1.W, p_1.X, p_2.X, c_1.U_B)$$ |
| Template-level graph | A graph over fixed variables, without any relationally constrained edges: $W \to X$, $W \to B$, $X \to B$. | A *relational causal graph* (RCG) over attribute types and relational nodes. It summarizes template-level dependencies, including dependencies through relations. | A path $C.B \overset{\mathsf{Ctrl}(S,C)}{\to} C.R_{\mathsf{Ctrl}} \to C.B$ denotes that a car's crossing status is affected by the status of all signals controlling it, in any traffic scene. |
| Ground-level graph | For repeated units, the graph contains one copy of the causal graph per unit, with no cross-unit edges under i.i.d. assumptions. | Given a skeleton $\rho$, the RCG induces a *ground graph* over instantiated variables. This graph may contain cross-object edges whenever objects are related in $\rho$. | If $\mathsf{Ctrl}(s_1, c_1)$, then the ground graph contains a path $$s_1.W \to c_1.R_{\mathsf{Ctrl}} \to c_1.B.$$ After marginalizing the relational node, this corresponds to a direct ground-level dependence $s_1.W \to c_1.B$. |
| Queries | From a given set of distributions (e.g., observational, $do(W = w)$) and graph, identify another distribution (e.g., $do(X = x)$) for the *same* variables. | From a given set of distributions over skeletons $\rho_1, \ldots, \rho_k$ and graph, identify an unseen distribution for a seen or unseen skeleton. | For example, another traffic scene scene could have $s_1, s_2, p_1, p_2, c_1, c_2$ with a different set of $\mathsf{Ctrl}$ and $\mathsf{Path}$ relations. We can query the distribution of $do(s_2.W = w)$ in this scene. |

*Table B.3.1.* Side-by-side comparison of standard SCM concepts and relational SCM concepts, illustrated using the traffic example.

# C. Further Examples and Comparison with Standard SCMs

## C.1. RSCMs versus SCMs: Student Example

A standard SCM may be viewed as a special case of a relational SCM with a single entity type and no relation types. The purpose of this example is to make this comparison explicit, and to show how assuming i.i.d.-ness can lead to incorrect causal effect estimates when the relational effects exist.

Consider measuring the effect of student tutoring on GPA. For each student, let $T \in \{0, 1\}$ indicate tutoring enrollment, where $T = 1$ means enrolled, and let $G \in \{0, 1\}$ indicate whether the student's GPA exceeds 3.5, where $G = 1$ means GPA $> 3.5$.

RELATIONAL SCHEMA

- **Standard SCM.** The units are students. Equivalently, the implicit relational schema is $\mathcal{S}_{\mathrm{std}} = \langle \mathcal{E}, \mathcal{R}, \mathcal{A} \rangle$,      $\mathcal{E} = \{\mathsf{Student}\}$,    $\mathcal{R} = \emptyset$, with observed attributes $\mathcal{A} = \{\mathsf{Student}.T, \mathsf{Student}.G\}$.

- **Relational SCM.** The entity type is again students, but now students may be related by friendship: $\mathcal{S}_{\mathrm{rel}} = \langle \mathcal{E}, \mathcal{R}, \mathcal{A} \rangle$,      $\mathcal{E} = \{\mathsf{Student}\}$,    $\mathcal{R} = \{\mathsf{Friend}(\mathsf{Student}, \mathsf{Student})\}$, with the same observed attributes $\mathcal{A} = \{\mathsf{Student}.T, \mathsf{Student}.G\}$.

RELATIONAL SKELETONS

Let the observed skeleton contain 100 students $s_1, \ldots, s_{100}$. We pair the students into friendship pairs so that, symmetrically,

$$\mathsf{Friend}(s_{2i-1}, s_{2i}) \quad \text{and} \quad \mathsf{Friend}(s_{2i}, s_{2i-1}) \qquad \text{for } i = 1, \ldots, 50.$$

Each student has exactly one friend.

- **Standard SCM.** The implicit skeleton consists of 100 student instances and no relations between them.

- **Relational SCM.** The skeleton consists of 100 student instances and the friendship relations above.

STRUCTURAL CAUSAL MODEL: TEMPLATE LEVEL

- **Standard SCM.** The standard SCM is $M_{\mathrm{std}} = \langle \mathbf{V}, \mathbf{U}, \mathcal{F}, P(\mathbf{U}) \rangle$, where $\mathbf{V} = \{T, G\}$ and $\mathbf{U} = \{U_T, U_G\}$. The mechanisms are

$$T \leftarrow f_T(U_T) = U_T, \qquad G \leftarrow f_G(T, U_G) = T \oplus U_G,$$

  where $\oplus$ denotes XOR. Thus $T$ affects $G$ for the same student only. The exogenous variables are independent, with $U_T \sim \mathrm{Bernoulli}(0.2)$, $U_G \sim \mathrm{Bernoulli}(0.3)$.

- **Relational SCM.** The relational SCM is $M_{\mathrm{rel}} = \langle \mathcal{S}_{\mathrm{rel}}, \mathbf{V}, \mathbf{U}, \mathcal{F}, P(\mathbf{U}) \rangle$, where $\mathbf{V} = \{\mathsf{Student}.T, \mathsf{Student}.G\}$, and $\mathbf{U} = \{\mathsf{Student}.U_T, \mathsf{Student}.U_G\}$. For a student $S$, the tutoring mechanism is

$$S.T \leftarrow f_{S.T}(S.U_T) = S.U_T.$$

Here, $S.T$ has one exogenous non-relational parent–$S.U_T$ for the same student–and no non-relational endogenous parents or relational endogenous or exogenous parents.

The GPA mechanism has one non-relational endogenous parent, the student's own tutoring status $S.T$, and one relational endogenous parent, the tutoring status of $S$'s friends: $\mathbf{Pa}_{S.G}^r = \{(\{\mathsf{Student}.T\}, \mathsf{Friend}(S, S'), \vee)\}$. The corresponding mechanism is

$$S.G \leftarrow f_{S.G}\left( S.T, S.U_G, \bigvee_{S' : \mathsf{Friend}(S, S')} S'.T \right),$$

with

$$S.G \leftarrow \left( S.T \wedge \bigvee_{S' : \mathsf{Friend}(S, S')} S'.T \right) \oplus S.U_G.$$

Here, tutoring improves a student's GPA only when both the student and at least one of the student's friends are tutored, up to noise.

The exogenous variables are independent across attributes and across student instances, with $S.U_T \sim$ Bernoulli$(0.2)$, $S.U_G \sim$ Bernoulli$(0.3)$.

STRUCTURAL CAUSAL MODELS: GROUND LEVEL

- **Ground standard SCM.** Although the standard SCM formalism does not usually introduce a separate notion of a ground model, the i.i.d. sampling assumption implicitly induces one copy of the SCM for each student. For the skeleton with students $s_1, \ldots, s_{100}$, the ground variables are

$$\mathbf{V}_\rho = \{T^{(s_i)}, G^{(s_i)} : i = 1, \ldots, 100\}, \qquad \mathbf{U}_\rho = \{U_T^{(s_i)}, U_G^{(s_i)} : i = 1, \ldots, 100\}.$$

  The ground mechanisms are

$$T^{(s_i)} \leftarrow U_T^{(s_i)}, \qquad G^{(s_i)} \leftarrow T^{(s_i)} \oplus U_G^{(s_i)}.$$

  The exogenous variables are mutually independent, with $U_T^{(s_i)} \sim$ Bernoulli$(0.2)$, $\qquad U_G^{(s_i)} \sim$ Bernoulli$(0.3)$.

- **Ground relational SCM.** The ground relational SCM for the friendship skeleton is $M_\rho = \langle \mathbf{V}_\rho, \mathbf{U}_\rho, \mathcal{F}_\rho, P(\mathbf{U}_\rho) \rangle$, with

$$\mathbf{V}_\rho = \{s_i.T, s_i.G : i = 1, \ldots, 100\}, \qquad \mathbf{U}_\rho = \{s_i.U_T, s_i.U_G : i = 1, \ldots, 100\}.$$

  For every student $s_i$, the tutoring mechanism is $s_i.T \leftarrow s_i.U_T$. Since each student has exactly one friend, let $\mathrm{fr}(s_{2i-1}) = s_{2i}$, $\mathrm{fr}(s_{2i}) = s_{2i-1}$. Then, the GPA mechanisms are

$$s_j.G \leftarrow (s_j.T \wedge \mathrm{fr}(s_j).T) \oplus s_j.U_G.$$

  The exogenous variables are mutually independent, with $s_i.U_T \sim$ Bernoulli$(0.2)$, $s_i.U_G \sim$ Bernoulli$(0.3)$.

CAUSAL GRAPH: TEMPLATE LEVEL

- **Standard causal graph.** The standard SCM induces the graph $\mathcal{G}_{\mathrm{std}} : T \to G$.

- **Relational causal graph.** The relational SCM induces a relational causal graph $\mathcal{G}_{\mathrm{rel}}$ with:

    1. non-relational nodes $S.T$ and $S.G$,
    2. a non-relational edge $S.T \to S.G$,
    3. a relational node $S.R_{\mathsf{Friend}}$, and
    4. relational edges $S.T \overset{\mathsf{Friend}(S,S'),\vee}{\to} S.R_{\mathsf{Friend}} \to S.G$

  Intuitively, $S.G$ depends on $S.T$ and on the logical OR of $S'.T$ among students $S'$ such that $\mathsf{Friend}(S, S')$.

CAUSAL GRAPH: GROUND AND MARGINALIZED GROUND

- **Ground standard graph.** The implicit ground graph contains one disconnected copy of $T \to G$ for each student: $\mathcal{G}_{\mathrm{std},\rho} : T^{(s_i)} \to G^{(s_i)}$, $\quad i = 1, \ldots, 100$. There are no edges between distinct students.

- **Ground relational graph.** The ground relational graph $\mathcal{G}_{\mathrm{rel},\rho}$ contains, for each student $s_j$, the non-relational edge $s_j.T \to s_j.G$, and a relational node $s_j.R_{\mathsf{Friend}}$ collecting the tutoring statuses of $s_j$'s friends: $\mathrm{fr}(s_j).T \to s_j.R_{\mathsf{Friend}} \to s_j.G$.

- **Marginalized ground relational graph.** After marginalizing the relational node $s_j.R_{\mathsf{Friend}}$, the resulting marginalized ground graph $\bar{\mathcal{G}}_{\mathrm{rel},\rho}$ has a direct edge into a given student's GPA from the tutoring status of that student's friend:

$$s_j.T \to s_j.G, \qquad \mathrm{fr}(s_j).T \to s_j.G.$$

CAUSAL EFFECTS

We now compare two within-skeleton interventional queries, for the given set of students. The first query compares tutoring all students to tutoring no students. The second query compares tutoring only the even-indexed students to tutoring no students.

For both queries, the outcome is the average GPA indicator $\overline{G} = \frac{1}{100} \sum_{i=1}^{100} G^{(s_i)}$ in the standard SCM, and $\overline{G} = \frac{1}{100} \sum_{i=1}^{100} s_i.G$ in the relational SCM.

**Query 1: tutoring all students versus tutoring no students.**   The first intervention contrast is $do(\mathbf{T} = \mathbf{1})$ versus $do(\mathbf{T} = \mathbf{0})$. where $\mathbf{T} = (T^{(s_i)})_{i=1,\dots,100}$ in the standard SCM and $\mathbf{T} = (s_i.T)_{i=1,\dots,100}$ in the relational SCM.

- **Standard SCM.** Becuase students are i.i.d. in the standard setting, we have

$$P(g^{(s_1)}, \dots g^{(s_{100})} \mid do(t^{(s_1)}, \dots, t^{(s_{100})})) = \prod_{i=1}^{100} P(g^{(s_i)} \mid do(t^{(s_1)}, \dots t^{(s_{100})}))$$

  (Rule 1 of do-calculus, students' grades are conditionally independent in $\mathcal{G}_{\text{std},\rho}$)

$$= \prod_{i=1}^{100} P(g^{(s_i)} \mid do(t^{(s_i)}))$$

  (Rule 2 of do-calculus, no directed path from one student's tutoring status to another's GPA in $\mathcal{G}_{\text{std},\rho}$)

  Since the students are identically distributed, we additionally have that $P(g^{(s_i)} \mid do(t^{(s_i)})) = P(g^{(s_j)} \mid do(t^{(s_j)}))$ for any $i, j$. By linearity of expectation,

$$\mathbb{E}[\overline{G} \mid do(\mathbf{T} = \mathbf{t})] = \frac{1}{100} \sum_{i=1}^{100} \mathbb{E}[G^{(s_i)} \mid do(T^{(s_i)} = t_i)].$$

  Under $do(\mathbf{T} = \mathbf{1})$, every student has $T^{(s_i)} = 1$, so $G^{(s_i)} \leftarrow 1 \oplus U_G^{(s_i)}$ implies $P(G^{(s_i)} = 1 \mid do(T^{(s_i)} = 1)) = P(U_G^{(s_i)} = 0) = 0.7$. Therefore,

$$\mathbb{E}[\overline{G} \mid do(\mathbf{T} = \mathbf{1})] = \frac{1}{100} \sum_{i=1}^{100} 0.7 = 0.7.$$

  Under $do(\mathbf{T} = \mathbf{0})$, a similar calculation yields $\mathbb{E}[\overline{G} \mid do(\mathbf{T} = \mathbf{0})] = 0.3$.
  Thus, $\text{ATE}_{\text{std}}^{\text{all}} = \mathbb{E}[\overline{G} \mid do(\mathbf{T} = \mathbf{1})] - \mathbb{E}[\overline{G} \mid do(\mathbf{T} = \mathbf{0})] = 0.7 - 0.3 = 0.4$.

- **Relational SCM.**

  In the relational SCM, recall that a student's outcome depends both on the student's own tutoring and on the tutoring status of the student's friend:
$$s_j.G \leftarrow (s_j.T \wedge \text{fr}(s_j).T) \oplus s_j.U_G.$$

  Under $do(\mathbf{T} = \mathbf{1})$, every student and every student's friend are tutored. Hence, for each $s_j$, $s_j.G \leftarrow (1 \wedge 1) \oplus s_j.U_G = 1 \oplus s_j.U_G$. Therefore, $\Pr(s_j.G = 1 \mid do(\mathbf{T} = \mathbf{1})) = \Pr(s_j.U_G = 0) = 0.7$. Thus, $\mathbb{E}[\overline{G} \mid do(\mathbf{T} = \mathbf{1})] = 0.7$.

  Under $do(\mathbf{T} = \mathbf{0})$, neither a student nor the student's friend is tutored. A similar calculation gives $\mathbb{E}[\overline{G} \mid do(\mathbf{T} = \mathbf{0})] = 0.3$. Therefore, $\text{ATE}_{\text{rel}}^{\text{all}} = \mathbb{E}[\overline{G} \mid do(\mathbf{T} = \mathbf{1})] - \mathbb{E}[\overline{G} \mid do(\mathbf{T} = \mathbf{0})] = 0.7 - 0.3 = 0.4$.

Therefore, for the all-treated versus none-treated query, $\text{ATE}_{\text{std}}^{\text{all}} = \text{ATE}_{\text{rel}}^{\text{all}} = 0.4$. By construction, the two models agree for this query numerically. Next, we consider a query for which they differ.

**Query 2: tutoring even-indexed students versus tutoring no students.** Now consider the intervention that tutors exactly the even-indexed students: $do(T^{(s_i)} = 1$ for even $i$, $T^{(s_i)} = 0$ for odd $i)$, compared to the baseline intervention $do(\mathbf{T} = \mathbf{0})$. This policy tutors exactly one student in each friendship pair.

- **Standard SCM.** Again, since each student's GPA depends only on that student's own tutoring, we recall the derivation in the previous query:

$$P(g^{(s_1)}, \ldots g^{(s_{100})} \mid do(t^{(s_1)}, \ldots, t^{(s_{100})})) = \prod_{i=1}^{100} P(g^{(s_i)} \mid do(t^{(s_i)}))$$

Since the even-indexed students have $\Pr(G^{(s_i)} = 1 \mid do(T^{(s_i)} = 1)) = 0.7$ and and the odd-indexed students have $\Pr(G^{(s_i)} = 1 \mid do(T^{(s_i)} = 0)) = 0.3$, we get

$$\mathbb{E}[\overline{G} \mid do(T^{(s_i)} = 1 \text{ for even } i, \ T^{(s_i)} = 0 \text{ for odd } i)] = \frac{50}{100}(0.7) + \frac{50}{100}(0.3) = 0.5.$$

As before, $\mathbb{E}[\overline{G} \mid do(\mathbf{T} = \mathbf{0})] = 0.3$, giving $\mathrm{ATE}_{\mathrm{std}}^{\mathrm{even}} = 0.5 - 0.3 = 0.2$.

- **Relational SCM.**

Recall that in the the relational SCM, $s_j.G = (s_j.T \wedge \mathrm{fr}(s_j).T) \oplus s_j.U_G$. Since each friendship pair contains exactly one tutored student, we have $s_j.T \wedge \mathrm{fr}(s_j).T = 0$. Therefore $s_j.G = 0 \oplus s_j.U_G = s_j.U_G$. Therefore, $\Pr(s_j.G = 1 \mid do(T^{(s_i)} = 1$ for even $i$, $T^{(s_i)} = 0$ for odd $i)) = 0.3$, and so

$$\mathbb{E}[\overline{G} \mid do(T^{(s_i)} = 1 \text{ for even } i, \ T^{(s_i)} = 0 \text{ for odd } i)] = 0.3.$$

As before, $\mathbb{E}[\overline{G} \mid do(\mathbf{T} = \mathbf{0})] = 0.3$. This implies that $\mathrm{ATE}_{\mathrm{rel}}^{\mathrm{even}} = 0.3 - 0.3 = 0$.

Therefore, for the even-treated versus none-treated query, $\mathrm{ATE}_{\mathrm{std}}^{\mathrm{even}} = 0.2$ whereas $\mathrm{ATE}_{\mathrm{rel}}^{\mathrm{even}} = 0..$ This second query highlights a key difference between the two models. In the standard SCM, tutoring an even-indexed student improves that student's GPA regardless of the treatment status of other students. In the relational SCM, however, tutoring a student improves GPA only when that student's friend is also tutored. Since the even-treated policy tutors exactly one student in each friendship pair, the GPA mechanism receives no effective tutoring signal for any student, and the ATE is 0. Thus, if the true data-generating process is this relational SCM, then the standard SCM methodology yields an incorrect estimate of the causal effect of this policy.

### C.2. Extended Traffic Example

In this section, we consider an extended version of our running example (Ex. 1) with unobserved confounding across objects.

**Example C.1** (Extended relational schema)**.** We extend the schema in Ex. 1 with an attribute Ped.$V$ for whether they a pedestrian is visible and Ped.$A$ for whether they look alert. An extended relational schema for traffic scenes would be

$$\mathcal{E} = \{\text{Signal (Sig), Car (Car), Pedestrian (Ped)}\}$$
$$\mathcal{R} = \{\text{Ctrl(Sig, Ped), Ctrl(Sig, Car), Path(Ped, Car)}\}$$
$$\mathcal{A} = \{\text{Sig}.W, \text{Ped}.V, \text{Ped}.A, \text{Ped}.X, \text{Car}.B\},$$

with all attributes binary-valued: Sig.$W \in \{1, 0\}$ denotes walk/drive; Ped.$V \in \{1, 0\}$ visible/not; Ped.$A \in \{1, 0\}$ alert/not; Ped.$X \in \{1, 0\}$ cross/wait; and Car.$B \in \{1, 0\}$ brake/go. Ctrl(Sig, Ped) (resp. Ctrl(Sig, Car)) indicates that a signal controls a pedestrian (resp. car), and Path(Ped, Car) indicates that the pedestrian is in the car's path.

Since the object and relation types are the same as in Ex. 1, the three relational skeletons shown in Fig. 1 can also be viewed as skeletons of this new schema. We give an example RSCM for this schema that departs from Ex. 3 in two ways: (i) it includes unobserved confounding, and (ii) it includes a relational parent consisting of more than one variable, illustrating why $\mathbf{W}$ in Def. 3.1 is a set and not a single variable.

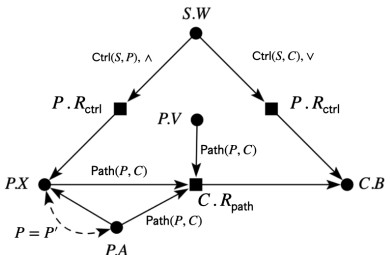

*(a)* Relational causal graph $\mathcal{G}$ for extended traffic RSCM in Ex 10.

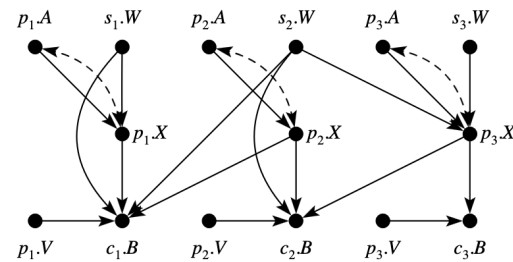

*(b)* Marginalized ground relational causal graph $\bar{\mathcal{G}}_{\rho_C}$ for skeleton $\rho_C$ in Fig. 1.

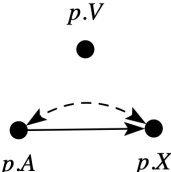

*(c)* Induced subgraph of $\bar{\mathcal{G}}_{\rho_C}$ on attributes of pedestrian $p_1$ (with instance identifiers omitted) for Ex. C.2.

*Figure C.2.1.* Relational causal graphs for the extended traffic examples.

*Example* 10 (RSCM for traffic scene). The endogenous variables are $\mathbf{V} = \{\text{Sig}.W, \text{Ped}.V, \text{Ped}., \text{Ped}.X, \text{Car}.B\}$. The exogenous variables $\mathbf{U}$ capturing unobserved factors (e.g., a pedestrian's intent to cross or a driver's alertness) are $\text{Sig}.U_W \sim \mathcal{B}(0.3), \text{Ped}.U_{XA} \sim \mathcal{B}(0.4), P, U_V \sim \mathcal{B}(0.8)$, and $\text{Car}.U_B \sim \mathcal{B}(0.2)$. The mechanisms are

$$\text{Sig}.W \leftarrow \text{Sig}.U_W,$$
$$\text{Ped}.V \leftarrow \text{Ped}.U_V$$
$$\text{Ped}.A \leftarrow \text{Ped}.U_{XA}$$
$$\text{Ped}.X \leftarrow \text{Ped}.U_{XA} \oplus \bigwedge_{\text{Ctrl}(\text{Sig},\text{Ped})} \text{Sig}.W,$$
$$\text{Car}.B \leftarrow \text{Car}.U_B \oplus \left( \bigvee_{\text{Ctrl}(\text{Sig},\text{Car})} \text{Sig}.W \vee \bigvee_{\text{Path}(\text{Ped},\text{Car})} (\text{Ped}.X \vee \text{Ped}.A) \wedge \text{Ped}.V \right).$$

For each pedestrian, $\text{Ped}.A$ and $\text{Ped}.X$ are confounded by the pedestrian's unobserved intent $\text{Ped}.U_{XA}$. $\text{Car}.B$ has a relational parent $(\{\text{Sig}.W\}, \text{Ctrl}(\text{Sig}, \text{Car}), \vee)$ as in Ex. 3. It also has the relational parent $(\{\text{Ped}.V, \text{Ped}.A, \text{Ped}.X\}, \text{Path}(\text{Ped}, \text{Car}), \bot)$ which means: for each pedestrian $p$ in the path of the car, the mechanism for $\text{Car}.B$ may jointly use that pedestrian's triple $(\text{Ped}.V, \text{Ped}.A, \text{Ped}.X)$. The relational causal graph for this RSCM is shown in Fig. C.2.1a.

It is important that $\{\text{Ped}.V, \text{Ped}.A, \text{Ped}.X\}$ appears as a single relational parent rather than as three separate relational parents $(\{\text{Ped}.V\}, \text{Path}(\text{Ped}, \text{Car})), (\{\text{Ped}.A\}, \text{Path}(\text{Ped}, \text{Car}))$, and $(\{\text{Ped}.X\}, \text{Path}(\text{Ped}, \text{Car}))$. This is because the braking condition depends on a within-pedestrian interaction: $\exists p$ in path : $\text{Ped}.V \wedge (\text{Ped}.A \vee \text{Ped}.X)$, i.e., a given pedestrian must be visible and either crossing or alert to trigger braking. If we aggregate $\text{Ped}.V, \text{Ped}.A$, and $\text{Ped}.X$ separately across pedestrians, we lose information about whether these properties are true of the same pedestrian. In general,

$$\bigvee_{\text{Path}(\text{Ped},\text{Car})} (\text{Ped}.X \vee \text{Ped}.A) \wedge \text{Ped}.V \neq \left( \bigvee_{\text{Path}(\text{Ped},\text{Car})} \text{Ped}.X \vee \bigvee_{\text{Path}(\text{Ped},\text{Car})} \text{Ped}.A \right) \wedge \bigvee_{\text{Path}(\text{Ped},\text{Car})} \text{Ped}.V$$

The right-hand side can be true even when no single pedestrian satisfies both conditions—for instance, one pedestrian is visible but not crossing/alert, while another is crossing/alert but not visible. The left-hand side is false in that situation.

This illustrates why Def. 3.1 allows a **relational parent to contain a set of variables**: it lets the mechanism represent interactions among attributes of the same related object. □

Next, we illustrate an application of Prop. 4.5 to show non-identifiability in this example.

**Example C.2** (Relational non-identifiability using Prop. 4.5). Continuing Ex. 10, consider the causal diagram $\mathcal{G}$ in Fig. C.2.1a, the source skeleton $\rho = \rho_A$, and the target skeleton $\rho_\star = \rho_C$ from Fig. 1. Say we have as input source distributions $\mathbb{P} = \{P(\mathbf{v}_\rho), P(\mathbf{v}_\rho \mid do(p_1.x)), P(\mathbf{v}_\rho \mid do(p_1.x, p_1.a))\}$, and we are interested in the query $P^{\rho_\star}(p_1.x \mid do(p_1.a))$ in $\rho_\star$, the causal effect of pedestrian $p_1$'s alertness on whether or not they cross. Notice how in the marginalized

ground graph $\bar{\mathcal{G}}_{\rho_C}$ (Fig. C.2.1b) we see a bow-graph (Pearl, 2009) structure over $p_1.A$ and $p_1.X$. Standard identification theory usually suggests that in this case, $P^{\rho_\star}(p_1.x \mid do(p_1.a))$ is not identifiable from $\mathcal{G}_{\rho_C}$ and $P(\mathbf{v}_{\rho_\star})$. We show, using Prop. 4.5, that it is also not relationally identifiable from $\mathbb{P}$ and $\mathcal{G}$.

Following the notation of Prop. 4.5, our query concerns attributes of the instance $x = p_1$ in target $\rho_\star$, where $\mathbf{V}_{p_1} = \{p_1.V, p_1.X, p_1.A\}$.

The induced subgraph of $\bar{\mathcal{G}}_{\rho_C}$ on $\mathbf{V}_{p_1}$ (with instance identifiers omitted) is given in Fig. C.2.1c. From $\mathbb{P}$, we construct the restriction $\mathbb{P}|_P$ for instances $p_1, p_2$ in $\rho$ of type $P$ (Pedestrian) as follows. Note that $\mathbf{V}_{p_1} = \{p_1.V, p_1.X, p_1.A\}$ and $\mathbf{V}_{p_2} = \{p_2.V, p_2.X, p_2.A\}$.

1. $P(\mathbf{v}_\rho) \in \mathbb{P}$
    - $p_1$ gives $P(\mathbf{v}_\rho \cap \mathbf{v}_{p_1}) = P(\mathbf{v}_{p_1})$
    - $p_2$ gives $P(\mathbf{v}_\rho \cap \mathbf{v}_{p_2}) = P(\mathbf{v}_{p_2})$

2. $P(\mathbf{v}_\rho \mid do(p_1.x))$
    - $p_1$ gives $P(\mathbf{v}_\rho \cap \mathbf{v}_{p_1} \mid do(\{p_1.x\} \cap \mathbf{v}_{p_1})) = P(\mathbf{v}_{p_1} \mid do(p_1.x))$
    - $p_2$ gives $P(\mathbf{v}_\rho \cap \mathbf{v}_{p_2} \mid do(\{p_1.x\} \cap \mathbf{v}_{p_2})) = P(\mathbf{v}_{p_2})$

3. $P(\mathbf{v}_\rho \mid do(p_1.x, p_1.a))$
    - $p_1$ gives $P(\mathbf{v}_\rho \cap \mathbf{v}_{p_1} \mid do(\{p_1.x, p_1.a\} \cap \mathbf{v}_{p_1})) = P(\mathbf{v}_{p_1} \mid do(p_1.x, p_1.a))$
    - $p_2$ gives $P(\mathbf{v}_\rho \cap \mathbf{v}_{p_2} \mid do(\{p_1.x, p_1.a\} \cap \mathbf{v}_{p_2})) = P(\mathbf{v}_{p_2})$

Omitting identifiers, we get the restriction $\mathbb{P}|_P = \{P(\mathsf{Ped}.v, \mathsf{Ped}.X, \mathsf{Ped}.a), P(\mathsf{Ped}.v, \mathsf{Ped}.X, \mathsf{Ped}.a \mid do(\mathsf{Ped}.X)), P(\mathsf{Ped}.v, \mathsf{Ped}.X, \mathsf{Ped}.a \mid do(\mathsf{Ped}.X, \mathsf{Ped}.a))\}$ and the query $P(\mathsf{Ped}.X \mid do(\mathsf{Ped}.a))$. By counterfactual calculus, since the subgraph $\mathcal{G}_P$ in Fig. C.2.1c contains a bow-structure over $\mathsf{Ped}.A$ and $\mathsf{Ped}.X$, the query $P(\mathsf{Ped}.X \mid do(\mathsf{Ped}.a))$ is non-identifiable from $\mathcal{G}_P$ and $\mathbb{P}|_P$. Then, by Prop. 4.5, the original query $P^{\rho_\star}(p_1.x \mid do(p_1.a))$ is non-identifiable from $\mathbb{P}$ and $\mathcal{G}$.

# D. Further Results and Proofs

## D.1. Proofs for Sec. 3

The following proposition justifies how two isomorphic skeletons induce the 'same' counterfactual distributions over variables.

**Proposition D.1** (Isomorphism-invariance of RSCM distributions). *Consider an RSCM $\mathcal{M} = \langle \mathcal{S}, \mathbf{V}, \mathbf{U}, \mathcal{F}, P(\mathbf{U}) \rangle$ and relational skeletons $\rho, \rho'$ isomorphic under a mapping $\pi$. Then, for any counterfactual events $\mathbf{Y_x}, \ldots, \mathbf{Z_w}$ over $\mathbf{V}_\rho$,*

$$P^{\mathcal{M}_\rho}(\mathbf{y_x}, \ldots, \mathbf{z_w}) = P^{\mathcal{M}_{\rho'}}(\pi(\mathbf{y})_{\pi(\mathbf{x})}, \ldots, \pi(\mathbf{z})_{\pi(\mathbf{w})})$$

*where $\pi(o.A) = \pi(o).A$ extends $\pi$ to ground variables $o.A \in \mathbf{V}_\rho$.*

*Proof.* Recall from Def. B.5 that

$$P^{\mathcal{M}_\rho}(\mathbf{y_x}, \ldots, \mathbf{z_w}) = \sum_{\mathbf{u}_\rho} \mathbf{1}[\mathbf{Y_x}(\mathbf{u}_\rho) = \mathbf{y}, \ldots, \mathbf{Z_w}(\mathbf{u}_\rho) = \mathbf{z}] \, P(\mathbf{u}_\rho),$$

and

$$P^{\mathcal{M}_{\rho'}}(\pi(\mathbf{y})_{\pi(\mathbf{x})}, \ldots, \pi(\mathbf{z})_{\pi(\mathbf{w})}) = \sum_{\mathbf{u}_{\rho'}} \mathbf{1}[\pi(\mathbf{Y})_{\pi(\mathbf{x})}(\mathbf{u}_{\rho'}) = \pi(\mathbf{y}), \ldots, \pi(\mathbf{Z})_{\pi(\mathbf{w})}(\mathbf{u}_{\rho'}) = \pi(\mathbf{z})] \, P(\mathbf{u}_{\rho'}).$$

We prove the desired equality by a change of variables $\mathbf{u}_{\rho'} = \pi(\mathbf{u}_\rho)$. The isomorphism $\pi : \rho \to \rho'$ induces a bijection on the exogenous assignments $\mathbf{u}_\rho \mapsto \pi(\mathbf{u}_\rho)$. Therefore, we re-index the second sum by writing $\mathbf{u}_{\rho'} = \pi(\mathbf{u}_\rho)$.

$$P^{\mathcal{M}_{\rho'}}(\pi(\mathbf{y})_{\pi(\mathbf{x})}, \ldots, \pi(\mathbf{z})_{\pi(\mathbf{w})}) = \sum_{\pi(\mathbf{u}_\rho)} \mathbf{1}[\pi(\mathbf{Y})_{\pi(\mathbf{x})}(\pi(\mathbf{u}_\rho)) = \pi(\mathbf{y}), \ldots, \pi(\mathbf{Z})_{\pi(\mathbf{w})}(\pi(\mathbf{u}_\rho)) = \pi(\mathbf{z})] \, P(\pi(\mathbf{u}_\rho)).$$

We claim that for any intervention assignment $\mathbf{x}$ and any exogenous assignment $\mathbf{u}_\rho$,

$$\pi(\mathbf{Y})_{\pi(\mathbf{x})}(\pi(\mathbf{u}_\rho)) = \pi(\mathbf{y}) \text{ in } \mathbf{M}_{\rho'} \iff \mathbf{Y}_{\mathbf{x}}(\mathbf{u}_\rho) = \mathbf{y} \text{ in } \mathbf{M}_\rho$$

and similarly for other counterfactual events $\mathbf{Z}_{\mathbf{w}}$.

To see this, fix a ground variable $o.A \in \mathbf{V}_\rho$. By Def. B.4, each relational parent $(\mathbf{W}, \phi, \text{AGG})$ in the template mechanism $f_{O.A}$ is instantiated in $\mathcal{M}_\rho$ as the multiset

$$\{\, t.W \mid T.W \in \mathbf{W}, \ t \in \rho(T), \ \phi(o, t) \text{ holds in } \rho \,\},$$

and analogously in $\mathcal{M}_{\rho'}$. Since $\pi$ is a skeleton isomorphism, it preserves relations and hence satisfaction of constraints:

$$\phi(o, t) \text{ holds in } \rho \iff \phi(\pi(o), \pi(t)) \text{ holds in } \rho'.$$

Hence the structural function (or intervened constant) for $\pi(o).A$ in $\mathcal{M}_{\rho'}$ is exactly the $\pi$-renaming of the structural function for $o.A$ in $\mathcal{M}_\rho$. This proves the claim, so that for every $\mathbf{u}_\rho$,

$$\mathbf{1}[\pi(\mathbf{Y})_{\pi(\mathbf{x})}(\pi(\mathbf{u}_\rho)) = \pi(\mathbf{y}), \ldots, \pi(\mathbf{Z})_{\pi(\mathbf{w})}(\pi(\mathbf{u}_\rho)) = \pi(\mathbf{z})]$$
$$= \mathbf{1}[\mathbf{Y}_{\mathbf{x}}(\mathbf{u}_\rho) = \mathbf{y}, \ldots, \mathbf{Z}_{\mathbf{w}}(\mathbf{u}_\rho) = \mathbf{z}].$$

It remains to show that for every assignment of values $\mathbf{u}_\rho$,

$$P(\mathbf{u}_\rho) \text{ in } \mathcal{M}_\rho = P(\pi(\mathbf{u}_\rho)) \text{ in } \mathcal{M}_{\rho'}$$

For each entity/relation type $O$, recall that the RSCM $\mathcal{M}$ specifies for each exogenous variable $O.U \in \mathbf{U}$ a distribution $O.U \sim P(O.U)$. By definition of the ground RSCM $\mathcal{M}_\rho$, for each $o \in \rho(O)$, $o.U \sim P(O.U)$. Since $\pi$ preserves types, we also have $\pi(o).U \sim P(O.U)$ by definition of the ground RSCM $\mathcal{M}_{\rho'}$. Therefore, the above equality follows. $\square$

*Theorem* 3.3 (Impossibility of observational inference across skeletons). Consider a schema $\mathcal{S}$, source skeletons $\rho_1, \ldots, \rho_l$, and target skeleton $\rho_\star$. Then, for any RSCM $\mathcal{M}$ over $\mathcal{S}$, there exists another RSCM $\mathcal{M}'$ over $\mathcal{S}$ such that $\mathcal{M}$ and $\mathcal{M}'$ agree on observational distributions $P(\mathbf{v}_{\rho_k})$ for every source skeleton $\rho_k$ but disagree on the observational distribution $P(\mathbf{v}_{\rho_\star})$ of the target skeleton.

*Proof idea.* We will prove this by constructing a relational constraint $\phi_\star$ that evaluates to true only on skeletons isomorphic to the given $\rho_\star$. So, given $\mathcal{M}$, we will construct another SCM $\mathcal{M}'$ that is almost identical to $\mathcal{M}$, except that it has different behaviour when $\phi_\star$ is true. For example, such a constraint for $\rho_A$ given in Ex. 1 would be:

$$\phi_A : \exists \text{ Signal } S_1, \text{ Pedestrian } P_1, P_2, \text{ Car } C_1, C_2 \text{ such that } (P_1 \neq P_2) \wedge (C_1 \neq C_2)$$
$$\wedge \, (\forall \text{ Signal } S, S = S_1)$$
$$\wedge \, (\forall \text{ Pedestrian } P, P = P_1 \vee P = P_2)$$
$$\wedge \, (\forall \text{ Car } C, C = C_1 \vee C = C_2)$$
$$\wedge \, \mathsf{Ctrl}(S_1, P_1) \wedge \mathsf{Ctrl}(S_1, P_2)$$
$$\wedge \, \mathsf{Ctrl}(S_1, C_1) \wedge \neg \mathsf{Ctrl}(S_1, C_2)$$
$$\wedge \, \mathsf{Path}(P_1, C_1) \wedge \mathsf{Path}(P_1, C_2)$$
$$\wedge \, \mathsf{Path}(P_2, C_1) \wedge \neg \mathsf{Path}(P_2, C_2)$$

*Proof.* Since $\rho_\star$ is a finite relational skeleton, there exists a first-order formula $\phi_\star$ that is true for a given skeleton $\rho$ if and only if $\rho \cong \rho_\star$. Such a $\phi_\star$ is constructed as follows. For each entity/relation type $O$ and instance $o$ of $O$ in $\rho_\star$, introduce one existentially quantified variable. Check that each of these variables (for a given type) are distinct. Introduce a universally

quantified variable of type $O$, and check that it is equal to atleast one of these $o$-variables. Finally, check that every relation $R$ in the schema $\mathcal{S}$ holds on exactly those instance variables for which it is true in $\rho_\star$, and no others. By construction, $\phi_\star$ is true only on skeletons isomorphic to $\rho_\star$.

Having constructed $\phi_\star$, consider the given RSCM $\mathcal{M}$ and skeletons $\rho_1, \ldots, \rho_l$. Let $\mathcal{M}'$ be the same as $\mathcal{M}$, with the following changes. First, $\mathcal{M}'$ contains, for some arbitrary entity or relation type $O$ with an observed attribute $O.A \in \mathbf{V}$, an additional exogenous variable $O.U$ with the same domain as $O.A$, so that $\mathbf{U}' = \mathbf{U} \cup \{O.U\}$. Second, $\mathcal{M}'$ has a function $f'_{O.A}$ as follows:

$$f'_{O.A}(\mathbf{pa}_{O.A}, \mathbf{u}'_{O.A}, \mathbf{pa}^r_{O.A}, \mathbf{u}^r_{O.A}) = \begin{cases} u_{O.A} & \phi_\star(X) \\ f_{O.A}(\mathbf{pa}_{O.A}, \mathbf{u}'_{O.A} \setminus \{O.u\}, \mathbf{pa}^r_{O.A}, \mathbf{u}^r_{O.A}) & \neg\phi_\star(X) \end{cases}$$

As a result, since $f'_{O.A}$ in $\mathcal{M}'$ is equal to $f_{O.A}$ in $\mathcal{M}$ whenever $\phi_\star$ is false, $\mathcal{M}'$ and $\mathcal{M}$ will induce the same observational distributions on skeletons $\rho_1, \ldots, \rho_l \not\cong \rho_\star$. On the skeleton $\rho_\star$, however, the distribution $P(O.U)$ in $\mathcal{M}'$ can be chosen depending on $P^{\mathcal{M}}(\mathbf{v}_{\rho_\star})$ so that the different behaviour of $f'_{O.A}$ and $f_{O.A}$ results in different observational distributions of $\mathcal{M}'$ and $\mathcal{M}$. $\qquad\square$

*Remark* D.1. The proof of Thm. 3.3 above does not rely on unobserved confounding between variables (be it of the same entity/relation instance, or across such instances). As such, Thm. 3.3 holds even if we restrict to the class of Markovian RSCMs.

*Theorem* 3.4 (Impossibility of causal inference within a skeleton). Consider a schema $\mathcal{S}$ where at least one entity or relation type has more than one observed attribute. For any relational SCM $\mathcal{M}$ over $\mathcal{S}$ and skeleton $\rho$, there exists another relational SCM $\mathcal{M}'$ over $\mathcal{S}$ such that $\mathcal{M}$ and $\mathcal{M}'$ agree on the observational distribution $P(\mathbf{v}_\rho)$ but disagree on some interventional distribution over $\mathbf{V}_\rho$.

*Proof idea.* We will construct an SCM $\mathcal{M}'$ such that for an entity/relation type $O$ with observed attributes $O.A$ and $O.B$, the function $f'_{O.B}$ in $\mathcal{M}'$ can 'detect' that $O.A$ has been intervened on (for the same instance). Then, $f'_{O.B}$ has different behaviour than $f_{O.B}$ when $O.A$ is under intervention, but the same behaviour otherwise. The fact that $O.A$ and $O.B$ belong to the same instance allows us to implement such 'detection', since the constraints in $f_{O.A}$ that hold for an instance $o$ will also hold for $o$ in $f'_{O.B}$.

*Proof.* By assumption, there exists some entity/relation type $O$ such that $O.A, O.B \in \mathbf{V}$. First, assume WLOG, that $O.B \notin \mathbf{Pa}_{O.A}$ in $\mathcal{M}$. Define $\mathcal{M}'$ to be the same as $\mathcal{M}$, but with two modifications.

First, $\mathcal{M}'$ contains an additional exogenous variable $O.U$ with the same domain as $O.B$, so that $\mathbf{U}' = \mathbf{U} \cup \{O.U\}$. Second, $\mathcal{M}'$ has a function $f'_{O.B}$ as follows:

$$f'_{O.B}(\mathbf{pa}_{O.B} \cup \mathbf{pa}_{O.A} \cup \{O.a\}, \mathbf{u}'_{O.B} \cup \mathbf{u}_{O.A}, \mathbf{pa}^r_{O.B} \cup \mathbf{pa}^r_{O.A}, \mathbf{u}^r_{O.B} \cup \mathbf{u}^r_{O.A})$$
$$= \begin{cases} f_{O.B}(\mathbf{pa}_{O.B}, \mathbf{u}'_{O.B} \setminus \{O.U\}, \mathbf{pa}^r_{O.B}, \mathbf{u}^r_{O.B}) & \text{if } O.a = f_{O.A}(\mathbf{pa}_{O.A}, \mathbf{u}'_{O.A} \setminus \{O.u\}, \mathbf{pa}^r_{O.A}, \mathbf{u}^r_{O.A}) \\ O.u & \text{otherwise} \end{cases}$$

Above, $f'_{O.B}$ has an extended parent set for $O.B$ that takes in all the parents (endogenous and exogenous, non-relational and relational) of $O.A$. Since $O.A$ and $O.B$ belong to the same instance, for any relational parent $(\mathbf{W}, \phi) \in \mathbf{Pa}^r_{O.A} \cup \mathbf{U}^r_{O.A}$, and any skeleton $\rho$, $\phi(o, t)$ will hold in $f'_{O.B}$ in $\rho$ iff it holds in $f_{O.A}$ in $\rho$.

In the observational regime, the condition $O.a = f_{O.A}(\ldots)$ is always true; therefore, $f'_{O.B}$ is exactly equal to $f_{O.B}$, and $\mathcal{M}$ and $\mathcal{M}'$ induce the same $P(\mathbf{v}_{\rho_\star})$. However, this is not the case under intervention. Consider an intervention that sets $o_1.A = a$ for some instance $o_1$ of $O$ in $\rho$. There is some assignment $\mathbf{u}_\rho$ to the exogenous variables such that the valuation $o_1.A(\mathbf{u}_\rho) \neq a$. Under this assignment, the condition $o_1.a = f_{O.A}(\ldots)$ fails, and thus $o_1.B \leftarrow x_1.u$. The probability $P(O.U)$ in $\mathcal{M}'$ can be chosen such that $P^{\mathcal{M}'_\rho}(o_1.B = b \mid do(o_1.A = a)) = P((o_1.U = b) \neq P^{\mathcal{M}_\rho}(o_1.B = b \mid do(o_1.A = a))$ for some $b$ in the domain of $O.B$.

$\qquad\square$

*Remark* D.2. The proof of Thm. 3.4 above does not rely on unobserved confounding between variables of different entity/relation instances. As such, Thm. 3.4 holds even if we restrict to the class of $\rho$-Markovian RSCMs.

### D.2. Proofs for Sec. 4

First, we will show how if an RSCM $\mathcal{M}$ induces by Def. 4.1 a causal graph $\mathcal{G}$, then for any skeleton $\rho$, the ground RSCM $\mathcal{M}_\rho$, when viewed as a standard SCM, induces (by the definition in (Sec. 2)) a causal graph that is equal to the marginalized ground graph $\bar{\mathcal{G}}_\rho$ (Def. B.7).

**Lemma D.3.** *Consider a relational schema $\mathcal{S}$, skeleton $\rho$, and RSCM $\mathcal{M}$ inducing the relational causal graph $\mathcal{G}$. Then, the ground RSCM $\mathcal{M}_\rho$ induces the marginalized ground graph $\bar{\mathcal{G}}_\rho$.*

*Proof.* Let $\mathcal{M}$, $\mathcal{G}$, $\rho$, and $\bar{\mathcal{G}}_\rho$ be as given in the statement of the lemma. Let $\mathcal{G}'$ be the causal graph induced by $\mathcal{M}_\rho$ viewed as a standard SCM, according to Sec. 2). $\mathcal{G}'$ and $\bar{\mathcal{G}}_\rho$ contain the same nodes by construction, since relational nodes have been marginalized from $\mathcal{G}_\rho$ to get $\bar{\mathcal{G}}_\rho$. We will show $\mathcal{G}' = \bar{\mathcal{G}}_\rho$ by showing that every edge in $\bar{\mathcal{G}}_\rho$ is also in $\mathcal{G}'$ and vice-versa.

**Within-instance edges.** Fix a type $O \in \mathcal{E} \cup \mathcal{R}$ and an instance $o \in \rho(O)$. For any variable $o.A, o.B \in \mathbf{V}_\rho$, the endogenous variables in $\mathcal{M}_\rho$,

1. Every within-instance edge in $\mathcal{G}'$ is also an edge in $\bar{\mathcal{G}}_\rho$.

$$
\begin{aligned}
& o.B \in \mathbf{Pa}_{o.A} \text{ in } \mathcal{M}_\rho \\
&\implies O.B \in \mathbf{Pa}_{O.A} \in \mathcal{M} && (\mathcal{M}_\rho \text{ grounds } \mathcal{M} \text{ on } \rho, \text{ Def. B.4}) \\
&\implies O.B \to O.A \in \mathcal{G} && (\text{by construction of } \mathcal{G} \text{ from } \mathcal{M}, \text{ Def. 4.1}) \\
&\implies o.B \to o.A \in \mathcal{G}_\rho && (\text{by construction of } \mathcal{G}_\rho \text{ from } \mathcal{G}, \text{ Def. B.6}) \\
&\implies o.B \to o.A \in \bar{\mathcal{G}}_\rho && (\text{by construction of } \bar{\mathcal{G}}_\rho \text{ from } \mathcal{G}_\rho, \text{ Def. B.7})
\end{aligned}
$$

$$
\begin{aligned}
& \exists o.U \in \mathbf{U}_\rho, o.U \in \mathbf{U}_{o.A} \cap \mathbf{U}_{o.B} \\
&\implies O.U \in \mathbf{U}_{O.A} \cap \mathbf{U}_{O.B} \text{ in } \mathcal{M} && (\text{since } \mathcal{M}_\rho \text{ grounds } \mathcal{M} \text{ on } \rho) \\
&\implies O.A \overset{O=O'}{\leftrightarrow} O.B \in \mathcal{G} && (\text{by construction of } \mathcal{G} \text{ from } \mathcal{M}) \\
&\implies o.A \leftrightarrow o.B \in \mathcal{G}_\rho && (\text{by construction of } \mathcal{G}_\rho \text{ from } \mathcal{G}) \\
&\implies o.A \leftrightarrow o.B \in \bar{\mathcal{G}}_\rho && (\text{by construction of } \bar{\mathcal{G}}_\rho \text{ from } \mathcal{G}_\rho)
\end{aligned}
$$

2. Every within-instance edge in $\bar{\mathcal{G}}_\rho$ is also an edge in $\mathcal{G}'$.

$$
\begin{aligned}
& o.B \to o.A \in \bar{\mathcal{G}}_\rho \\
&\implies o.B \to o.A \in \mathcal{G}_\rho && (\text{within-instance edges preserved by construction of } \bar{\mathcal{G}}_\rho \text{ from } \mathcal{G}_\rho) \\
&\implies O.B \to O.A \in \mathcal{G} && (\text{by construction of } \mathcal{G}_\rho \text{ from } \mathcal{G}) \\
&\implies O.B \in \mathbf{Pa}_{O.A} \text{ in } \mathcal{M} && (\text{by construction of } \mathcal{G} \text{ from } \mathcal{M}) \\
&\implies o.B \in \mathbf{Pa}_{o.A} \text{ in } \mathcal{M}_\rho && (\text{since } \mathcal{M}_\rho \text{ grounds } \mathcal{M} \text{ on } \rho)
\end{aligned}
$$

$$
\begin{aligned}
& o.B \leftrightarrow o.A \in \bar{\mathcal{G}}_\rho \\
&\implies o.B \leftrightarrow o.A \in \mathcal{G}_\rho && (\text{within-instance edges preserved by construction of } \bar{\mathcal{G}}_\rho \text{ from } \mathcal{G}_\rho) \\
&\implies O.B \overset{O=O'}{\leftrightarrow} O.A \in \mathcal{G} && (\text{by construction of } \mathcal{G}_\rho \text{ from } \mathcal{G}) \\
&\implies \exists O.U \in \mathbf{U}, O.U \in \mathbf{U}_{O.A} \cap \mathbf{U}_{O.B} \text{ in } \mathcal{M} && (\text{by construction of } \mathcal{G} \text{ from } \mathcal{M}) \\
&\implies \exists o.U \in \mathbf{U}_\rho, o.U \in \mathbf{U}_{o.A} \cap \mathbf{U}_{o.B} \text{ in } \mathcal{M}_\rho && (\mathcal{M}_\rho \text{ grounds } \mathcal{M} \text{ on } \rho)
\end{aligned}
$$

**Cross-instance edges.** Fix types $O, T \in \mathcal{E} \cup \mathcal{R}$ and non-identical instances $o \in \rho(O), t \in \rho(T)$. For any variables $o.A, t.B \in \mathbf{V}_\rho$,

1. Every cross-instance edge in $\mathcal{G}'$ is also an edge in $\bar{\mathcal{G}}_\rho$.

$t.B \in \mathbf{Pa}_{o.A}$ in $\mathcal{M}_\rho$

$\implies \exists R = (\mathbf{W}, \phi, \text{AGG})$ with $T.B \in \mathbf{W}, R \in \mathbf{Pa}^r_{O.A} \in \mathcal{M}$ such that $\phi(o,t)$     ($\mathcal{M}_\rho$ grounds $\mathcal{M}$ on $\rho$, Def. B.4)

$\implies T.B \overset{\phi,\text{AGG}}{\to} O.R \to O.A \in \mathcal{G}$     (by construction of $\mathcal{G}$ from $\mathcal{M}$, Def. 4.1)

$\implies t.B \overset{\phi,\text{AGG}}{\to} o.R \to o.A \in \mathcal{G}_\rho$     (by construction of $\mathcal{G}_\rho$ from $\mathcal{G}$, Def. B.6)

$\implies o.B \to o.A \in \bar{G}_\rho$     (by construction of $\bar{\mathcal{G}}_\rho$ from $\mathcal{G}_\rho$)

$\exists z.U \in \mathbf{U}_\rho, z.U \in \mathbf{U}_{o.A} \cap \mathbf{U}_{o.B}$

$\implies \exists R_1 = (\mathbf{W_1}, \phi_1, \text{AGG}_1) \in \mathbf{U}^r_{O.A}$ and $R_2 = (\mathbf{W_2}, \phi_2, \text{AGG}_2) \in \mathbf{U}^r_{T.B}$ in $\mathcal{M}$

     with $Z.U \in \mathbf{W_1} \cap \mathbf{W_2}$ and $\phi_1(o,z) \wedge \phi_2(t,z)$     ($\mathcal{M}_\rho$ grounds $\mathcal{M}$ on $\rho$)

$\implies O.A \overset{\exists Z : \phi_1(O,Z) \wedge \phi_2(T,Z)}{\leftrightarrow} T.B \in \mathcal{G}$     (by construction of $\mathcal{G}$ from $\mathcal{M}$)

$\implies o.A \leftrightarrow o.B \in \mathcal{G}_\rho$     (by construction of $\mathcal{G}_\rho$ from $\mathcal{G}$)

$\implies o.A \leftrightarrow o.B \in \bar{\mathcal{G}}_\rho$     (by construction of $\bar{\mathcal{G}}_\rho$ from $\mathcal{G}_\rho$, which preserves bidirected edges)

2. Every cross-instance edge in $\bar{\mathcal{G}}_\rho$ is also an edge in $\mathcal{G}'$.

$t.B \to o.A \in \bar{\mathcal{G}}_\rho$

$\implies t.B \to o.R \to o.A \in \mathcal{G}_\rho$ for some $R = (\mathbf{W}, \phi, \text{AGG})$     (by construction of $\bar{\mathcal{G}}_\rho$ from $\mathcal{G}_\rho$)

$\implies T.B \overset{\phi,\text{AGG}}{\to} O.R \to O.A \in \mathcal{G}$     (by construction of $\mathcal{G}_\rho$ from $\mathcal{G}$)

$\implies R \in \mathbf{Pa}^r_{O.A}$ in $\mathcal{M}$     (by construction of $\mathcal{G}$ from $\mathcal{M}$)

$\implies t.B \in \mathbf{Pa}^r_{o.A}$ in $\mathcal{M}_\rho$     (since $\mathcal{M}_\rho$ grounds $\mathcal{M}$ on $\rho$)

$t.B \leftrightarrow o.A \in \bar{\mathcal{G}}_\rho$

$\implies o.B \leftrightarrow o.A \in \mathcal{G}_\rho$     (bidirected edges preserved by construction of $\bar{\mathcal{G}}_\rho$ from $\mathcal{G}_\rho$)

$\implies O.B \overset{\phi}{\leftrightarrow} O.A \in \mathcal{G}$ for some $\phi$ such that $\phi(o,t)$     (by construction of $\mathcal{G}_\rho$ from $\mathcal{G}$)

$\implies \exists Z.U \in \mathbf{U}, z \in \rho(Z), R_1 = (\mathbf{W_1}, \phi_1, \text{AGG}_1) \in \mathbf{U}^r_{O.A}$ and $R_2 = (\mathbf{W_2}, \phi_2, \text{AGG}_2) \in \mathbf{U}^r_{T.B}$ in $\mathcal{M}$

     with $Z.U \in \mathbf{W_1} \cap \mathbf{W_2}$ and $\phi_1(o,z) \wedge \phi_2(t,z)$     (by construction of $\mathcal{G}$ from $\mathcal{M}$)

$\implies \exists z.U \in \mathbf{U}_\rho, z.U \in \mathbf{U}_{o.A} \cap \mathbf{U}_{t.B}$ in $\mathcal{M}_\rho$     (since $\mathcal{M}_\rho$ grounds $\mathcal{M}$ on $\rho$)

$\square$

**Corollary D.1.** *Consider a relational schema $\mathcal{S}$, skeleton $\rho$, and RSCM $\mathcal{M}$ inducing the relational causal graph $\mathcal{G}$. Then, the ground RSCM $\mathcal{M}_\rho$ induces counterfactual distributions that satisfy all counterfactul equality constraints encoded in $\bar{\mathcal{G}}_\rho$.*

*Proof.* By Lemma D.3, we have that $\mathcal{M}_\rho$ induces $\bar{\mathcal{G}}_\rho$. (Xia et al., 2023, Lemma 1) shows that if a standard SCM $\mathcal{M}$ induces a causal diagram $\mathcal{G}$, then its induced distributions satisfy all counterfactual equality constraints encoded in $\mathcal{G}$. The result follows. $\square$

*Theorem* 4.3 (Observational identification across skeletons). Consider a schema $\mathcal{S}$, relational causal graph $\mathcal{G}$, source skeleton $\rho$, and target skeleton $\rho_\star$. Let $o.A$ be an unconfounded variable in $\mathbf{V}_{\rho_\star}$. The conditional $P(o.a \mid \mathbf{pa}_{o.A}, \mathbf{pa}^r_{o.A})$ is relationally identifiable from $\mathcal{G}$ and $P(\mathbf{v}_\rho)$ if there exists a source instance $o' \in \rho$ such that $o'.A$ is unconfounded and $\text{dom}(\mathbf{Pa}^r_{o.A}) \subseteq \text{dom}(\mathbf{Pa}^r_{o'.A})$. In this case, $P(o.a \mid \mathbf{pa}_{o.A}, \mathbf{pa}^r_{o.A}) = P(o'.a \mid \mathbf{pa}_{o'.A}, \mathbf{pa}^r_{o'.A})$.

*Proof.* Consider any two RSCMs $\mathcal{M}, \mathcal{M}'$ compatible with $\mathcal{G}$ such that $P^{\mathcal{M}_\rho}(\mathbf{v}_\rho) = P^{\mathcal{M}'_\rho}(\mathbf{v}_\rho)$. We need to show that

$$P^{\mathcal{M}_{\rho\star}}(o.a \mid \mathbf{pa}_{o.A}, \mathbf{pa}^r_{o.A}) = P^{\mathcal{M}'_{\rho\star}}(o.a \mid \mathbf{pa}_{o.A}, \mathbf{pa}^r_{o.A}).$$

Consider the $o' \in \rho$ given by the assumption. Since $\mathcal{M}, \mathcal{M}'$ agree on the observational distribution over $P(\mathbf{v}_\rho)$, this implies that

$$P^{\mathcal{M}_\rho}(o'.a \mid \mathbf{pa}_{o'.A}, \mathbf{pa}^r_{o'.A}) = P^{\mathcal{M}'_\rho}(o'.a \mid \mathbf{pa}_{o.A}, \mathbf{pa}^r_{o'.A})$$

It suffices, then, to show that for all values $o.a = o'.a$, $\mathbf{pa}_{o.A} = \mathbf{pa}_{o'.A}$, and $\mathbf{pa}^r_{o.A} = \mathbf{pa}^r_{o'.A}$ (with the latter comparison made for values in the support of $\mathbf{pa}^r_{o.A}$).

$$P^{\mathcal{M}_\rho}(o'.a \mid \mathbf{pa}_{o'.A}, \mathbf{pa}^r_{o'.A}) = P^{\mathcal{M}_{\rho\star}}(o.A \mid \mathbf{pa}_{o.A}, \mathbf{pa}^r_{o.A}) \tag{1}$$

and

$$P^{\mathcal{M}'_\rho}(o'.a \mid \mathbf{pa}_{o'.A}, \mathbf{pa}^r_{o'.A}) = P^{\mathcal{M}'_{\rho\star}}(o.a \mid \mathbf{pa}_{o.A}, \mathbf{pa}^r_{o.A}). \tag{2}$$

Let the structural equation for $o.A$ in $\mathcal{M}_{\rho\star}$ be $o.A \leftarrow f_{O.A}(\mathbf{pa}_{o.A}, \mathbf{u}_{o.A}, \mathbf{pa}^r_{o.A})$, where $\mathbf{U}_{o.A}$ are the exogenous parents of $o.A$ in $\mathcal{M}_{\rho\star}$. By condition (1) of unconfoundedness, $o.A$ shares no bidirected edge with any variable in $\bar{\mathcal{G}}_\rho$. By Prop. D.3, this implies that $\mathbf{U}_{o.A} \perp\!\!\!\perp (\mathbf{Pa}_{o.A}, \mathbf{Pa}^r_{o.A})$. Therefore, for any values $a, \mathbf{pa}_{o.A}, \mathbf{pa}^r_{o.A}$,

$$P^{\mathcal{M}_{\rho\star}}(o.a \mid \mathbf{pa}_{o.A}, \mathbf{pa}^r_{o.A}) = \sum_{\mathbf{u}_{o.A}} P(o.a \mid \mathbf{u}_{o.A}, \mathbf{pa}_{o.A}, \mathbf{pa}^r_{o.A}) P(\mathbf{u}_{o.A} \mid \mathbf{pa}_{o.A}, \mathbf{pa}^r_{o.A})$$

$$= \sum_{\mathbf{u}_{o.A}} P(o.a \mid \mathbf{u}_{o.A}, \mathbf{pa}_{o.A}, \mathbf{pa}^r_{o.A}) P(\mathbf{u}_{o.A}) \qquad (\mathbf{U}_{o.A} \perp\!\!\!\perp (\mathbf{Pa}_{o.A}, \mathbf{Pa}^r_{o.A}))$$

$$= \sum_{\mathbf{u}_{o.A}} \mathbf{1}[f_{O.A}(\mathbf{pa}_{o.A}, \mathbf{u}_{o.A}, \mathbf{pa}^r_{o.A}) = a] P(\mathbf{u}_{o.A})$$

Analogously, for $o'.A$ in $\mathcal{M}_\rho$, we get

$$P^{\mathcal{M}_\rho}(o'.a \mid \mathbf{pa}_{o'.A}, \mathbf{pa}^r_{o'.A}) = \sum_{\mathbf{u}_{o'.A}} \mathbf{1}[f_{O.A}(\mathbf{pa}_{o'.A}, \mathbf{u}_{x].A}, \mathbf{pa}^r_{o'.A}) = a] P(\mathbf{u}_{o'.A})$$

Since $\mathcal{M}_\rho$ and $\mathcal{M}_{\rho\star}$ are both groundings of the same RSCM $\mathcal{M}$, $\mathbf{U}_{o'.A}$ and $\mathbf{U}_{o.A}$ have the same domains, and whenever $\mathbf{u}_{o'.A} = \mathbf{u}_{o.A}$, we also have $P(\mathbf{u}_{o'.A}) = P(\mathbf{u}_{o.A})$. Additionally, whenever $\mathbf{pa}_{o.A} = \mathbf{pa}_{o'.A}$ and $\mathbf{pa}^r_{o.A} = \mathbf{pa}^r_{o'.A}$, because the mechanism $f_{O.A}$ is shared, we also have $\mathbf{1}[f_{O.A}(\mathbf{pa}_{o'.A}, \mathbf{u}_{x].A}, \mathbf{pa}^r_{o'.A}) = a] = \mathbf{1}[f_{O.A}(\mathbf{pa}_{o.A}, \mathbf{u}_{o.A}, \mathbf{pa}^r_{o.A}) = a]$. This proves the equality in Eq.(1). A similar calculation for the groundings of $\mathcal{M}'$ proves the equality in Eq.(2). $\square$

**Corollary D.4** (Observational identification across skeletons - Markovian)**.** *Consider a schema $\mathcal{S}$ and a Markovian relational causal graph $\mathcal{G}$. Given source skeletons $\rho_1, \ldots, \rho_k$ and a target skeleton $\rho_\star$, $P(\mathbf{v}_{\rho_\star})$ is identifiable from the distributions $\{P(\mathbf{v}_{\rho_k})\}_{k=1}^l$ and $\mathcal{G}$ assuming the support condition of Thm. 4.3 is met for every variable $o.A \in \mathbf{V}_{\rho_\star}$ by some $\rho_k$.*

*Proof.* A Markovian graph $\mathcal{G}$ contains no bidirected edges. Therefore, condition (1) of Thm. 4.3 is met for every $o.A \in \mathbf{V}_{\rho_\star}$ by some $o'.A$ in some $\rho_k$. Additionally, since $\mathcal{G}$ is Markovian, for any RSCM $\mathcal{M}$ compatible with $\mathcal{G}$, the ground RSCM $\mathcal{M}_{\rho_\star}$ is also Markovian. Since, for each $o.A \in \mathbf{V}_{\rho_\star}$, $P(o.a \mid \mathbf{pa}_{o.A}, \mathbf{pa}^r_{o.A})$ is identifiable from some $P(\mathbf{v}_{\rho_k})$ by Thm. 4.3, so is the joint $P(\mathbf{v}_{\rho_\star})$ by the by the Markov factorization (Bareinboim, 2025, Thm. 2.4.1) and the compatibility of $P(\mathbf{v}_{\rho_\star})$ and the marginalized ground graph $\bar{\mathcal{G}}_\rho$.

$$P(\mathbf{v}_{\rho_\star}) = \prod_{o.a \in \mathbf{v}_{\rho_\star}} P(o.a \mid \mathbf{pa}_{o.A}, \mathbf{pa}^r_{o.A}).$$

Above, $\mathbf{Pa}_{o.A}, \mathbf{Pa}^r_{o.A} \subseteq \mathbf{V}_\rho$ are the graphical parents of $o.A$ in $\bar{\mathcal{G}}_\rho$ containing variables within the same instance and from different instances respectively. $\square$

*Proposition* 4.4 (Sufficient condition for same-skeleton relational identification)**.** Consider a schema $\mathcal{S}$, relational causal graph $\mathcal{G}$, skeleton $\rho$, and family of interventional distributions $\mathbb{P}$ over $\mathbf{V}_\rho$. If $P(\mathbf{y}_* \mid \mathbf{x}_*)$ is identifiable via ctf-calculus from the marginalized ground graph $\bar{\mathcal{G}}_\rho$ and $\mathbb{P}$, then it is also relationally identifiable from $\mathcal{G}$ and $\mathbb{P}$.

*Proof.* Assume that the given $P(\mathbf{y}_* \mid \mathbf{x}_*)$ is identifiable via ctf-calculus from the marginalized ground graph $\bar{G}_\rho$ and $\mathbb{P}$. By the soundness of ctf-calculus (Correa & Bareinboim, 2025), this implies that for all (standard) SCMs $\mathcal{N}, \mathcal{N}'$ consistent with $\bar{G}_\rho$ and agreeing on $\mathbb{P}$, $\mathcal{N}, \mathcal{N}'$ also agree on $P(\mathbf{y}_* \mid \mathbf{x}_*)$. We need to show that for any two RSCMs $\mathcal{M}, \mathcal{M}'$ over $\mathcal{S}$ consistent with $\mathcal{G}$, if the ground RSCMs $\mathcal{M}_\rho$ and $\mathcal{M}'_\rho$ agree on $\mathbb{P}$, then they also agree on $P(\mathbf{y}_* \mid \mathbf{x}_*)$. By Lemma D.3, $\mathcal{M}_\rho$ and $\mathcal{M}'_\rho$ induce the marginalized ground graph $\bar{G}_\rho$. Since $\mathcal{M}_\rho$ and $\mathcal{M}'_\rho$ are a subset of the space of standard NCMs over $\mathbf{v}_\rho$, the result follows. $\qquad\square$

**Corollary D.2** (Relational backdoor adjustment). *Consider a schema $\mathcal{S}$, relational causal graph $\mathcal{G}$, skeleton $\rho$, and observational distribution $P(\mathbf{v}_\rho)$. Let $P(\mathbf{y} \mid do(\mathbf{x}))$ be some query with $\mathbf{X}, \mathbf{Y} \subseteq \mathbf{V}_\rho$ and let $\mathbf{Z} \subseteq \mathbf{V}_\rho$ be such that*

1. *$\mathbf{Z}$ contains no descendants of $\mathbf{X}$ in the marginalized ground graph $\bar{\mathcal{G}}_\rho$, and*

2. *$\mathbf{Z}$ blocks every path between $\mathbf{X}$ and $\mathbf{Y}$ that contains an arrow into $\mathbf{X}$ in $\bar{\mathcal{G}}_\rho$.*

*Then, $P(\mathbf{y} \mid do(\mathbf{x}))$ is relationally identifiable from $\mathcal{G}$ and $P(\mathbf{v}_\rho)$ as*

$$P(\mathbf{y} \mid do(\mathbf{x})) = \sum_{\mathbf{z}} P(\mathbf{y} \mid \mathbf{x}, \mathbf{z}) P(\mathbf{z})$$

*Proof.* This follows from Prop. 4.4 and the validity of backdoor adjustment (Bareinboim, 2025). $\qquad\square$

*Proposition* 4.5 (Necessary condition for within-instance relational identification). Consider a schema $\mathcal{S}$, relational causal graph $\mathcal{G}$, source skeletons $\rho_1, \ldots, \rho_l$ with available interventional distributions $\mathbb{P}$, and a target skeleton $\rho_\star$. Let $o \in \rho_\star$ be a target instance and consider a counterfactual query $P(\mathbf{y}_\star \mid \mathbf{x}_\star)$ with $\mathbf{Y}_\star, \mathbf{X}_\star \subseteq \mathbf{V}_o$, the attributes of $o$.

Let the restriction $\mathbb{P}|_O$ be as follows. For each source skeleton $\rho_k$, each distribution $P(\mathbf{v}_{\rho_k} \mid do(\mathbf{x}_{k,j})) \in \mathbb{P}$, and each object $o' \in \rho_k(O)$, include $P(\mathbf{v}_{\rho_k, o'} \mid do(\mathbf{x}_{k,j} \cap \mathbf{v}_{\rho_k, o'}))$ in $\mathbb{P}|_O$, with instance identifiers omitted. Let $\mathcal{G}_o$ be the induced subgraph of the marginalized ground graph $\bar{\mathcal{G}}_{\rho_\star}$ on $\mathbf{V}_o$ with instance identifiers omitted.

If $P(\mathbf{y}_\star \mid \mathbf{x}_\star)$ is non-identifiable via ctf-calculus from $\mathbb{P}|_O$ and $\mathcal{G}_o$, then it is relationally non-identifiable from $\mathcal{G}$ and $\mathbb{P}$.

*Proof.* Fix an instance $o \in \rho_\star(O)$ and a query $P(\mathbf{y}_* \mid \mathbf{x}_*)$ where $\mathbf{Y}_*, \mathbf{X}_* \subseteq \mathbf{V}_{\rho,x}$ as described.

Assume that $P(\mathbf{y}_* \mid \mathbf{x}_*)$ is not identifiable via ctf-calculus from $\mathcal{G}_o$ and the restriction $\mathbb{P}|_O$. Note that here, and in the remainder of the result, we ignore instance identifiers in $\mathbb{P}|_O$ and $\mathcal{G}_o$, simply considering within-instance attributes as in the standard SCM setting.

Then, by the completeness of ctf-calculus (Correa & Bareinboim, 2025), there exist two (standard) SCMs $\mathcal{N}, \mathcal{N}'$ consistent with $\mathcal{G}_o$ that agree on all distributions in $\mathbb{P}|_O$ but disagree on $P(\mathbf{y}_* \mid \mathbf{x}_*)$. Using $\mathcal{N}, \mathcal{N}'$, we will construct two RSCMs $\mathcal{M}, \mathcal{M}'$ that (when grounded) agree on $\mathbb{P}$ but not on $P(\mathbf{y}_* \mid \mathbf{x}_*)$.

Construct $\mathcal{M}$ as follows. Let $\mathcal{M}$ contain the endogenous variables given in $\mathcal{G}$.

1. First, consider attributes of type $O$ in $\mathcal{M}$. For each exogenous variable $U$ in $\mathcal{N}$, let $\mathcal{M}$ contain exogenous variable $O.U$. For attributes $O.A$ belonging to type $O$, let the function determining $O.A$ in $\mathcal{M}$ be the same as that determining $o.A$ in $\mathcal{N}$. In particular, $\mathbf{Pa}_{O.A}^{\mathcal{M}} = \mathbf{Pa}_{o.A}^{\mathcal{N}}$ and $\mathbf{U}_{O.A}^{\mathcal{M}} = \mathbf{U}_{o.A}^{\mathcal{N}}$. $O.A$ has no non-relational parents (endogenous or exogenous) in $\mathcal{M}$.

2. Next, consider attributes of type $Y \neq X$ in $\mathcal{M}$. For each attribute $T.B$, let $\mathcal{M}$ contain an exogenous variable $T.U_B \sim \mathcal{B}(0.5)$. Define the function $f_{T.B} : T.B \leftarrow T.U_B$. In other words, for each type $Y$, each attribute $T.B$ is determined by an independent fair coin flip.

Define $\mathcal{M}'$ similarly but using functions from $\mathcal{N}$. Then, for any skeleton $\rho$, the groundings $\mathcal{M}_\rho, \mathcal{M}'_\rho$ consist of 'copies' of $\mathcal{N}$ and $\mathcal{N}'$ for every instance of type $O$, and coin flips for all other variables. Importantly, in both $\mathcal{M}$ and $\mathcal{M}'$, there are no relational effects; all instances are independent in any grounding.

We need to show that $\mathcal{M}$ and $\mathcal{M}'$ are consistent with $\mathcal{G}$. Since $\mathcal{M}$ and $\mathcal{M}'$ contain no relational effects, it suffices to show that they are consistent with non-relational edges in $\mathcal{G}$. Since attributes of type $Y \neq X$ are all independent by construction,

$\mathcal{M}$ and $\mathcal{M}'$ are consistent with any edges in $\mathcal{G}$ incident to such attributes. For attributes of type $O$, note that for any instances $o', o''$ of type $O$ in any skeleton $\rho$, the induced subgraphs $\mathcal{G}_{o'}$ and $\mathcal{G}_o$ are the same.

Next, we show that $\mathcal{M}$ and $\mathcal{M}'$ agree on $\mathbb{P}$. Consider some distribution $P(\mathbf{v}_{\rho_k} \mid do(\mathbf{x}_{k,j}))$ in $\mathbb{P}$. Then,

$$P^{\mathcal{M}_{\rho_k}}(\mathbf{v}_{\rho_k} \mid do(\mathbf{x}_{k,j}) = \prod_{T \in \mathcal{E} \cup \mathcal{R}} \prod_{t \in \rho_k(T)} p(\mathbf{v}_y \mid do(\mathbf{x}_{k,j}))$$

$$\text{(no relational effects in } \mathcal{M} \implies \text{ all instances independent in } \mathcal{M}_{\rho_k})$$

$$= \prod_{T \in \mathcal{E} \cup \mathcal{R}} \prod_{t \in \rho_k(T)} P^{\mathcal{M}_{\rho_k}}(\mathbf{v}_y \mid do(\mathbf{x}_{k,j} \cap \mathbf{v}_y), do(\mathbf{x}_{k,j} \setminus \mathbf{v}_y))$$

$$= \prod_{T \in \mathcal{E} \cup \mathcal{R}} \prod_{t \in \rho_k(T)} P^{\mathcal{M}_{\rho_k}}(\mathbf{v}_y \mid do(\mathbf{x}_{k,j} \cap \mathbf{v}_y)) \qquad \text{(all instances independent in } \mathcal{M}_{\rho_k})$$

$$= \left( \prod_{T \in \mathcal{E} \cup \mathcal{R} \setminus X} \prod_{t \in \rho_k(T)} P^{\mathcal{M}_{\rho_k}}(\mathbf{v}_y \mid do(\mathbf{x}_{k,j} \cap \mathbf{v}_y)) \right) \cdot \prod_{o \in \rho_k(O)} P^{\mathcal{M}_{\rho_k}}(\mathbf{v}_x \mid do(\mathbf{x}_{k,j} \cap \mathbf{x}_y))$$

$$= \left( \prod_{T \in \mathcal{E} \cup \mathcal{R} \setminus X} \prod_{t \in \rho_k(T)} P^{\mathcal{M}'_{\rho_k}}(\mathbf{v}_y \mid do(\mathbf{x}_{k,j} \cap \mathbf{v}_y)) \right) \cdot \prod_{o \in \rho_k(O)} P^{\mathcal{M}_{\rho_k}}(\mathbf{v}_x \mid do(\mathbf{x}_{k,j} \cap \mathbf{x}_y))$$

$$\text{(variables of all but } X-\text{type objects are coin flips in } \mathcal{M}, \mathcal{M}')$$

$$= \left( \prod_{T \in \mathcal{E} \cup \mathcal{R} \setminus X} \prod_{t \in \rho_k(T)} P^{\mathcal{M}'_{\rho_k}}(\mathbf{v}_y \mid do(\mathbf{x}_{k,j} \cap \mathbf{v}_y)) \right) \cdot \prod_{o \in \rho_k(O)} P^{\mathcal{N}}(\mathbf{v}_x \mid do(\mathbf{x}_{k,j} \cap \mathbf{x}_y))$$

$$\text{(by construction of } \mathcal{M})$$

$$= \left( \prod_{T \in \mathcal{E} \cup \mathcal{R} \setminus X} \prod_{t \in \rho_k(T)} P^{\mathcal{M}'_{\rho_k}}(\mathbf{v}_y \mid do(\mathbf{x}_{k,j} \cap \mathbf{v}_y)) \right) \cdot \prod_{o \in \rho_k(O)} P^{\mathcal{N}'}(\mathbf{v}_x \mid do(\mathbf{x}_{k,j} \cap \mathbf{x}_y))$$

$$\text{(since } \mathcal{N}, \mathcal{N}' \text{ agree on } \mathbb{P}_x)$$

$$= P^{\mathcal{M}'_{\rho_k}}(\mathbf{v}_{\rho_k} \mid do(\mathbf{x}_{k,j}) \qquad \text{(by a symmetric derivation)}$$

Finally, since $\mathcal{N}$ and $\mathcal{N}'$ disagree on $P(\mathbf{y}_* \mid \mathbf{x}_*)$, so do $\mathcal{M}_\rho$ and $\mathcal{M}'_\rho$.

$\square$

## D.3. Proofs for Sec. 5

The proofs of this section resemble that of the expressvity and correctness of NCM training in (Xia et al., 2023), adapted to enforce parameter sharing for variables of the same type and permutation-invariance for relational parents.

**Assumptions.** We assume, that in any domain of interest, the true RSCM $\mathcal{M} = \langle \mathcal{S}, \mathbf{V}, \mathbf{U}, \mathcal{F}, P(\mathbf{U}) \rangle$ is as follows.

(I) $\mathcal{M}$ is $\rho$-Markovian.

(II) All variables in $\mathbf{V}$ have discrete, finite domains.

(III) There is a non-negative integer $D$ such that for any skeleton $\rho$, the grounding $\mathcal{M}_\rho$ has variables $o.A$ with relational parent multisets of size at most $D$.

### D.3.1. DISCRETE RSCMS

First, we introduce *discrete RSCMs*, following (Zhang et al., 2022b).

**Definition D.1** (Discrete RSCM). An RSCM $\mathcal{M} = \langle \mathcal{S}, \mathbf{V}, \mathbf{U}, \mathcal{F}, P(\mathbf{U}) \rangle$ is said to be a discrete RSCM if all variables in $\mathbf{V}$ are discrete and finite, and all variables in $\mathbf{U}$ are discrete.

We will show that the space of discrete RSCMs is equally expressive as that of all RSCMs satisfying assumptions (I)-(III). This licenses the assumption, without loss of generality, that the exogenous variables have discrete and finite domains.

We adapt certain definitions from (Zhang et al., 2022b).

**Definition D.5** (Equivalence classes (Zhang et al., 2022b, Def. A.1)). For an RSCM $\mathcal{M} = \langle \mathcal{S}, \mathbf{V}, \mathbf{U}, \mathcal{F}, P(\mathbf{U}) \rangle$, for every $O.A \in \mathbf{V}$, let functions in $\mathrm{dom}_{\mathbf{Pa}_{O.A}} \times \mathrm{dom}_{\mathbf{Pa}^r_{O.A}} \mapsto \mathrm{dom}(O.A)^7$ be ordered by $\{h^{(i)}_{O.A} \mid i \in I_{O.A}\}$, where $I_{O.A} = \{1, \ldots, m_{O.A}\}$ and $m_{O.A} = |\mathrm{dom}_{\mathbf{Pa}_{O.A}} \times \mathrm{dom}_{\mathbf{Pa}^r_{O.A}} \mapsto \mathrm{dom}(O.A)|$. An *equivalence class* $\mathcal{U}^{(i)}_{O.A}$ for function $h^{(i)}_{O.A}$, $i = 1, \ldots, m_{O.A}$, is a subset of $\mathrm{dom}(\mathbf{U}_{O.A})$ such that

$$\mathcal{U}^{(i)}_{O.A} = \{ \mathbf{u}_{O.A} \in \mathrm{dom}(\mathbf{U}_{O.A}) \mid f_{O.A}(\cdot, \mathbf{u}_{O.A}) = h^{(i)}_{O.A} \}. \tag{26}$$

**Definition D.6** (Canonical Partition (Zhang et al., 2022b, Def. A.2)). For an RSCM $\mathcal{M} = \langle \mathcal{S}, \mathbf{V}, \mathbf{U}, \mathcal{F}, P(\mathbf{U}) \rangle$, $\{\mathcal{U}^{(i)}_{O.A} \mid i \in I_{O.A}\}$ is the *canonical partition* over the exogenous domain $\mathrm{dom}(\mathbf{U}_{O.A})$ for every $O.A \in \mathbf{V}$.

**Corollary D.3.** *For any RSCM $\mathcal{M} = \langle \mathcal{S}, \mathbf{V}, \mathbf{U}, \mathcal{F}, P(\mathbf{U}) \rangle$, for each $O.A \in \mathbf{V}$, the function $f_{O.A} \in \mathcal{F}$ can be decomposed as:*

$$f_{O.A}(\mathbf{pa}_{O.A}, \mathbf{pa}^r_{O.A}, \mathbf{u}_{O.A}) = \sum_{i \in I_{O.A}} h^{(i)}_{O.A}(\mathbf{pa}_{O.A}, \mathbf{pa}^r_{O.A})\mathbf{1}_{\mathbf{u}_{O.A} \in \mathcal{U}^{(i)}_{O.A}}. \tag{27}$$

*Proof.* This is an immediate consequence of Lemma A.3 in (Zhang et al., 2022b), considering the union of relational and non-relational parents. □

**Corollary D.4.** *Consider an RSCM $\mathcal{M} = \langle \mathcal{S}, \mathbf{V}, \mathbf{U}, \mathcal{F}, P(\mathbf{U}) \rangle$. For any relational skeleton $\rho$, let the ground RSCM be $\mathcal{M}_\rho = \langle \mathbf{V}_\rho, \mathbf{U}_\rho, \mathcal{F}_\rho, P(\mathbf{U})_\rho \rangle$. Let the indexing set $\mathbf{I}$ be $\mathbf{I} = \bigtimes_{O.A \in \mathbf{V}} \bigtimes_{o \in \rho(O)} I_{O.A}$, i.e., a product of indexing sets across the different types of variables, and for instances of each type. Then, for any variables $\mathbf{Y}, \ldots, \mathbf{Z}, \mathbf{X}, \ldots, \mathbf{W} \subseteq \mathbf{V}$,*

$$P^{\mathcal{M}_\rho}(\mathbf{y_x}, \ldots \mathbf{z_w})$$

$$= \sum_{\mathbf{i} \in \mathbf{I}} \mathbf{1}_{\mathbf{Y_x(i)} = \mathbf{y}, \ldots, \mathbf{Z_w(i)} = \mathbf{z}} \cdot P \left( \bigcap_{O.A \in \mathbf{V}, o \in \rho(O)} \mathcal{U}^{(i)}_{O.A} \right) \quad \text{(where } i \text{ is the index for } o.A \text{ in } i)$$

*where $\mathbf{Y_x(i)} = \{Y_{\mathbf{x}}(\mathbf{i}) = y \mid o.B \in \mathbf{Y}\}$ and each $o.B_{\mathbf{x}}(\mathbf{i})$ is recursively computed as*

$$o.B_{\mathbf{x}}(\mathbf{i}) = \begin{cases} \mathbf{x}_{o.B} & \text{if } o.B \in \mathbf{X} \\ h^{(i)}_{O.B}((\mathbf{Pa}_{o.B})_{\mathbf{x}}(\mathbf{i})) & \text{otherwise} \end{cases}$$

*Proof.* This follows from (Zhang et al., 2022b, Lemma A.4), noting that each instance $o.A$ is determined by the same function $f_{O.A}$ of its parents in $\mathcal{M}_\rho$. □

**Corollary D.5.** *Consider an RSCM $\mathcal{M} = \langle \mathcal{S}, \mathbf{V}, \mathbf{U}, \mathcal{F}, P(\mathbf{U}) \rangle$ inducing relational causal graph $\mathcal{G}$. For any relational skeleton $\rho$, let the indexing set be $\mathbf{I}$ as in Cor. D.4. For a given type $X \in \mathcal{E} \cup \mathcal{R}$, $\mathcal{C}(\mathcal{G})_X = \{\mathbf{C} \subseteq \mathbf{V} \mid C$ is a bidirected connected component in $\mathcal{G}$ containing some $O.A\}$. Then, for the ground RSCM $\mathcal{M}_\rho$,*

$$P \left( \bigcap_{O.A \in \mathbf{V}, o \in \rho(O)} \mathcal{U}^{(i)}_{O.A} \right) = \prod_{X \in \mathcal{E} \cup \mathcal{R}} \prod_{o \in \rho(O)} \prod_{\mathbf{C} \in \mathcal{C}(\mathcal{G})_X} P( \bigcap_{O.A \in \mathbf{C}} \mathcal{U}^i_{O.A}).$$

*Proof.* This follows by a similar argument as in (Zhang et al., 2022b, Lemma A.5), noting that since $\mathcal{M}$ is $\rho$-Markovian, exogenous variables are independent across different instances $o$, and identically distributed across different instances $o$ of the same type $O$. □

**Corollary D.6** (Discrete RSCM expressiveness). *Consider an RSCM $\mathcal{M} = \langle \mathcal{S}, \mathbf{V}, \mathbf{U}, \mathcal{F}, P(\mathbf{U}) \rangle$ satisfying assumptions (I)-(III) and inducing relational causal graph $\mathcal{G}$. Then, there exists a discrete RSCM $\mathcal{M}'$ consistent with $\mathcal{G}$ such that for any skeleton $\rho$, the ground RSCMs $\mathcal{M}_\rho$ and $\mathcal{M}'_\rho$ agree on all counterfactual distributions over $\mathbf{V}_\rho$.*

---

[7]Here, $\mathrm{dom}_{\mathbf{Pa}^r_{O.A}}$ contains multisets of size $\leq D$ for each relational parent, per assumption (II).

*Proof.* This follows from (Zhang et al., 2022b, Lemma A.6), Cor. D.5, and Cor. D.4.

$\square$

### D.3.2. Neural RSCMs

*Theorem* 5.2 (Expressivity of RNCMs). Consider a relational schema $\mathcal{S}$. For every RSCM $\mathcal{M}$ over $\mathcal{S}$ inducing relational causal graph $\mathcal{G}$, there exists a $\mathcal{G}$-RNCM $\mathcal{N}$ such that for every skeleton $\rho$, the ground RSCMs $\mathcal{M}_\rho$ and $\mathcal{N}_\rho$ induce the same counterfactual distributions over $\mathbf{V}_\rho$.

*Proof.* Fix an RSCM $\mathcal{M}^*$ over $\mathcal{S}$ inducing relational causal graph $\mathcal{G}$. By the discrete RSCM expressiveness corollary above, there exists a discrete RSCM $\mathcal{M}' = \langle \mathcal{S}, \mathbf{V}, \mathbf{U}', \mathcal{F}', P(\mathbf{U}') \rangle$ consistent with $\mathcal{G}$ such that for every skeleton $\rho$, the ground rscms $\mathcal{M}_\rho^*$ and $\mathcal{M}'_\rho$ agree on all counterfactual distributions over $\mathbf{V}_\rho$. Thus, it suffices to construct a $\mathcal{G}$-RNCM $\mathcal{N}$ that agrees with $\mathcal{M}'_\rho$ on all counterfactuals, for every $\rho$.

We follow the proof strategy of (Xia et al., 2023): (i) represent the discrete exogenous distribution using uniform $\mathcal{U}([0,1])$ noise via a neural inverse probability integral transform [Lemma 5](Xia et al., 2021), and (ii) represent each mechanism $f_{O.A}$ using a multi-layer perceptron (MLP).

Since $\mathcal{M}'$ is $\rho$-Markovian (Assumption (I)), bidirected edges occur only among attributes of the same instance $o$. Fix a type $X \in \mathcal{E} \cup \mathcal{R}$, and let $\mathcal{C}(\mathcal{G})_X = \{\mathbf{C}_1, \ldots, \mathbf{C}_k\}$ be the set of bidirected maximal cliques among $\{O.A : O.A \in \mathbf{V}\}$ in $\mathcal{G}$. We construct $\mathcal{N}$ as in Def. 5.1 by introducing, for each $\mathbf{C} \in \mathcal{C}(\mathcal{G})_X$, an exogenous variable $\hat{U}_\mathbf{C} \sim \mathcal{U}([0,1])$. For each clique $\mathbf{C}$, let $\mathbf{U}'_\mathbf{C}$ denote the tuple of discrete exogenous variables in $\mathcal{M}'$ that are the shared exogenous parents of the variables in $\mathbf{C}$ (within an instance of type $O$). Since $\mathbf{U}'_\mathbf{C}$ has a finite domain, Lemma 5 of (Xia et al., 2021) provides an MLP $g_\mathbf{C}$ such that $g_\mathbf{C}(\hat{U}_\mathbf{C})$ has the same distribution as $\mathbf{U}'_\mathbf{C}$.

Next, consider some variable $O.A \in \mathbf{V}$. Under Assumptions (II)–(III), the input space $\text{dom}(\mathbf{Pa}_{O.A}) \times \text{dom}(\mathbf{Pa}^r_{O.A}) \times \text{dom}(\mathbf{U}'_{O.A})$ is finite (the relational-parent domain is finite because each multiset has size at most $D$). Hence, by Lemma 4 of (Xia et al., 2021), there exists an MLP $h_{O.A}$ that agrees with $f'_{O.A}$ for each possible input.

Composing MLPs $h_{O.A}$ with every MLP $g_\mathbf{C}$ such that $O.A \in \mathbf{C}$ yields a function $\hat{f}_{O.A} : \text{dom}(\mathbf{Pa}_{O.A}) \times \text{dom}(\mathbf{Pa}^r_{O.A}) \times \text{dom}(\hat{\mathbf{U}}_{O.A}) \to \text{dom}(O.A)$.

For any skeleton $\rho$, the ground RSCM $\mathcal{N}_\rho$ shares the same function $\hat{f}_{O.A}$ across all instances $o \in \rho(O)$; this is also the case in $\mathcal{M}_\rho$. Therefore, for every ground variable $o.A \in \mathbf{V}_\rho$, $\mathcal{M}_\rho$ and $\mathcal{N}_\rho$ share the same distribution over exogenous parents and the same mechanism. It follows that $\mathcal{M}'_\rho$ and $\mathcal{M}^*_\rho$ agree on all counterfactual distributions. $\square$

**Definition D.2** (Data-dependent relational counterfactual identification). Consider a schema $\mathcal{S}$, relational causal graph $\mathcal{G}$, source skeletons $\rho_1, \ldots, \rho_l$, source distributions $\mathbb{P} = \{\{P(\mathbf{v}_{\rho_k} \mid do(\mathbf{x}_{k,j})) > 0\}_{j=1}^{m_k}\}_{k=1}^l$, and target skeleton $\rho_\star$. Let $P(\mathbf{y}_* \mid \mathbf{x}_*)$ be a target query with $\mathbf{Y}_\star, \mathbf{X}_\star \subseteq \mathbf{V}_{\rho_\star}$.

We say $P(\mathbf{y}_\star \mid \mathbf{x}_\star)$ is *relationally identifiable* from $\mathcal{G}$ and $\mathbb{P}$ if for any RSCMs $\mathcal{M}, \mathcal{M}'$ consistent with $\mathcal{G}$ agreeing on the source data, so that for every $\rho_k$ and $j = 1, \ldots, m_k$,

$$P^{\mathcal{M}_{\rho_k}}(\mathbf{v}_{\rho_k} \mid do(\mathbf{x}_{k,j})) = P^{\mathcal{M}'_{\rho_k}}(\mathbf{v}_{\rho_k} \mid do(\mathbf{x}_{k,j})) = P(\mathbf{v}_{\rho_k} \mid do(\mathbf{x}_{k,j}))$$

they also agree on the query:

$$P^{\mathcal{M}_{\rho_\star}}(\mathbf{y}_\star \mid \mathbf{x}_\star) = P^{\mathcal{M}'_{\rho_\star}}(\mathbf{y}_\star \mid \mathbf{x}_\star).$$

Otherwise, the query is *relationally non-identifiable dependent on the data*.

**Corollary D.7** (Correctness of RelationalNeuralID (Alg. 1)). *Consider a schema $\mathcal{S}$, relational causal graph $\mathcal{G}$, source skeletons $\rho_1, \ldots, \rho_l$, source distributions $\mathbb{P} = \{\{P(\mathbf{v}_{\rho_k} \mid do(\mathbf{x}_{k,j})) > 0\}_{j=1}^{m_k}\}_{k=1}^l$, and target skeleton $\rho_\star$. Let $P(\mathbf{y}_* \mid \mathbf{x}_*)$ be a target query with $\mathbf{Y}_\star, \mathbf{X}_\star \subseteq \mathbf{V}_{\rho_\star}$. Let $Q$ be the output of RelationalNeuralID (Alg. 1) given these inputs. Then, $Q$ is not FAIL if and only if the query $P(\mathbf{y}_* \mid \mathbf{x}_*)$ is relationally identifiable dependent on the data from $\mathcal{G}$ and $\mathbb{P}$, in which case $Q = P(\mathbf{y}_* \mid \mathbf{x}_*)$.*

*Proof.* Fix a relational causal diagram $\mathcal{G}$. By Thm. 5.2, for every RSCM $\mathcal{M}$ consistent with $\mathcal{G}$, there exists a $\mathcal{G}$-RNCM agreeing with $\mathcal{M}$ on counterfactual distributions for every possible skeleton $\rho$. Additionally, every $\mathcal{G}$-RNCM is consistent

---

**Algorithm 2** RelationalNeuralEstimation

---

**Input:** schema $\mathcal{S}$, relational causal graph $\mathcal{G}$, source data $\mathcal{D} = \left\{ \left( \rho_k, \{ P(\mathbf{v}_{\rho_k} \mid do(\mathbf{x}_{k,j})) \}_{j=1}^{m_k} \right) \right\}_{k=1}^{l}$, target skeleton $\rho_\star$,
query $P(\mathbf{y}_* \mid \mathbf{x}_*)$
$\hat{M} \leftarrow \mathcal{G}$-RNCM
$\hat{\theta} \leftarrow \theta \in \Theta(\hat{M})$ subject to $\forall k, j \; P^{\hat{M}_{\rho_k}(\theta)}(\mathbf{v}_{\rho_k} \mid do(\mathbf{x}_{k,j})) = P(\mathbf{v}_{\rho_k} \mid do(\mathbf{x}_{k,j}))$
$q \leftarrow P^{\hat{M}_{\rho_\star}(\hat{\theta}_l)}(\mathbf{y}_* \mid \mathbf{x}_*)$
**return** $q$

---

with $\mathcal{G}$ by construction. Therefore, minimizing / maximizing the query $Q$ in the space of RSCMs consistent with $\mathcal{G}$ and inducing the given $\mathbb{P}$ is equivalent to minimizing / maximizing the query $Q$ in the space of $\mathcal{G}-$ RSCMs inducing the given $\mathbb{P}$. The correctness of Alg. 1 thus follows from the definition of data-dependent relational identification (Def. D.2).

$\square$

### D.4. Additional Algorithms

**Estimating identifiable queries.**  Alg. 2 provides an algorithm for point-estimation of a given query from graph and data. It modifies Alg. 1 by simply fitting a single $\mathcal{G}$-RNCM to the available data, and outputting the value of the query for that RNCM. Since Alg. 2 does not minimize/maximize the query, it is correct only assuming that the given query is relationally identifiable from the given graph and distributions.

## E. Further Experiments and Experimental Details

### E.1. Compute and Implementation

All implementations are in Python, adapted from the Neural Causal Models codebase (Xia et al., 2023). Neural Causal Models are built using PyTorch (Paszke et al., 2019) and trained using PyTorch Lightning (Falcon & Cho, 2020). We trained our models on a single NVIDIA H100 GPU on a shared compute cluster with 2x Intel Xeon Platinum 8480+ CPUs (112 cores total, 224 threads) at up to 3.8 GHz, and 210 MiB L3 cache

### E.2. Data Generation

We generated all synthetic data using *regional canonical models* (Xia et al., 2023, Def. 11) adapted to share functions across variables of the same type. Concretely, given a $\rho$-Markovian relational causal graph $\mathcal{G}$, we introduce one exogenous variable $U_{\mathbf{C}} \sim \mathcal{U}([0, 1])$ for each maximal bidirected clique $\mathbf{C}$ in $\mathcal{G}$, to represent within-instance latent confounding encoded by $\mathcal{G}$. For each endogenous variable $O.A$, we then sample a random deterministic function

$$f_{O.A} : \text{dom}(\mathbf{Pa}_{O.A}) \times \text{dom}(\mathbf{Pa}_{O.A}^r) \times \text{dom}(\mathbf{u}_{O.A}) \rightarrow \text{dom}(O.A)$$

according to the canonical construction of Xia et al. (2023), where $\mathbf{u}_{O.A} = \{U_{\mathbf{C}} : O.A \in \mathbf{C}\}$.

For each relational parent $(\mathbf{W}, \phi, \text{AGG}) \in \mathbf{Pa}_{O.A}^r$, we set its domain to be the set of histograms over $\text{dom}(\mathbf{W})$ induced by multisets of size at most $D$; in our experiments $D = 5$, which upper-bounds the relational-parent multiset sizes in Fig. 1. Note that since we assume discrete endogenous domains, the count is a sufficient statistic for a multiset of $\mathbf{w}$-values.

For each trial, we sample a random regional canonical model as above. Given a skeleton $\rho$, to generate data from $P(\mathbf{v}_\rho)$, we instantiate variables $o.A$ and $o.U_{\mathbf{C}}$ for every instance $o$, and reuse the same mechanism for every variable of the same type. Due to the expressivity of canonical models (Zhang et al., 2022b; Xia et al., 2023), this avoids bias in data-generation induced by choosing a particular parametric model.

### E.3. Model Architecture and Training

We refer readers to the comprehensive Appendix B.6 in (Xia et al., 2023) for details on the MLE-NCM architecture and training procedure. In addition to the clique-level noise variables (Def. 5.1), an MLE NCM additionally includes a Gumbel noise variable for each attribute.

Given a relational causal graph $\mathcal{G}$, we initialize a module for each mechanism $f_{O.A}$ to get an RNCM $\hat{\mathcal{M}}$. This is reused for

every instance $o.A$ across source and target skeletons. To train to maximize a query $P(\mathbf{y}_* \mid \mathbf{x}_*)$ on a target $v_{\rho_*}$ given data $\mathcal{D}$ from sources $\rho_1, \ldots, \rho_k$, where each source $\rho_k$ has data $\{\mathbf{v}_{\rho_\mathbf{k}, \mathbf{z_{j,k}}}{}^{(i)}\}_{i=1}^{n_{j,k}}$ comprising $n_{j,k}$ datapoints from $j = 1, \ldots, m_k$ regimes, we use the following modified MLE-NCM loss.

$$L(\hat{\mathcal{M}}, \mathcal{D}) = \sum_{k=1}^{l} \sum_{j=1}^{m_k} \frac{1}{n_{j,k}} \sum_{i=1}^{n_{j,k}} -\log P^{\hat{\mathcal{M}}_{\rho_k}}(\mathbf{v}_{\rho_k, \mathbf{z}_{j,k}}^{(i)}) - \lambda \log P^{\hat{\mathcal{M}}_{\rho_\star}}(\mathbf{y} \mid \mathbf{x}_*) \tag{3}$$

Here, $\lambda$ is a parameter that decreases during training. To minimize the query, we replace the $\lambda \log P^{\hat{\mathcal{M}}_{\rho_\star}}(\mathbf{y} \mid \mathbf{x}_*)$ term with $\lambda(1 - \log P^{\hat{\mathcal{M}}_{\rho_\star}}(\mathbf{y} \mid \mathbf{x}_*))$. In our estimation experiments (Exp. 6.1), we set $\lambda = 0$ and train only one RNCM.

All modules contain 2 hidden layers with 128 neurons each, and were trained for 200 epochs (often converging earlier) with a learning rate of $10^{-3}$ and a batch size of 1000 datapoints.

### E.4. RNCM Estimation Experiments: Details

**Specification of queries.** In Table E.4.1, we give the the exact target queries used in Exp. 6.1.

**Proof of identifiability.** We give an example derivation to show that $P^{\rho_C}(c_2.b \mid do(p_2.x, p_3.x))$ is identifiable from $P(\mathbf{v}_{\rho_A})$. The proof for other identifiable cases follows similarly, by application of same-skeleton backdoor adjustment (Cor. D.2) and cross-skeleton observational inference (Thm. 4.3).

**Proposition E.1.** *Given source skeleton $\rho_A$ and target skeleton $\rho_C$ as in Exp. 6.1, and relational causal graph $\mathcal{G}$ (Fig. 2), the distribution $P^{\rho_C}(c_2.b \mid do(p_2.x, p_2.x))$ is relationally identifiable from $P(\mathbf{v}_{\rho_A})$ and $\mathcal{G}$.*

*Proof.* By Cor. D.2, $P^{\rho_C}(c_2.b \mid do(p_2.x, p_2.x)))$ is identifiable from $P(\mathbf{v}_{\rho_C})$ via backdoor adjustment on the set $\{s_2.W\}$, using the formula

$$P^{\rho_C}(c_2.b \mid do(p_2.x, p_2.x)) = \sum_{s_2.w} P^{\rho_C}(c_2.b \mid s_2.w, p_2.x, p_3.x) P^{\rho_C}(s_2.W)$$

By Thm. 4.3, we have that $P^{\rho_C}(c_2.b \mid do(p_2.x, p_2.x)) = P^{\rho_A}(c_1.b \mid s_1.w, p_1.x, p_2.x)$ and $P^{\rho_C}(s_2.W) = P^{\rho_A}(s_1.W)$, and we are done. $\square$

**Proof of non-identifiability.** We show that in the case where RNCMs underperform relative to the gold standard NCM* (training on source $\rho_A$ and evaluating on car $c_1$ in target $\rho_C$), the query is in fact non-identifiable from the available data and assumptions.

**Proposition E.2.** *Given source skeleton $\rho_A$ and target skeleton $\rho_C$ as in Exp. 6.1, and relational causal graph $\mathcal{G}$ (Fig. 2), the distribution $P^{\rho_C}(c_1.B = 1 \mid do(p_1.X = 1, p_2.X = 1))$ is relationally non-identifiable from $P(\mathbf{v}_{\rho_A})$ and $\mathcal{G}$.*

*Proof.* We construct two RSCMs inducing $\mathcal{G}$ that agree on the observational distribution over the source skeleton $\rho_A$, but disagree on the target interventional query.

Consider an RSCM $\mathcal{M}$ as follows. The endogenous variables are $\mathbf{V} = \{\mathsf{Sig}.W, \mathsf{Ped}.X, \mathsf{Car}.B\}$. The exogenous variables $\mathbf{U}$ are $\mathsf{Sig}.U_W \sim \mathcal{B}(0.3), \mathsf{Ped}.U_X \sim \mathcal{B}(0.4)$ and $\mathsf{Car}.U_B \sim \mathcal{B}(0.2)$, mutually independent. The mechanisms are

$$\mathsf{Sig}.W \leftarrow \mathsf{Sig}.U_W,$$
$$\mathsf{Ped}.X \leftarrow \mathsf{Ped}.U_X \oplus \mathbf{1}[|\{\mathsf{Sig} \mid \mathsf{Sig}.W = 1, \mathsf{Ctrl}(\mathsf{Sig}, \mathsf{Ped})\}| > 0],$$
$$\mathsf{Car}.B \leftarrow \mathsf{Car}.U_B \oplus H(\mathsf{Car})$$

where

$$H(\mathsf{Car}) = \mathbf{1}[|\{\mathsf{Sig} \mid \mathsf{Sig}.W = 1, \mathsf{Ctrl}(\mathsf{Sig}, \mathsf{Car})\}| > 0] \vee \mathbf{1}[|\{\mathsf{Ped} \mid \mathsf{Ped}.X = 1, \mathsf{Path}(\mathsf{Ped}, \mathsf{Car})\}| > 0].$$

Next, consider an RSCM $\mathcal{M}'$, identical to $\mathcal{M}$ except with a modified braking mechanism

$$\mathsf{Car}.B \leftarrow \mathsf{Car}.U_B \oplus H(\mathsf{Car}) \oplus Z(\mathsf{Car})$$

where

$$Z(\mathsf{Car}) = \mathbf{1}[|\{\mathsf{Sig} \mid \mathsf{Sig}.W = 1, \mathsf{Ctrl}(\mathsf{Sig}, \mathsf{Car})\}| > 1].$$

Both $\mathcal{M}$ and $\mathcal{M}'$ induce the same relational causal graph $\mathcal{G}$, since the additional term $Z(\mathsf{Car})$ depends only on signal variables already related to $\mathsf{Car}.B$.

We first show that $\mathcal{M}$ and $\mathcal{M}'$ agree on the source skeleton $\rho_A$. In $\rho_A$, every car is controlled by at most one signal. Thus, for every car $c$ in $\rho_A$, $Z(c) = 0$. Therefore the braking mechanisms of $\mathcal{M}$ and $\mathcal{M}'$ coincide on $\rho_A$. Since all other mechanisms are identical by construction, $P^{\mathcal{M}_{\rho_A}}(\mathbf{v}_{\rho_A}) = P^{\mathcal{M}'_{\rho_A}}(\mathbf{v}_{\rho_A})$.

We now show that the two models disagree on the target query. In $\rho_C$, car $c_1$ is controlled by two signals, $s_1$ and $s_2$, and has $p_1, p_2$ in its path. Under the intervention $do(p_1.X = 1, p_2.X = 1)$, we have $H(c_1) = 1$ in both models, regardless of the signal values. Therefore,

$$\begin{aligned}
P^{\mathcal{M}_{\rho_C}}&(c_1.B = 1 \mid do(p_1.X = 1, p_2.X = 1)) \\
&= P^{\mathcal{M}_{\rho_C}}(c_1.U_B \oplus 1 = 1) \\
&= P^{\mathcal{M}_{\rho_C}}(c_1.U_B = 0) = 0.8.
\end{aligned}$$

In $\mathcal{M}'$, the additional term $Z(c_1)$ equals 1 exactly when both controlling signals are active, i.e. when $s_1.W = s_2.W = 1$. Since the signal mechanisms are unchanged by the intervention and $P(s.W = 1) = 0.3$, we have $P^{\rho_C}(Z(c_1) = 1) = P(s_1.W = 1, s_2.W = 1) = 0.3^2 = 0.09$. Thus,

$$\begin{aligned}
P^{\mathcal{M}'_{\rho_C}}&(c_1.B = 1 \mid do(p_1.X = 1, p_2.X = 1)) \\
&= P^{\mathcal{M}'_{\rho_C}}(c_1.U_B \oplus 1 \oplus Z(c_1) = 1) \\
&= P(Z(c_1) = 0)P(c_1.U_B = 0) + P(Z(c_1) = 1)P(c_1.U_B = 1) \\
&= (1 - 0.09)(0.8) + (0.09)(0.2) \\
&= 0.746.
\end{aligned}$$

Thus $\mathcal{M}$ and $\mathcal{M}'$ agree on $P(\mathbf{v}_{\rho_A})$ while inducing different values for the target query. $\qquad\square$

**Design of baselines.** We compare RNCMs with five baselines, constructed as follows.

1. **NCM-X, a non-relational causal baseline.** This trains a $\mathcal{G}$-NCM for the flattened graph $\mathcal{G}$ with edges $W \to X$, $W \to B$, $X \to B$. It is trained on 'Cartesian' flattened data, splitting each data point into all triples of cars, pedestrians, and signals. For example, a datapoint of source skeleton $\rho_A$ is split into four datapoints as follows:

$$\begin{aligned}
(s_1.w, p_1.x, p_2.x, c_1.b, c_2.b) \to \;&(s_1.w, p_1.x, c_1.b) \\
&(s_1.w, p_1.x, c_2.b) \\
&(s_1.w, p_2.x, c_1.b) \\
&(s_1.w, p_2.x, c_2.b).
\end{aligned}$$

If the original dataset for source $\rho_A$ contains $10^4$ datapoints, the flattened dataset thus contains $4 \times 10^4$ datapoints.

2. **NCM-J, a causal baseline with partial relational information.** It is trained on 'join' flattened data, splitting each data point into all triples of cars, pedestrians, and signals that are pairwise related. For example, a datapoint of source skeleton $\rho_A$ is split into two datapoints as follows:

$$\begin{aligned}
(s_1.w, p_1.x, p_2.x, c_1.b, c_2.b) \to \;&(s_1.w, p_1.x, c_1.b) \\
&(s_1.w, p_2.x, c_1.b).
\end{aligned}$$

| Car in $\rho_C$ | Neighbourhood | Query |
|---|---|---|
| $c_1$ | 2 signals, 2 pedestrians | $P(c_1.B = 1 \mid do(p_1.X = 1, p_2.X = 1))$ |
| $c_2$ | 1 signal, 2 pedestrians | $P(c_2.B = 1 \mid do(p_2.X = 1, p_3.X = 1))$ |
| $c_3$ | 2 signals, 2 pedestrians | $P(c_2.B = 1 \mid do(p_3.X = 1))$ |

*Table E.4.1.* Target queries used Exp. 6.1 and Fig. 3.

Note how $(s_1.w, p_1.x, c_2.b)$ is not an included triple since $\mathsf{Ctrl}(s_1, c_2)$ is false. If the original dataset for source $\rho_A$ contains $10^4$ datapoints, the flattened dataset thus contains $2 \times 10^4$ datapoints.

3. **REL-MLP, a relational non-causal baseline.** This baseline trains an MLP to predict $c.B$ from count features of the objects related to $c$. Each relational datapoint is split into one supervised example per car. The feature vector for car $c$ is

$$\Big( \sum_{s:\mathsf{Ctrl}(s,c)} s.W, \ |\{s : \mathsf{Ctrl}(s,c)\}|, \ \sum_{p:\mathsf{InPath}(p,c)} p.X, \ |\{p : \mathsf{InPath}(p,c)\}| \Big).$$

Thus the model observes both the number of related signals and pedestrians, and the number of active related signals and pedestrians. For example, a datapoint from $\rho_A$ is split into two car-level examples:

$$(s_1.w, p_1.x, p_2.x, c_1.b, c_2.b) \mapsto (s_1.w, \ 1, \ p_1.x + p_2.x, \ 2, \ c_1.b),$$
$$(0, \ 0, \ p_2.x, \ 1, \ c_2.b).$$

Here the first four entries are the input features and the final entry is the prediction target. Since this baseline is not causal, it cannot directly compute interventional quantities. To estimate our target query $P(c.B = 1 \mid do(p_1.X = 1, \ldots, p_k.X = 1))$, we use a conditional probability. rst, the intervention is mapped to the active pedestrian count it induces in the target car's neighborhood: $m_c = \sum_{j=1}^{k} \mathbf{1}\{\mathsf{InPath}(p_j, c)\} x_j$. We then average the MLP predictions over the subset of rows $\mathcal{I}$ in the source data whose active pedestrian count is $m_c$:

$$\widehat{P}(c.B = 1 \mid do(p_1.X = 1, \ldots, p_k.X = 1)) = \frac{1}{|\mathcal{I}|} \sum_{i \in \mathcal{I}} \sigma(f_\theta(i)),$$

If $\mathcal{I}$ is empty, the estimate is left undefined.

4. **REL-MLP + DEG, a degree-matched variant of REL-MLP.** This baseline uses the same trained MLP and the same count features as REL-MLP, but uses a different query estimator. In addition to matching the active pedestrian count induced by the intervention, it also requires the source car row to have the same number of related signals and pedestrians as the target car, thus choosing a smaller subset $\mathcal{I}$ above. For example, suppose the target query concerns a car $c$ with one related signal and two related pedestrians, under an intervention setting two related pedestrians to $1$. Then REL-MLP averages MLP predictions over all source rows with two active related pedestrians. REL-MLP + DEG further restricts this average to source rows whose cars also have exactly one related signal and two related pedestrians.

5. **NCM\*, a gold standard target-only NCM..** Finally, for a target skeleton $\rho_\star$, NCM\* is simply a $\bar{\mathcal{G}}_{\rho_\star}$-constrained NCM trained on data from the target, where $\bar{\mathcal{G}}_{\rho_\star}$ is the marginalized ground graph of the true relational causal graph on $\rho_\star$. Note that since it is a standard NCM, it does not share parameters across different objects of the same type.

## E.5. Experiment with Front-door Estimation

In this section, we present an additional experiment with a front-door graph, in addition to the backdoor graphs considered in Exp. 6.1. We use similar set of source and target relational skeletons as in Exp. 6.2. We train RNCMs using the graph in Fig. E.5.1 to estimate the identifiable same-car query $P(c_1.B = 1 \mid do(c_1.X = 1))$ in the target skeleton. This query is computable via front-door adjustment on $\{c_1.Z, c_2.Z, c_3.Z\}$. Averaging across five trials, RNCMs achieve an MSE of $0.00082 \pm 0.0018$ from the ground truth, thus estimating the query with high accuracy.

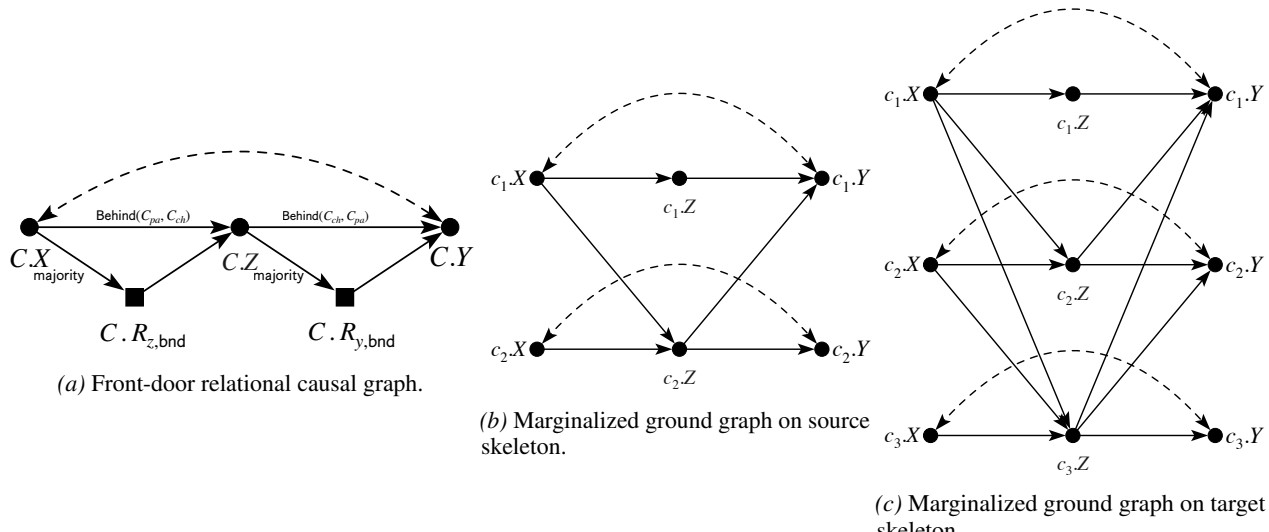

*(a)* Front-door relational causal graph.

*(b)* Marginalized ground graph on source skeleton.

*(c)* Marginalized ground graph on target skeleton.

*Figure E.5.1.* Front-door graphs used in Exp. E.5

### E.6. RNCM Identification Experiments: Details

In Fig. E.6.1, we show the marginalized ground causal graphs for the source skeleton $\rho$ and target skeleton $\rho_\star$ of both the relational bow and the relational IV graphs used in Exp. 6.2.

We also prove the identifiability result for the different-car query in both cases.

**Proposition E.3.** *Given source skeleton $\rho$ and target skeleton $\rho_\star$ as in Exp. 6.2, and causal graph $\mathcal{G}_{\text{bow}}$ (Fig. E.6.1, top left), the distribution $P^{\rho_\star}(c_3.y)$ is relationally identifiable from $P(\mathbf{v}_\rho)$ and $\mathcal{G}_{\text{bow}}$.*

*Proof.* Consider any RSCM $\mathcal{M}$ consistent with $\mathcal{G}_{\text{bow}}$. Since $\mathcal{G}_{\text{bow}}$ is $\rho$-Markovian, so is $\mathcal{M}$. Let $\mathbf{U}_X$ (resp., $\mathbf{U}_Y$) denote the non-relational exogenous variables affecting $\text{Car}.X$ (resp., $\text{Car}.Y$), and $\mathbf{U}'_X = \mathbf{U}'_X \setminus \mathbf{U}'_Y$ Then, in the ground RSCM $\mathcal{M}_{\rho_\star}$,

$$
P^{\mathcal{M}_{\rho_\star}}(c_3.y) = \sum_{c_3.x, c_3.\mathbf{u}_Y}^{\mathcal{M}_{\rho_\star}} P^{\mathcal{M}_{\rho_\star}}(c_3.y \mid c_3.x, c_3.\mathbf{u}_Y) P^{\mathcal{M}_{\rho_\star}}(c_3.x \mid c_3.\mathbf{u}_Y) P^{\mathcal{M}_{\rho_\star}}(c_3.\mathbf{u}_Y)
$$

$$
= \sum_{c_3.x, c_3.\mathbf{u}_Y} \mathbf{1}_{f_{\text{Car}.Y}(c_3.x, \mathbf{u}_Y) = c_3.y} \sum_{c_3.\mathbf{u}'_X} \mathbf{1}_{f_{\text{Car}.X}(\mathbf{u}_X) = c_3.x} P^{\mathcal{M}_{\rho_\star}}(c_3.\mathbf{u}_Y, c_3.\mathbf{u}'_X)
$$

$$
\text{($c_3.Y$ has no relational parents)}
$$

$$
= \sum_{c_3.x, c_3.\mathbf{u}_Y} \mathbf{1}_{f_{\text{Car}.Y}(c_3.x, \mathbf{u}_Y) = c_3.y} \sum_{c_3.\mathbf{u}'_X} \mathbf{1}_{f_{\text{Car}.X}(\mathbf{u}_X) = c_3.x} P^{\mathcal{M}_\rho}(c_2.\mathbf{u}_Y, c_2.\mathbf{u}'_X)
$$

$$
\text{(exogenous distributions are shared across $c$ in $\rho, \rho_\star$)}
$$

$$
= \sum_{c_2.x, c_2.\mathbf{u}_Y} \mathbf{1}_{f_{\text{Car}.Y}(c_2.x, \mathbf{u}_Y) = c_2.y} \sum_{c_2.\mathbf{u}'_X} \mathbf{1}_{f_{\text{Car}.X}(\mathbf{u}_X) = c_2.x} P^{\mathcal{M}_\rho}(c_2.\mathbf{u}_Y, c_2.\mathbf{u}'_X)
$$

$$
\text{(functions $f_{\text{Car}.X}, f_{\text{Car}.Y}$ are shared across $c$ in $\rho, \rho_\star$)}
$$

$$
= P^{\mathcal{M}_\rho}(c_2.y) \qquad \text{(by a symmetric derivation)}
$$

This proves that for any RSCMs $\mathcal{M}, \mathcal{M}'$ consistent with $\mathcal{G}_{\text{bow}}$, we have $P^{\mathcal{M}_\rho}(c_2.y) = P^{\mathcal{M}_{\rho_\star}}(c_3.y)$ and $P^{\mathcal{M}'_\rho}(c_2.y) = P^{\mathcal{M}'_{\rho_\star}}(c_3.y)$. If $\mathcal{M}, \mathcal{M}'$ agree on the source data, then $P^{\mathcal{M}_\rho}(c_2.y) = P^{\mathcal{M}'_\rho}(c_2.y)$, and we are done. $\qquad \square$

**Proposition E.4.** *Given source skeleton $\rho$ and target skeleton $\rho_\star$ as in Exp. 6.2, and causal graph $\mathcal{G}_{\text{bow}}$ (Fig. E.6.1, top left), the query $P^{\rho_\star}(c_3.y \mid do(c_2.x))$ is relationally identifiable from $P(\mathbf{v}_\rho)$ and $\mathcal{G}_{\text{bow}}$.*

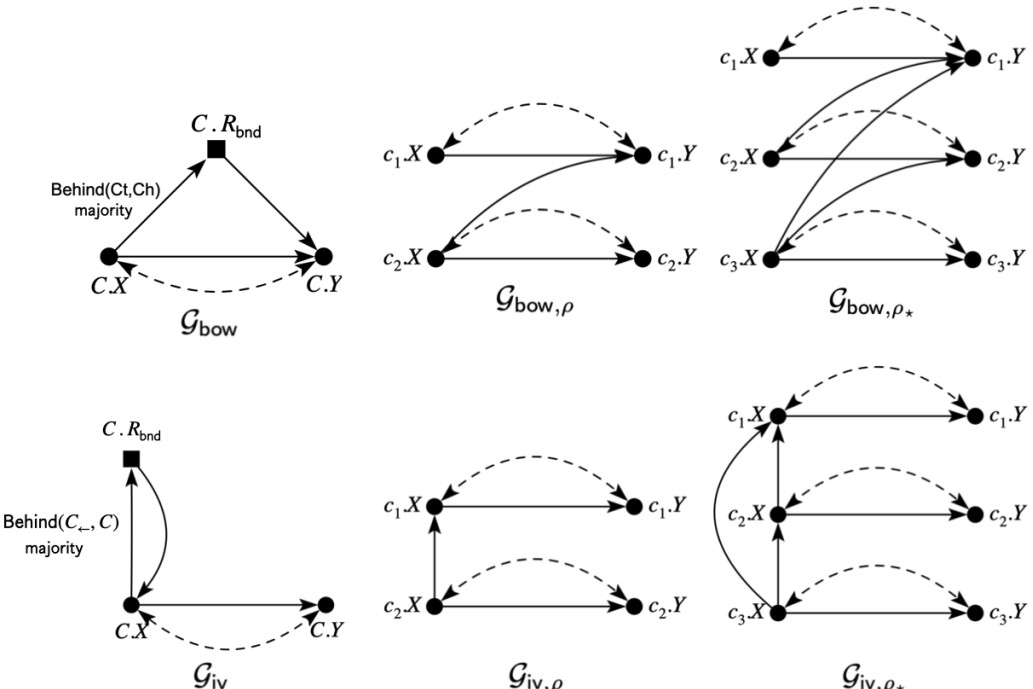

*Figure E.6.1.* Marginalized ground graphs corresponding to $\mathcal{G}_{\mathsf{bow}}$ (top) and $\mathcal{G}_{\mathsf{iv}}$ (bottom) for the source $\rho$ (middle) and target $\rho_\star$ (right) in Exp. 6.2.

*Proof.* There is no directed path from $c_2.X$ to $c_3.Y$ in $\mathcal{G}_{\mathsf{bow},\rho_\star}$ (Fig. E.6.1, top right). By do-calculus and Prop. 4.4, $P^{\rho_\star}(c_3.y \mid do(c_2.x)) = P^{\rho_\star}(c_3.y)$. By Prop. E.3, $P^{\rho_\star}(c_3.y)$ is relationally identifiable, and we are done. $\qquad\square$

An identical derivation shows that $P^{\rho_\star}(c_3.y \mid do(c_2.x))$ is also relationally identifiable from $P(\mathbf{v}_\rho)$ and $\mathcal{G}_{\mathsf{iv}}$.

In both graphs, the non-identifiability of the same-car query follows from Prop. 4.5, by a similar argument as in Ex. C.2.

### E.7. Experiments with Larger Relational Structures

In this section, we evaluate the scability of RNCMs on larger relational structures with up to 75 variables, extending the estimation experiments of Sec. 6.1.

**Setup.** We use the relational causal graph in Exp. 6.1, Fig. 2. For $n \in \{10, 20, 30, 40, 50\}$, we train on a source skeleton with $n$ objects and evaluate on two distinct target skeletons: one with $n$ objects and one with $1.5n$ objects, all with similar neighborhood sizes but different topology. We generate $5 \times 10^3$ datapoints using a regional CTM similar to Exp. 6.1 with count aggregation, with three trials per source skeleton. We compute a large-scale aggregate query: the average of $P(c.B = 1 | do(p.X = 1))$ for all pedestrians $p$ and cars $c$ such that $\mathsf{Path}(p, c)$.

**Generating relational skeletons.** We generate source skeletons $(n_S, n_P, n_C) \in \{(2, 4, 4), (4, 8, 8), (6, 12, 12), (8, 16, 16), (10, 20, 20),\}$ signals, pedestrians, and cars respectively. For each source size, we construct two target skeletons: a matched-size target $\rho_\star$, and a larger target $\rho_{\star\star}$ with $(n'_S, n'_P, n'_C) = (\lceil 1.5 n_S \rceil, \lceil 1.5 n_P \rceil, \lceil 1.5 n_C \rceil)$. many signals, pedestrians, and cars respectively.

For each relation type $R(A, B)$, we generate ground relations as follows. Consider objects of type $A$ and $n_B$ objects of type $B$ ordered as $a_0, \ldots, a_{n_A - 1}$ and $b_0, \ldots, b_{n_B - 1}$. For each object $b_j$ we assign a local neighbourhood of objects $a \in \rho(A)$ as follows. We hoose an anchor index $a(j) \in \{0, \ldots, n_A - 1\}$, and include

$$R(a_i, b_j) \iff |i - a(j)| \le \Delta_R,$$

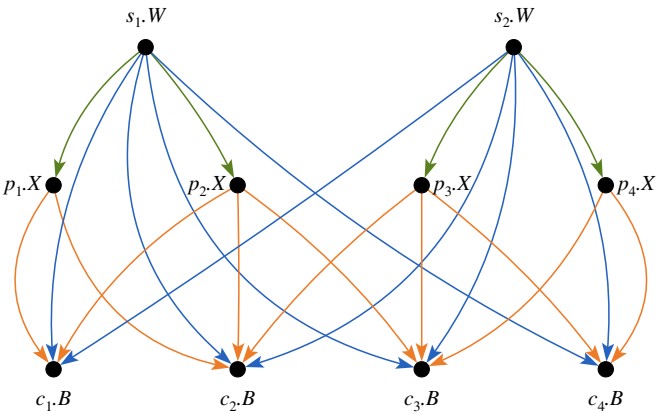

*(a)* Source skeleton $\rho$ with 10 objects.

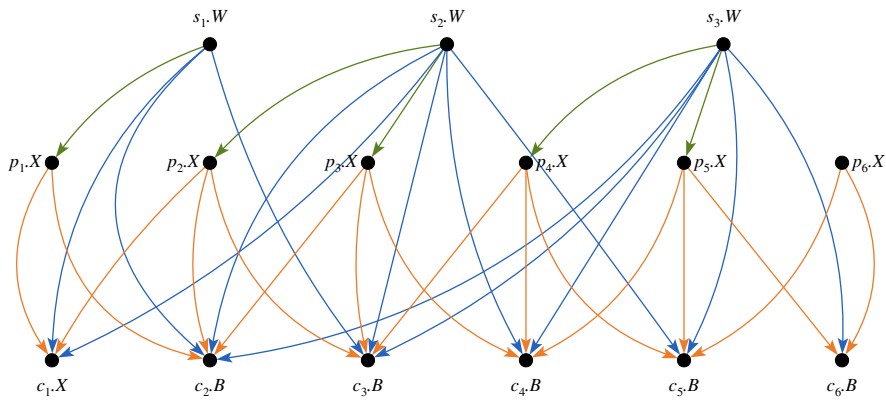

*(b)* Target skeleton $\rho_{\star\star}$ with 15 objects.

*Figure E.7.1.* Marginalized ground graphs for relational skeletons used in scaling experiment (Exp. E.7).

where $\Delta_R$ is a relation-specific neighborhood radius. We use $\Delta_R = 0$ for $\mathsf{Controls}(S, P)$, $\Delta_R = 1$ for $\mathsf{Controls}(S, C)$, and $\Delta_R = 1$ for $\mathsf{Path}(P, C)$.

The source and target skeletons differ in how the anchor $a(j)$ is chosen. For the source skeleton, we use

$$a_{\mathrm{src}}(j) = \left\lfloor \frac{j n_A}{n_B} \right\rfloor.$$

For both target skeletons, we use an alternating floor/ceiling anchor

$$a_{\mathrm{tgt}}(j) = \begin{cases} \left\lfloor \dfrac{j n_A}{n_B} \right\rfloor, & j \text{ even,} \\ \left\lceil \dfrac{j n_A}{n_B} \right\rceil, & j \text{ odd.} \end{cases}$$

Thus, the target skeletons preserve the same local neighborhood radii as the source skeletons but the two are non-isomorphic. See Fig. E.7.1 for a visualization of the marginalized ground graphs of the source skeleton and large target skeleton for $n = 10$.

**Results.** In Fig. E.7.2, we report MSE and runtime of RNCMs for this setting. Accuracy remains stable and high with an increasing number of objects; interestingly, accuracy improves slightly fom $n = 10$ to $n = 20$ objects, which might

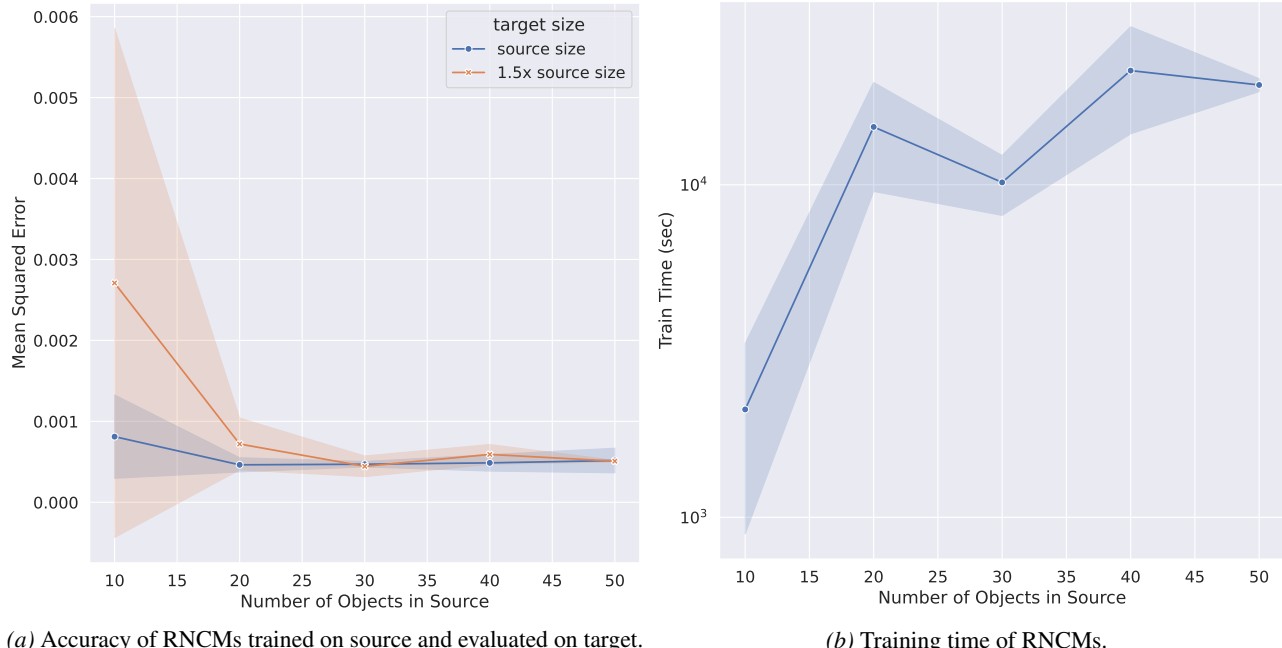

*(a)* Accuracy of RNCMs trained on source and evaluated on target.        *(b)* Training time of RNCMs.

*Figure E.7.2.* Performance of RNCMs on large relational structures (Exp. E.7) averaged across 10 trials. An RNCM is trained on a source skeleton with $n$ objects, and evaluated on two distinct target skeletons: one with $n$ objects, and another with $1.5n$ objects. Accuracy remains stable with increasing number of objects, alongside a manageable increase in runtime.

be explained by how a larger number of objects means a larger number of effective training datapoints for each shared mechanism. While runtime increases with the number of objects, the growth is modest beyond small sizes, suggesting that parameter sharing mitigates the combinatorial explosion typically associated with increased dimensionality. This highlights the scalability of RNCMs to larger relational structures.

### E.8. Experiments with Misspecified Assumptions

In this section, we conduct further experiments to evaluate the sensitivity of RNCMs to two types of misspecification in the assumptions: incorrect edges in the graph structure and incorrect aggregators.

#### E.8.1. MISSPECIFIED GRAPHS

**Setup.** Across 10 trials, we generate $10^4$ observational datapoints for source $\rho_A$ (Fig. 1) using a random regional CTM following the true causal graph in Fig. 2 (with count aggregation). For each trial, we train four RNCMs, each given a different graph as input: the true graph; an underspecified graph without a causal effect from $S.W$ to $C.B$; an underspecified graph without a causal effect from $S.W$ to $P.X$; and an overspecified graph, similar to the true graph but without any relational constraints on the edges (so that every signal affects every car/pedestrian, and every pedestrian affects every car). We evaluate each RNCM on the identifiable query $P^{\rho_C}(c_2.B = 1 \mid do(p_2.X = 1, p_3.X = 1))$ in the target skeleton $\rho_C$. We also evaluate the flat NCM baselines; the two under-specified graph removes the $W \to X$ and $W \to B$ edges respectively, and the 'over-specified' graph is simply the true (complete) graph.

**Results.** We report average MSE across 10 trials in Fig. E.8.1. Misspecification does not necessarily hurt RNCM performance. For instance, the underspecfied graph without the $S.W \to P.X$ causal effect achieves results in similar RNCM accuracy as the true graph. However, the underspecified graph without the $S.W \rightsquigarrow C.B$ path significantly harms RNCM accuracy. Overspecification also results in a decrease in accuracy, though less so than a missing $S.W \rightsquigarrow C.B$ effect. Importantly, despite misspecification in the input graph, RNCMs have greater or comparable accuracy to flat baselines even if the flat baselines are given the true flattened graph.

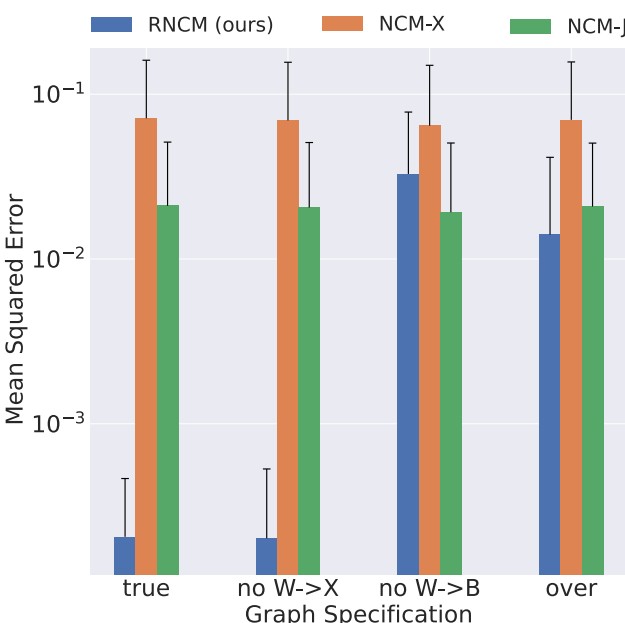

*Figure E.8.1.* Graphs and results of Exp. E.8.1 for evaluating the sensitivity of RNCMs to misspecification in the input graph. Misspecification does not necessarily hurt accuracy: a missing $S.W \rightarrow P.X$ effect yields accuracy similar to the true graph. This shows how sensitivity to misspecification is highly dependent on the graph, query, and type of misspecification. Importantly, RNCMs given a misspecified graph have accuracy greater than or comparable to flat baselines given the true graph.

### E.8.2. MISSPECIFIED AGGREGATORS

**Setup.** We use the graph structure in Fig. 4.1 for both generating the data and training RNCMs, while varying the role aggregators. For each of the count and majority aggregators, we generate $10^4$ observational datapoints each across 10 trials for source $\rho_A$. For each dataset, we train two RNCMs: one using the true aggregator in (count, majority), and one a misspecified aggregator in (majority, count). We evaluate each RNCM on the identifiable query $P^{\rho_C}(c_2.B = 1 \mid do(p_2.X = 1, p_3.X = 1))$in the target skeleton $\rho_C$, similar to Exp. E.8.1.

**Results.** We report average MSE across 10 trials in Fig. E.8.2. Correctly specified count aggregation gives the lowest error. However, estimation using majority aggregation when the true aggregation is count yields in a substantial decrease in accuracy. Correctly specified majority aggegation yields the next lowest error. When the true aggregator is majority, correctly specified majority aggregation yields the lowest error, but estimation using count aggregation yields only slightly lower accuracy, with RNCMs still remaining performant (MSE below $10^{-3}$. This can be explained by how count is a sufficient statistic for computing majority, with the slight decrease in accuracy possibly due to the increased dimensionality of the problem.

## F. Frequently Asked Questions

Q1. What is the difference between a schema, skeleton, RSCM, ground SCM, RCG, and ground graph? How are they related to standard SCMs?

**Answer.** A summary of these concepts, with an illustrative example and a comparison to standard SCMs, can be found in Table B.3.1. Another example with a more explicit comparison to standard SCMs is given in Sec. C.1.

Q2. How does an RSCM differ from a standard SCM?

**Answer.** A standard SCM is defined over a fixed set of variables. It assumes that each 'unit' or object in the domain (e.g., cars, signals, or pedestrians) can be described by this fixed set of variables. Typically, it also assumes that units are i.i.d.. This assumption can be restrictive when units causally affect each other. This has motivated a number of generalizations of SCMs to settings with interference between units. However, these methods still assume a fixed set of units with a fixed set of relationships between them, allowing for queries about this network to be answered using data

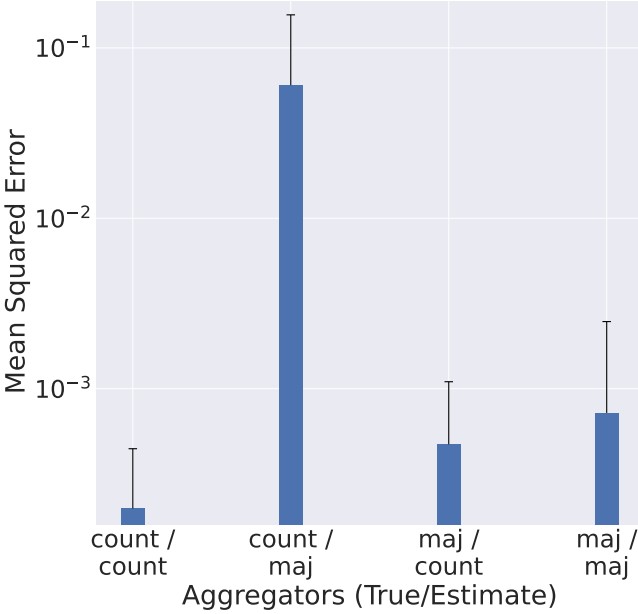

*Figure E.8.2.* Accuracy of RNCMs under misspecification of aggregators (Exp. E.8.2). When the true aggregator is count, using majority aggregation for estimation yields a substantial decrease in accuracy. When the true aggregator is count, using majority aggregation only slightly decreases accuracy, still maintaining MSE under $10^{-3}$.

from the same network. RSCMs generalize this fixed-unit view by defining a causal data-generating process that can be instantiated for any set of units (of possibly different types) and relations between them. RSCMs achieve this by making causal relationships contingent on relational constraints. For example, they may define that a signal's state affects a car's braking speed whenever the signal controls the car, and that the same mechanism for this effect is shared across cars. This mechanism can be applied to any set of signals and cars. Thus, RSCMs enable answering queries about one network of units using data from another (we call this cross-skeleton relational identification (Def. 4.2), evaluated in Exp. 6.1).

Q3. Does grounding an RSCM just produce an ordinary SCM? If so, why not work directly with that SCM?

**Answer.** For any fixed skeleton $\rho$, grounding an RSCM $\mathcal{M}$ does produce an ordinary SCM $\mathcal{M}_\rho$ over instantiated variables. The advantage of the RSCM is that it explains how these ordinary SCMs across different skeletons are related. Different traffic scenes may contain different numbers of cars, signals, and pedestrians, giving rise to different ground SCMs. The RSCM ties these ground SCMs together by sharing mechanisms across object and attribute types. This is what allows the framework to ask whether causal information from one skeleton can transfer to another.

Q4. Do RSCMs require stronger assumptions than standard SCMs?

**Answer.** Not necessarily. RSCMs require explicit assumptions about the relational structure, but standard SCMs also require this in assuming that units are i.i.d.. RSCMs allow a researcher to relax this assumption, allowing more general interactions between units to be made explicit. Often, these interactions themselves can be learned from data; see Q6 for an example.

Q5. When is a standard SCM preferable, and when is an RSCM preferable?

**Answer.** A standard SCM is preferable when the causal problem naturally has a fixed set of variables and the units can reasonably be treated as i.i.d.. If the setting involves units that causally affect each other, an RSCM is preferable for both theoretical guarantees and empirical accuracy (Fig. 3) as suggested by our experiments.

Q6. What prior knowledge is needed in practice?

**Answer.** In our identification results, we assume that the schema, skeletons, and relational causal graph are given. This is analogous to graphical identification in standard causal inference, where the causal graph is assumed to be given.

This prior knowledge can come from expert knowledge, or learned from data. Our framework is complementary to relational causal discovery for learning the causal graph and to methods that infer relational structure (i.e., skeletons) from raw data, such as scene graph extraction in vision.

Q7. What is the intuition behind the non-identifiability result for unseen skeletons (Thm. 3.3)?

**Answer.** The key issue is that source skeletons may not reveal how the model behaves under relational configurations that occur only in the target skeleton. Two RSCMs can agree on every observed source skeleton but disagree on an unseen target skeleton. For example, suppose the target skeleton contains a relational pattern, such as three pedestrians being in the path of a car, that never occurs in the source skeletons. One RSCM may define the car's behavior to change when this pattern occurs, while another RSCM may ignore that pattern. If the pattern is absent from all source skeletons, both models induce the same source distributions, but they can induce different target distributions. Therefore, the target distribution is not identifiable without further assumptions.

Q8. How should Thm. 3.3 be interpreted for relational machine learning?

**Answer.** Thm. 3.3 formalizes a limitation of generalization across relational structures (e.g., as in inductive graph learning) even for observational queries–the lowest rung of the causal hierarchy, that does not involve interventions or counterfactuals. The theorem states that even if mechanisms are shared across objects of the same type (as is common in relational machine learning), the observational distributions of a given set of skeletons may not uniquely determine the observational distribution of an unseen skeleton. This suggests that parameter sharing alone is not enough to guarantee cross-skeleton generalization.

This does not mean that relational models such as GNNs cannot generalize in practice. Rather, it clarifies that successful generalization relies on additional assumptions, such as smoothness, regularity, invariance, or architectural biases. Our framework leverages relational causal graphs to make such assumptions explicit.

Q9. How does the approach scale to larger relational structures?

**Answer.** The main source of scalability is parameter sharing. The same structural mechanism is reused across all instances of a given attribute type. For example, the same car-braking mechanism applies to every car, regardless of how many cars appear in a particular skeleton.

In additional experiments, we train on skeletons with up to $50$ objects and evaluated on unseen skeletons with up to $75$ objects (Exp. E.7). Accuracy remains stable as the number of objects increased, while runtime increases moderately. That said, scalability also depends on the choice of relational aggregation. High-dimensional aggregators, such as histograms over large neighborhoods, can be expensive; lower-dimensional summaries such as sums, means, maxima, or attention pooling may be preferable in large-scale applications.

Q10. Does the framework require discrete variables?

**Answer.** The identification theory of Sec. 4 does not require discrete variables. The discreteness assumption enters in our neural implementation (Sec. 5), which follows discrete canonical neural causal model constructions. Extending the neural implementation to continuous variables with theoretical guarantis an important direction for future work.

Conceptually, continuous attributes can be included in an RSCM in the same way as discrete attributes: they are attributes of entities or relations and have structural equations. The main challenge is practical estimation and optimization, not the relational semantics themselves.

Q11. How do attributes on relations fit into the framework?

**Answer.** Relation attributes are attributes attached to relation instances rather than entity instances. For example, consider entities Student and Course, and a relation Takes(Student, Course). The relation instance Takes$(s, c)$ could have an attribute Takes.$G$, representing the grade student $s$ receives in course $c$. Relation attributes behave like entity attributes in the formalism.

