# OpenReview forum: "Relational Structural Causal Models"
_ICML.cc/2026/Conference — ICML 2026 regular_

### Official Review · Reviewer_q4Bq · 2026-03-13

**Soundness:** 4
**Presentation:** 3
**Significance:** 3
**Originality:** 3
**Overall Recommendation:** 5
**Confidence:** 3

**Summary:**

This paper introduces Relational Structural Causal Models (RSCMs), extending traditional SCMs to enable causal reasoning and combinatorial generalization across varying object configurations. The framework addresses the limitations of standard causal models, which typically assume a fixed number of variables with a fixed structure, and of relational models, which are associational and lack causal semantics for answering interventional and counterfactual queries. Two non-identifiability results are provided: (1) the impossibility of identifying observational distributions across relational skeleteons, and (2) the impossibility of identifying interventional and counterfactual distributions within a relational skeleteon without further assumptions. To overcome these limits, the authors introduce RCGs, which encode relational constraints and provide symbolic criteria for identifying causal effects across skeletons. They develop a neural implementation of RSCMs that shares parameters across different object types. These models are provably expressive enough to represent any RSCM and can perform causal identification in a practical scenario. Through simulated traffic scenes involving cars, signals, and pedestrians, the authors demonstrate that RNCMs significantly outperform non-relational baselines and can accurately estimate causal effects on entirely new object combinations.

**Compliance With Llm Reviewing Policy:**

Affirmed.

**Final Justification:**

All my primary concerns have been addressed in the authors' detailed rebuttal. The authors have promised to make the following changes in the revision.
- Clarity and presentation: Simplify mathematical notations and include a summary table.
- Include an intuitive observational non-identifiability example corresponding to Thm 3.3 and discuss its implications on existing relational work in machine learning.
- Include sensitivity experiments on the choice of aggregators in the main paper or appendix.
- Include comparison with a more general relational baseline.

Having acknowledged these changes, I maintain my original positive score and recommend acceptance.

**Key Questions For Authors:**

1. **Clarifying intuition behind Theorem 3.3.**: From what I understand from the proof of Theorem 3.3 in the Appendix, the key idea is that it is possible to design a relational constraint (e.g., if a pedestrian is in the path of a car) that is only satisfied in the target skeleton, and we can define two RSCMs, M and M', where one of them captures the mechanism behind this constraint — that it is possible for a pedestrian to be in the path of a car and hence switches to a different behavior in the target skeleton — while M always assumes this to be false, so it does not capture this constraint. While M and M' both agree on all source skeletons, their behavior would differ in the target skeleton, leading to non-identifiability. Is this a correct, simple example illustrating the non-identifiability of observed distributions? Isn't this equivalent to a lack of observability of all possible values of constraints in the source skeleton, which the support condition in Theorem 4.3 is addressing?
2. What are the implications of the results of Theorem 3.3 for existing work on relational modeling in machine learning?
3. How sensitive is the model to the choice of aggregators?
4. Which aspects of the framework have the potential to be learned from data for scaling the approach to large-scale, real-world settings? Rather than predefining constraints such as Path(P, C), could the framework be adapted to learn these relationships directly from raw spatial data while preserving causal semantics?

Despite the weaknesses mentioned above, I understand that this is a first step toward formally establishing this new class of causal models, and some of the suggestions — e.g., reducing reliance on prior knowledge — are promising directions for future work. Given the soundness and rigor of the results presented, I recommend acceptance.

**Limitations:**

The paper is transparent about the constraints of the proposed framework and discusses the theoretical limits, neural assumptions, and empirical failure modes. I'd recommend that authors also include a discussion of the risks of misspecification of the assumed relational causal graph and violations of assumptions in the real world.

**Strengths And Weaknesses:**

**Strengths**
- The paper establishes a rigorous formal framework for reasoning about causality in dynamic environments, which is an important research problem for practical applications of causal reasoning.
- The authors provide important impossibility results (Theorems 3.3 and 3.4), proving that without specific structural assumptions, it is impossible to identify even observational distributions for unseen object combinations, which is a very interesting and surprising result. The impossibility of causal inference from observational data without further assumptions is well-known, but Theorem 3.3, in my opinion, is one of the most important contributions of this work. Hence, the paper would greatly benefit from more intuitive explanations and a discussion of its implications for existing work on relational models in machine learning (see the points below).

**Weaknesses**
- The paper is quite notation- and math-heavy, which is justified to ensure formality and rigor, but it would help if more intuition about the theoretical proofs were provided in the main paper.
  - More concretely, Theorem 3.3 seems quite important, but it is not immediately clear how an observational distribution can fail to be identified across skeletons. Including an intuitive example of non-identifiability in the main paper would certainly improve readability, especially for general readers. In addition, discussing how realistic those non-identifiability cases are in current relational benchmarks would help clarify the impact of this work on existing work.
- **Large dependency on prior knowledge for practical applications**: To apply the framework in practice, RNCMs require knowledge of the schema, causal graphs, aggregator functions, etc. If the user lacks accurate prior knowledge of the causal and relational structure, the framework cannot guarantee identification or accurate estimation. Do you have ideas or suggestions on how the framework could be adapted to those settings?
- **A baseline comparison with a relational non-causal approach is missing:** It would be useful to see how relational approaches would perform (e.g., graph neural networks and other approaches introduced in Battaglia et al., 2018), especially in combinatorial generalization settings.

---

> ### Author Rebuttal · Authors · 2026-03-31
>
> We thank the reviewer for their positive evaluation of our work and their constructive suggestions.
>
> **W1, Q1: Notation clarity and intuition for Thm. 3.3**
>
> We will:
> - simplify notation (e.g., use more idiomatic variables like “o” for objects),
> - add a summary table and examples comparing standard SCMs and RSCMs, and
> - include a concrete example of observational non-identifiability across skeletons in the main paper.
>
> The reviewer’s example for Thm. 3.3 is exactly correct. Regarding the connection between Thm. 3.3 and the support condition of Thm. 4.3: these are closely related but not equivalent. Thm. 3.3 shows inference across skeletons is impossible without a graph, while Thm. 4.3 characterizes when identification becomes possible given one. In particular, there are cases where the support condition is satisfied and the query is identifiable given the graph, but not identifiable without it. We will clarify this distinction and give an example in the appendix.
>
> **Q2: Implications of Theorem 3.3 for relational machine learning**
>
> This result provides a formal perspective on known generalization failures in relational models such as GNNs. In particular, the theorem generalizes findings that models underperform on structures dissimilar to their training data (e.g., different sizes or configurations), even for simple tasks [1].
>
> We highlight two aspects of Thm. 3.3:
> - It is not a worst-case non-identifiability result, since it applies to any given schema, skeleton, and observational distribution.
> - It holds even under parameter sharing across instances, a key assumption in relational deep learning.
>
> As such, Thm. 3.3 suggests that purely associational relational models may fail under distribution shift in relational structure, and motivates incorporating causal structure to avoid reliance on spurious correlations [2, 3, 4, 5]. We will expand this discussion in the paper.
>
> That being said, it is possible that for the many RSCMs consistent with some observational distribution, some of these RSCMs are more ‘regular’ than others. Our proof of Thm. 3.3 constructs a counterexample RSCM for a given true RSCM. It’s possible that the counterexample RSCM is less ‘regular’ than the true RSCM, and that the solutions learned by deep learning methods are biased towards more ‘regular’ solutions by architectural choices such as regularization. Thm. 3.3 highlights the importance of explicitly characterizing and evaluating if such ‘regularity’ assumptions license generalization.
>
> **Q3: Sensitivity to aggregator choice**
>
> To evaluate robustness to aggregation choices, we conducted an additional experiment mirroring Exp. 6.1 (source A, target C), where data is generated and the model is trained under different combinations of count and majority aggregation (plots: https://anonymous.4open.science/r/rscm-rebuttal-6549/README.md).
> We observe that misspecification of the aggregator leads to a degradation in accuracy, but the effect is asymmetric.
> - When the true aggregator is count and estimation uses majority, the reduction in accuracy is relatively small, suggesting that lossy aggregators can still yield accurate effect estimation.
> - In contrast, when the true aggregator is majority and estimation uses count, the degradation is larger, possibly due to the increased dimensionality of the model.
>
>
> **W2, Q4: Learning relational skeletons and causal graphs from data**
>
> Our work is complementary to research on learning causal and relational structure. One could combine learned relational structures (e.g., scene graphs from vision models) and learned causal graphs (e.g., from relational PC) with our framework. Our formalization also provides a foundation for extending causal discovery methods (e.g., FCI-style algorithms) to relational settings with latent confounding. We will expand discussion of these directions in our paper.
>
> **W3: Missing relational non-causal baseline**
>
> The NCM-J baseline incorporates relational structure by only considering related data triples. The challenge in designing a better baseline is that standard relational methods do not distinguish P(y | do(x)) from P(y) or P(y | x). Moreover, these methods are complementary to our approach, since they can be used to estimate observational probabilities that are then plugged into a causal estimand. Nevertheless, we will include a more general relational baseline, as we agree this comparison would clarify whether relational structure alone suffices for the task.
>
> [1] Yehudai, G. et al. On Size Generalization in Graph Neural Networks. arXiv:2010.08853 2021.
>
> [2] Li, H. et al. OOD-GNN: Out-of-Distribution Generalized Graph Neural Network. IEEE TKDE 2023.
>
> [3] Fan, S. et al. Generalizing Graph Neural Networks on Out-Of-Distribution Graphs. IEEE TPAMI 2023.
>
> [4] Bevilacqua, B. et al. Size-Invariant Graph Representations for Graph Classification Extrapolations. ICML 2021.
>
> [5] Wu, Y. et al. Discovering Invariant Rationales for Graph Neural Networks. ICLR 2022.

---

> > ### Author Rebuttal · Reviewer_q4Bq · 2026-04-01
> >
> > Please see my official comment above.
> >
> > Edit - I meant please see my original review. I’d like to maintain my positive score. Apologies for confusion.

---

> > > ### Author Response · Authors · 2026-04-03
> > >
> > > We thank the reviewer for acknowledging our rebuttal! Unfortunately, we are unable to currently view any official comments. We would appreciate if the reviewer could edit their rebuttal acknowledgment to include any remaining questions, and we would be happy to answer them by editing this response.
> > >
> > > Edit: Understood! We thank the reviewer for engaging with our rebuttal, and for their continued positive evaluation of our work.

---

### Official Review · Reviewer_Zeez · 2026-03-13

**Soundness:** 3
**Presentation:** 2
**Significance:** 3
**Originality:** 3
**Overall Recommendation:** 4
**Confidence:** 3

**Summary:**

This paper studies how to extend structural causal models to relational settings where both the number of objects and the relation structure can vary across instances, such as different traffic scenes with different cars, pedestrians, and signals. Their framework is termed Relational Structural Causal Models (RSCMs), which define a high-level causal template over a relational schema and can be instantiated on different skeletons, each skeleton yields a SCM for a particular scene. On the theory side, the paper shows that queries about unseen skeletons are generally not identifiable without further assumptions, then introduces relational causal graphs and develops identifiability results for when observational, interventional, or counterfactual queries can be transferred across or within skeletons. On the method side, it proposes Relational Neural Causal Models as a neural network instantiation of this framework and evaluates them on simulated traffic-scene tasks. Overall, the paper’s main contribution is a causal framework for reasoning across changing object-relation structures, together with identification theory and a practical neural implementation.

**Compliance With Llm Reviewing Policy:**

Affirmed.

**Final Justification:**

This paper proposes Relational Structural Causal Models (RSCMs), extending standard structural causal models to settings with varying object configurations and relational structure. My initial main concerns are the limited experimental scope (restricted to simulated traffic scenes), the clarity of notation and presentation, and the practical applicability of the framework. The authors’ rebuttal addressed these concerns satisfactorily by providing additional experiments on larger relational structures and more complex queries, clarifying the role of simulations, and committing to improved presentation clarity. While the empirical validation remains limited to synthetic settings and the framework still relies on strong structural assumptions (e.g., a known relational causal graph), I find the work novel and likely to inspire further work in causal inference. I therefore maintain my recommendation of a weak accept.

**Key Questions For Authors:**

See weaknesses

**Limitations:**

Yes

**Strengths And Weaknesses:**

## Strengths

- The paper makes a substantial technical contribution. It introduces Relational Structural Causal Models (RSCMs), proves impossibility results for unseen skeletons without additional assumptions, develops relational causal graphs and corresponding identification results, and then proposes Relational Neural Causal Models (RNCMs) as a practical instantiation.

- The experiments are well aligned with the paper’s goals. The traffic-scene setup is a sensible testbed for variable object-relation structure, and RNCMs outperform flat non-relational neural causal baselines in several settings.

- The originality is strong. To the best of my knowledge, the combination of a relational analogue of SCM-style reasoning with identification theory across changing skeletons is novel.

## Weaknesses

- The experiments are limited to simulated traffic scenes. It would be helpful if the authors clarified whether this is mainly because it is difficult in practice to obtain real-world data with the level of relational and causal supervision needed to evaluate this framework. A brief discussion of this would help readers better understand the current experimental scope.

- While the paper is reasonably organized, I found the notation and terminology hard to follow, even coming from a Pearl-style SCM background. In particular, the distinctions among schema, skeleton, grounded SCM, RCG, and RSCM are easy to lose track of. An explicit side-by-side comparison with the standard SCM / causal graph setting would likely make the paper much easier to follow.

- The paper could do more to explain the practical tradeoff between RSCMs and standard SCMs: what additional assumptions or data are needed in the relational setting, and when the relational framework is actually preferable in practice.

---

> ### Author Rebuttal · Authors · 2026-03-31
>
> We thank the reviewer for their feedback and for recognizing the technical contribution, empirical backing, and originality of our work.
>
> **W1: Experiments limited to simulated traffic scenes**
>
> We agree this is an important limitation and will clarify this in the paper. While there are real-world datasets with relational supervision (e.g., traffic scenes [1], images [2], molecules [3]), evaluating causal methods in these settings is challenging for two main reasons:
>
> - Constructing a reliable causal graph often requires expert knowledge (e.g., in biological or chemical settings)
>
> - Ground-truth causal effects and distributions are typically unavailable if only observational samples are provided.
>
> Therefore, we use simulations to evaluate our theoretical claims. We will expand our discussion of how our work may be applied to real-world data, e.g., extending to modalities such as images by adapting existing work on causal image generation [4].
>
> In addition, following reviewer feedback, we have conducted additional experiments on larger relational structures, more complex causal queries, and misspecified graphs/aggregators, which further demonstrate the practical behavior of our method.
>
> **W2: Clarity of notation and comparison to standard SCMs**
>
> We agree that the distinctions between the various terms we define can be challenging to track. We will add a summary table as well as illustrative examples in the appendix comparing our framework to standard SCMs.
>
> **W3: Practical tradeoffs between RSCMs and standard SCMs**
>
> Thank you for the chance to clarify this point. RSCMs are particularly useful in settings with relational or non-i.i.d. structure, where standard SCM assumptions fail. Applying standard SCM tools requires flattening the data (as in our join baselines), which is both theoretically unsound and yields worse accuracy in our experiments.
>
> In addition, relational SCMs avoid costly and error-prone feature engineering and joins that can lead to feature explosion, motivations shared with relational deep learning more broadly [6].
>
> While RSCMs may appear to require additional assumptions (e.g., specifying a relational skeleton), this generalizes the stronger i.i.d. assumption in standard SCMs. In this sense, our framework relaxes assumptions in standard SCMs. That being said, prior work shows that in some regimes (e.g., linear-Gaussian settings with certain graph structures), standard SCM approaches can still perform well despite violations of i.i.d., and may be preferable for simplicity [5].
>
> We hope these clarifications improve both the readability and practical positioning of our work.
>
> [1] Caesar, H. et al. nuScenes: A multimodal dataset for autonomous driving. CVPR 2020.
>
> [2] Krishna, R. et al. Visual Genome: Connecting Language and Vision Using Crowdsourced Dense Image Annotations.  IJCV 2017.
>
> [3] Tingle, B. et al. ZINC-22—A Free Multi-Billion-Scale Database of Tangible Compounds for Ligand Discovery. J Chem Inf Model 2023.
>
> [4] Pan, Y. and Bareinboim, E. Counterfactual image editing. ICML 2024.
>
> [5] Zhang, C., Mohan, K., and Pearl, J. Causal Inference with Non-IID Data using Linear Graphical Models. NeurIPS 2022.
>
> [6] Ranjan, R. et al. Relational Transformer: Toward Zero-Shot Foundation Models for Relational Data. ICLR 2026.

---

> > ### Author Rebuttal · Reviewer_Zeez · 2026-04-03
> >
> > Thank you for your detailed response. Please include the additional discussion in the updated version of the paper, and I retain my initial positive score.

---

> > > ### Author Response · Authors · 2026-04-05
> > >
> > > We are happy to hear that the reviewer's concerns have been addressed, and thank them for their valuable feedback!

---

### Official Review · Reviewer_wLgQ · 2026-03-13

**Soundness:** 3
**Presentation:** 3
**Significance:** 4
**Originality:** 2
**Overall Recommendation:** 5
**Confidence:** 4

**Summary:**

This paper introduces relational structural causal models (RSCMs), i.e., causal models that build upon relational structures. RSCMs can contain unobserved confounding and can manifest in different skeletons (grounding). Identifiability of causal queries is analyzed, showing that there is no general identifiability from observational data, but also showing conditions under which identification is possible. Relational neural causal models are described, which can learn relational causal models from data and identifiability results are shown experimentally using these neural models.

**Compliance With Llm Reviewing Policy:**

Affirmed.

**Final Justification:**

I had three types of concerns that have all been addressed:

1. Clarity: The authors promised to clarify contributions (introduction) and include additional clarifications in the appendix (e.g., minor point 3).
2. The authors promised to include a separate related work section in the appendix.
3. The additional experiments on larger structures, as well as frontdoor and backdoor estimation strengthen the empirical evaluation.

Neither do I have further concerns remaining, nor did any of the other reviews and rebuttals highlight another concern that I missed that would lead me to recommend a lower score. Assuming the promised changes will be included in the final manuscript, I recommend acceptance (5).

**Key Questions For Authors:**

1. Can you give an example of an attribute on a relation? This is allowed following the definition for a relational schema, but it is never used in the examples as far as I can tell. It would be great if the authors could provide an example of a relational skeleton and (relational) causal graph where relations have attributes. What is the role of such attributes when it comes to identifiability?

2. In the NCM section, interventional input data is given. Why? I understand that interventional data is helpful and can be used as well, but this makes it seem as if that was required. Is it correct that interventional data is only specified here, because it is a valid (and informative) input, or would the algorithm actually fail if only observational data were supplied?

---
**Edit after rebuttal:** My concerns have been addressed (see rebuttal acknowledgment). I raise my score from 3 to 5.

**Limitations:**

Yes

**Strengths And Weaknesses:**

**Strengths**

- **S1** The traffic scene example of Figure 1 is used throughout the paper. It is intuitive and greatly helps to visualize and understand the discussed method. I particularly like how the authors repeatedly go back to it when defining new theoretical concepts.
- **S2** The theory and methodology is rigorous and sound. Assumptions and limitations are stated clearly and proofs are spelled out comprehensibly.
- **S3** The contributions are significant and relatively original. Combining the fields of relational and causal models is a promising and interesting direction of research and this paper considers some settings (e.g., confounding) that have not been considered before.
- **S4** The NCM for relational models is a valuable contribution as it allows for easily applying such models in practice without strong assumptions on the function space and the experiments show how it can successfully distinguish identifiable and non-identifiable cases.

**Weaknesses**

- **W1 Missing related work section / shallow discussion of related work.** A separate related work section is missing entirely. While many references are included in the introduction, there is no dedicated part of the paper where related works are actually discussed and, at least briefly, compared to. While several relevant papers are cited, none of them are described individually and several important references are even missing entirely (see next weakness as well). The contextualization of new work into existing literature is a crucial element of scholarly work in academia and should not be disregarded. I strongly suggest that the authors reconsider the omission of the related work section and properly compare this paper to the relevant body of existing work.
- **W2 Missing references.** There are several important papers on relational causal models that are not included in the current draft [1,2,3]. While the contributions of this submission still stand, they should still be discussed.
- **W3 Introduction does not nicely introduce the paper.** As a result of the missing related work section, the introduction could do a better job of introducing the content of the paper. In its current state, its purpose seems to be rather to replace the related work section, while only touching on the results of the paper. The actual paper content/contributions only start with the "Contributions" block on the second page and, as a result, are kept very brief, resulting in less clear expectations from the reader's point of view and impacting the flow of the paper.
- **W4 Experimental results are weak.** The experimental results are relatively small and not extensive. Due to this narrow application, how well such models would actually perform in more complex settings remains unclear
  - There is overall only a small amount of experimental evidence. The main evaluation considers only one specific setting and there is only one experiment with two queries on identification.
  - There are neither evaluations on real-world data, nor are there any experiments to investigate how well the proposed methodology scales to larger (even synthetic) problems.
  - The queries in the identification experiment are very easy. For $Q_d$, $c_3.Y$ is not even an ancestor of $c_2.X$, so this interventional query is equivalent to just omitting the intervention. The query for $Q_s$ is a valuable sanity check, but the non-identifiability of a causal relation between two adjacent variables that are confounded is the minimal example for an unidentifiable query. More complex queries (e.g., mediator, back-door, front-door) were not considered.


While, in its current form, I lean towards rejection, I think that the weaknesses are in large parts very addressable and not inherent flaws of the methodology. I am open to increasing my score during the rebuttal phase if appropriate changes are made.

**Minor Points**

1. Are there multiple errors in Figure 1? As per my understanding, the causal graphs should have the same edges as their respective relational skeleton. This does not hold for "City Block" and "T-Junction". Is that a mistake in the figure or in my understanding?
2. The concept of the "target skeleton" is not explained. Is this simply the skeleton for which a query should be computed? This could be described more clearly. Also, Figure 1 describes $\rho_B$ as both source and target skeleton, I assume $\rho_C$ should be the target skeleton.
3. The paper deals with many different types of graphs and skeletons (relational, causal, grounded versions...). It can be confusing and take time to get used to the respective differences. Perhaps a section could be added to the appendix that briefly contains examples for all of them side-by-side, along with other terminology. Such a reference would make it easier to follow along when reading the paper.
4. In Figure 1, adding the respective indices to the entities in the "real-world" would simplify understanding the skeletons a bit.
5. I suggest vectorizing the figures whenever possible, as they will then not be dependent on whatever resolution is used.

Typos

6. In Definition A.4, there is a missing closing bracket under point 1.

7. There is missing punctuation at the end of footnote 1.

8. There is a typo in Definition A.6 "in in".

9. There is a broken reference in Definition A.7.

10. In the proof idea of Theorem 3.4, there seems to be a verb missing in the second sentence.

11. In the proof of 4.3, above equation 1, it says $\textbf{pa}_{x].A}$, the "]" should be removed.

12. In line 1043, a closing bracket is missing.

[1] Ahsan, Ragib, David Arbour, and Elena Zheleva. "Learning relational  causal models with cycles through relational acyclification." *Proceedings of the AAAI Conference on Artificial Intelligence*. Vol. 37. No. 10. 2023.

[2] "Luttermann, Malte, et al. "Lifted causal inference in relational domains." *Causal Learning and Reasoning*. PMLR, 2024."

[3] Negro, Matteo, et al. "Relational causal discovery with latent confounders." *Proceedings of the Forty-First Conference on Uncertainty in Artificial Intelligence*. 2025.

---

> ### Author Rebuttal · Authors · 2026-03-31
>
> We thank the reviewer for their attentive reading and for their openness to increasing their score given that the weaknesses are “very addressable and not inherent flaws of the methodology.” We agree that the main issues are presentation and empirical scope, and we address both below.
>
> **W1-2: Missing related work / references.**
>
> We agree that better contextualization is important. In the revision, we will add a dedicated related work section (in the appendix) that explicitly compares our framework to prior work on relational causal discovery and lifted inference, including the works you listed.
>
> Our key distinction is that we study *identifiability and transfer across varying relational skeletons*, whereas most prior work assumes a fixed skeleton (e.g., a single relational database). We will clarify this more explicitly.
>
> **W3: Introduction and clarity of contributions**
>
> We will revise the introduction to more clearly foreshadow our contributions and results, and use Fig. 1 earlier to concretely illustrate the task and setting.
>
> **W4: Experimental scope**
>
> Our experiments are designed as controlled tests of the theoretical results; each experiment isolates a specific phenomenon (e.g., cross-skeleton transfer or non-identifiability). These phenomena cannot be reliably evaluated on real-world data due to the absence of ground-truth causal effects. To address the reviewer’s concerns, we conduct several additional experiments that strengthen the empirical evidence (plots: https://anonymous.4open.science/r/rscm-rebuttal-6549/README.md).
>
> a. *Larger relational structures (scalability and transfer)*: We train on skeletons with up to 50 objects and evaluate on unseen skeletons with up to 75 objects (1.5x larger). We observe stable and high accuracy with increasing size, along with moderate runtime growth. This demonstrates that learned mechanisms generalize across object counts, consistent with our theoretical framework (see response to reviewer Bpzd, Q4 for details).
>
> b. *More complex queries and additional settings*: ​​To address concerns about query simplicity, we evaluate:
>
> - A *frontdoor estimation* setting requiring adjustment over multiple variables (c1.Z, c2.Z, and c3.Z), using the same source/target as in Exp. 6.2. We found an average MSE of 0.000817921 ± 0.0017969 across five trials, showing correct estimation of this identifiable query.
>
> - A *backdoor identification* setting requiring adjustment over multiple variables (c1.Z, c2.Z, and c3.Z), extending Exp. 6.2. The observed convergence of the min/max estimators shows correct identification.
>
> All evaluations are performed on unseen skeletons, testing cross-structure transfer.
>
> **Minor points**
> 1. The reviewer is correct; Fig. 1 is missing s2.W -> p2.X in skeletons B and C.
> 2. We will clarify that the target skeleton is the skeleton on which the query is to be evaluated.
> 3. We will add the suggested examples and a side-by-side summary table.
> 4. We will fix the noted typos, references, and figure indices.
>
> **Q1: Example of an attribute on a relation**
>
> A standard example is a schema with entities Student and Course and relation Takes(Student, Course), where the relation has an attribute such as Grade. Relational attributes are treated exactly like entity attributes, and can participate in queries such as “what is the effect of course difficulty on a student’s grade in that course”? We will add an example to illustrate this in the appendix.
>
> **Q2: Is interventional input required for RNCMs?**
>
> Thank you for the chance to clarify this point. Interventional data is not required as input. We allow it since our definition of relational identification permits arbitrary source distributions, and interventional data can enable identification when available. However, it is not required for correctness in theory or practice, and our current experiments use observational data only as input.
>
> **Conclusion.** We hope that, taken together, these additions address the reviewer’s concerns regarding empirical scope by demonstrating scalability to large structures, accuracy on non-trivial identifiability queries, and evidence for the theoretical guarantees of our framework.

---

> > ### Author Rebuttal · Reviewer_wLgQ · 2026-04-03
> >
> > I thank the authors for the comprehensive and accommodating rebuttal. The changes regarding clarity (W3, minor points) sound reasonable. In particular, I welcome the changes promised regarding discussion on related work (W1, W2). My concerns have never been about the contributions of the paper, but that the contextualization of the prior literature was lacking. For this reason, I was leaning towards rejection, but am now happy to recommend acceptance. The additional experiments strengthen the empirical evaluation of this paper (W4) and I recommend including them in the final manuscript. Both questions were answered sufficiently, raising neither follow-up questions nor new concerns. As all concerns have been addressed and the empirical evaluation (larger-scale experiment, frontdoor and backdoor estimation) has been extended, I have no major weakness remaining. While, naturally, evaluation on real-world experiments would always be a very valuable insight, the difficulty of finding or coming up with useful real-world benchmarks is a well-known problem in causal research, which can reasonably be seen to be outside the scope of this paper.
> >
> > Therefore, I assume that the promised changes will all be integrated well into the final manuscript and I raise my score to 5 (accept) and have no follow-up questions.

---

> > > ### Author Response · Authors · 2026-04-05
> > >
> > > We thank the reviewer for engaging with our rebuttal, and for their positive assessment of our work!

---

### Official Review · Reviewer_Bpzd · 2026-03-15

**Soundness:** 3
**Presentation:** 2
**Significance:** 3
**Originality:** 3
**Overall Recommendation:** 4
**Confidence:** 1

**Summary:**

This paper introduces Relational Structural Causal Models (RSCMs), a generalization of Pearl's structural causal models (SCMs) to domains where the number, types, and relations among objects can vary across settings (i.e., relational skeletons). The authors first establish impossibility results showing that without structural assumptions, neither observational nor interventional distributions for unseen skeletons can be identified from data collected on other skeletons. The authors further introduce relational causal graphs that encode type-level causal structure and derive conditions under which causal knowledge transfers across skeletons. They then propose relational neural causal models (RNCMs) that parameterize shared mechanisms as neural networks, prove their expressiveness, and provide a min/max optimization algorithm for identification. Experiments on synthetic traffic scenes validate these theoretical results.

**Compliance With Llm Reviewing Policy:**

Affirmed.

**Final Justification:**

I maintain my original positive score

**Key Questions For Authors:**

1. Please see my comments on the Weaknesses part.

2. How sensitive are the RNCM estimation results to misspecification of the relational causal graph $G$? If $G$ includes a spurious edge or omits a true edge, how does estimation accuracy degrade? An empirical investigation of robustness to graph misspecification would substantially strengthen the practical relevance of the approach.

3. Can the framework be extended to continuous attributes?

4. How does the approach scale to larger relational structures?

**Limitations:**

Yes

**Strengths And Weaknesses:**

__Strengths__:

1. The paper addresses a gap at the intersection of causal inference and relational machine learning. The formalization of RSCMs as a unifying generative model for varying relational structures is reasonable and well-motivated.

2. I did not read the proofs in detail, but from my understanding, the theoretical analysis is rigorous. In particular, the impossibility results (Theorems 3.3 and 3.4) are well-constructed and valuable because they precisely characterize the limits of learning in this setting and motivate the need for additional graphical assumptions. The identification theory provides both sufficient conditions for cross-skeleton transfer (Theorem 4.3) and a necessary condition for non-identifiability (Proposition 4.5), giving a complete picture of when relational identification is possible.

3. The paper is well-structured and well-positioned against prior work.

__Weaknesses__:

1. My main concern is that the experiments are entirely synthetic and small-scale: all use known, correct relational causal graphs with binary attributes and skeletons of at most 9 variables. While I appreciate that this is primarily a theory paper, the introduction motivates the work through autonomous driving, robotics, and biology: domains involving far larger, continuous-valued settings. Even without real-data experiments, a discussion of the practical path from the current setup to these domains (e.g., handling continuous attributes, scaling the histogram aggregation, obtaining or validating the relational causal graph) would strengthen the paper.

2. The entire identification part assumes $G$ is given and correct, with no discussion of how $G$ might be learned, validated, or how sensitive results are to misspecification. I think that this is more consequential than in standard SCMs: a wrong edge at the type level propagates to every instance across all skeletons, amplifying the error.

---

> ### Author Rebuttal · Authors · 2026-03-31
>
> We thank the reviewer for their thoughtful feedback and for appreciating our theoretical contributions. Please see https://anonymous.4open.science/r/rscm-rebuttal-6549/README.md for plots supporting the findings below.
>
> **W1-2: Learning the causal graph**
>
> Our work is complementary to related literature on relational causal discovery, i.e., learning the graph (or an equivalence class of graphs) from data. We characterize what is identifiable given such a structure. We will clarify this in the paper.
>
> **W1-2, Q2: Empirical sensitivity to misspecification of the graph**
>
> To address this concern, we conducted an additional experiment mirroring the setup in Exp. 6.2, comparing:
> - the true graph
> - an under-specified graph (missing edge S.W -> C.B)
> - an over-specified graph (complete graph, i.e., no relational constraints on edges)
>
> We used source skeleton A and target skeleton C, evaluating the query P(c1.B | do(p1.X)), which requires backdoor adjustment on s1.W.
>
> We found that misspecification does not necessarily degrade performance. Overspecification yields slightly improved accuracy: the adjustment set in the overspecified graph is valid, if redundant, and the larger number of parameters can improve fit to data. In contrast, the underspecified graph harms accuracy, as the model cannot learn the conditional P(c1.B | p1.X, s1.W) needed for adjustment. We will include these results in our appendix.
>
>
> **W1, Q3: Continuous attributes**
>
> Our identification theory in Sec. 4 does not assume discrete variables; the assumption arises only in our neural approach in Sec. 5, which follows discrete canonical SCM constructions [1]. Existing neural causal approaches for continuous variables (e.g., images [2], robotics [3]) and a recent work on continuous canonical SCMs [4] can, in future work, be used to extend our approach, which we will discuss.
>
>
> **Q4: Larger relational structures**
>
> The scalability of our method arises from parameter sharing across instances of the same type. To evaluate this, we conducted an additional experiment training on skeletons with up to 50 objects and evaluating on skeletons with up to 75 objects, following Exp. 6.2.
>
> *Details.* We train on a source with n objects and evaluate on two distinct targets: one with n objects and one with 1.5n objects, all with similar neighborhood sizes but different topology. We compute a large-scale aggregate query: the average of P(c.B = 1 | do(p.X = 1)) for all p, c such that InPath(p, c).
>
> *Results.* Accuracy remains stable and high with an increasing number of objects. While runtime increases with the number of objects, the growth is modest beyond small sizes, suggesting that parameter sharing mitigates the combinatorial explosion typically associated with increased dimensionality.
>
>
> **W1: Scaling of histogram aggregation**
>
> We agree that histogram aggregation increases dimensionality and can limit scalability. We used it in our experiments as a stress test, since it creates a more challenging estimation problem for RNCMs.
>
> Notably, our method does not require histogram aggregation. In practice, relational machine learning methods often use lower-dimensional summaries (e.g., sum, mean, max, attention pooling) which scale better to large neighborhoods. We will clarify this in our paper.
>
> Following reviewer q4Bq’s suggestion, we conducted an additional experiment where data is generated using histogram aggregation but estimation is performed with a lower-dimensional summary. We find that accurate estimation is still possible under these compressed representations, indicating robustness to aggregation misspecification and better reflecting practical settings.
>
>
> We hope these clarifications shed light on the practical applicability and extensibility of our work.

---

> > ### Author Rebuttal · Reviewer_Bpzd · 2026-04-02
> >
> > I thank the authors for the rebuttal. All of my concerns have been resolved, and I maintain my initial score.

---

> > > ### Author Response · Authors · 2026-04-05
> > >
> > > We thank the reviewer once again for their perceptive feedback, and are glad their concerns have been resolved!

---

### Decision · Program_Chairs · 2026-04-30

**Decision:**

Accept (regular)

**Comment:**

All reviewers agree that extending the framework of structural causal models to a setting where objects and their relations vary is an interesting research question. The proposed solution of relational causal graphs is well motivated, clearly defined, and it is appreciated that the authors discuss both their limitations and identifiability. Not all is perfect, though. The instantiation, relational neural causal models, is not well-developed, especially in terms of formal analysis or guarantees. The empirical evaluation is limited, and does not convincingly investigate the boundaries of where we would expect the method to shine. (The results in Figure 3 are unreadable.) During the rebuttal, the authors took the opportunity to alleviate the main concerns of the reviewers. I agree with their positive leaning and recommend the paper for acceptance. I do wish to urge the authors to make sure that the camera ready version indeed includes a simplified notation, adds (running) examples, extends the experiments, and clearly positions the paper with regard to related work.